



# Time-varying copula and design life level-based nonstationary risk analysis of extreme rainfall events

**Authors:** Pengcheng Xu[1], Dong Wang[*2], Vijay P. Singh[3], Yuankun Wang[2], Jichun Wu[2], Huayu Lu[1], Lachun Wang[1], Jiufu Liu[4], Jianyun Zhang[4]

[1]School of Geographic and Oceanographic Science, Nanjing University,

Nanjing, P.R China

[2]Key Laboratory of Surficial Geochemistry, Ministry of Education, Department of

Hydrosciences, School of Earth Sciences and Engineering, State Key Laboratory of

Pollution Control and Resource Reuse, Nanjing University, Nanjing, P.R. China

[3]Department of Biological and Agricultural Engineering, Zachry Department of

Civil Engineering, Texas A & M University, College Station, TX77843, USA; and

National Water Center, UAE University, Al Ain, UAE

[4]Nanjing Hydraulic Research Institute, Nanjing, P.R. China

(*Corresponding author: Dong Wang, wangdong@nju.edu.cn;

Huayu Lu; huayulu@nju.edu.cn)



**Key points:**

- The time-varying GEV model and copula models are developed for marginal and multivariate frequency analysis, respectively.

- A design life level-based risk analysis is implemented for hydraulic engineering practice.

- A systematic risk analysis incorporating nonstationarity is emphasized in comparison with stationary models.



**Abstract:** Due to global climate change and urbanization, more attention has been paid to decipher the nonstationary multivariate risk analysis from the perspective of probability distribution establishment. Because of the climate change, the exceedance probability belonging to a certain extreme rainfall event would not be time invariant any more, which impedes the widely-used return period method for the usual hydrological and hydraulic engineering practice, hence calling for a time dependent method. In this study, a multivariate nonstationary risk analysis of annual extreme rainfall events, extracted from daily precipitation data observed at six meteorological stations in Haihe River basin, China, was done in three phases: (1) Several statistical tests, such as Ljung-Box test, and univariate and multivariate Mann-Kendall and Pettist tests were applied to both the marginal distributions and the dependence structures to decipher different forms of nonstationarity; (2) Time-dependent Archimedean and elliptical copulas combined with the Generalized Extreme Value (GEV) distribution were adopted to model the distribution structure from marginal and dependence angles; (3) A design life level-based (DLL-based) risk analysis associated with Kendall's joint return period ($JRP_{ken}$)and AND's joint return period ($JRP_{and}$) methods was done to compare stationary and nonstationary models. Results showed DLL-based risk analysis through the $JRP_{ken}$ method exhibited more sensitivity to the nonstationarity of marginal and bivariate distribution models than that through the $JRP_{and}$ method.

**Key words**: multivariate risk analysis; time-varying copula; design life level; nonstationarity; Kendall's joint return period





## 1. Introduction

Due to climate change and increasing urbanization, heavy rains-induced floods have occurred more frequently all over the world in recent decades, which is becoming a major deterrent to the sustainable development of social economy (Mishra and Singh, 2009; Donat et al., 2016; Ali and Mishra, 2018). Various kinds of social activities, such as infrastructure constructions, agricultural irrigation and ecological maintenance would be influenced by hydrometeorological extreme events. A systematic risk analysis of these extreme events would provide sufficient strategies for decision makers.

A multitude of studies have addressed the effect of climate change and urbanization on hydrological design to alleviate associated risks. Traditional hydrological frequency analysis or risk analysis is based on the stationary assumption, which recommends that environmental impact indexes, such as climatic factors and land use rate, have a constant mechanism or pattern that affects hydrological variables all the time (Madsen et al., 2017; Milly et al., 2015). The feasibility of hydrological frequency and risk analysis based on stationary assumptions is being challenged because of the multiple effects of climate change, urbanization, and heat island effects. Accordingly, water authorities should amend the present planning, design and management strategies to develop nonstationary distribution models based on the signals of climate change. Therefore, it is urgent to develop an efficient and systematic risk analysis approach from time dependent side to serve for hydraulic design of hydrological infrastructures to cope with the effect of climate change.





In recent years, nonstationary hydrological frequency analysis has received a great
deal of attention because of increasing attention to climate change (Chen and Sun, 2017;
Call et al., 2017; Ghanbari et al. 2019). The time-varying moment approach is widely
used to involve time variant probabilistic parameters for mimicking the changing
behavior of extreme hydrometeorological variables. Nonstationarity modeling of
probability distribution has been conducted for univariate cases in recent years (Zhang
et al., 2015; Ganguli and Coulibaly, 2017; Agilan and Umamahesh, 2018).
Du et al. (2015) modelled nonstationary low-flow series in Weihe River basin,
China, based on the Generalized Additive Models in Location, Scale and Shape
(GAMLSS) framework. Results showed that inappropriately estimated statistical
parameters would lead to the overstatement of risk corresponding to a low-flow event.
Gu et al. (2016) incorporated time, climate indices, precipitation, and temperature into
the GAMLSS model to detect nonstationarity in flood frequency. For the univariate
case, nonstationary risk analysis, based on the time-varying moment approach, can be
decomposed into four steps: (1) Descriptive and exploratory monitoring of hydrological
sequences and monitoring of outliers; (2) implementation of the stationarity hypothesis
to verify the nonstationarity of hydrological series; (3) development of a hydrological
frequency analysis model and estimation of model parameters using different covariates;
and (4) risk assessment based on the selected frequency model.
The above studies were conducted under nonstationary conditions for univariate
cases, while it is known that natural hydrometeorological extreme events are

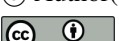



multivariate, characterized by multi-attribute properties which can be statistically
correlated. For instance, floods are characterized by volume, peak, and duration, while
extreme rainfall events have the attributes of duration, intensity, total amount. As a
result, univariate nonstationary risk analysis cannot fully encompass the dependence
structure between hydrological attributes. It is therefore desirable to develop a
multivariate model to simulate the probabilistic behavior of two or more properties.
Copulas, a useful tool for modelling the structure of dependence between hydrological
variables regardless of the types of marginal distributions, have been widely used for
multivariate frequency analysis of rainfall extreme events (Zhang and Singh, 2007; Kao
and Govindaraju, 2008; Rauf and Zeephongsekul, 2014; Vandenberghe et al., 2010);
droughts (De Michele et al., 2013; Serinaldi et al., 2009; Shiau, 2006; Song and Singh,
2010; Wong et al., 2010); floods (Grimaldi and Serinaldi, 2006; Zhang and Singh, 2006).
However, these studies assumed a time invariant dependence pattern, ignoring the
influence of climate change and hence did not consider the impact of nonstationarity
on the dependence structure.

Recently, studies on multivariate distribution fitting have addressed the superiority

of dynamic copula-based method to model the nonstationary dependence structure,
which are generally caused by complex environment and rapid urbanization (Milly et
al., 2015). Former studies have detected nonstationarity in dependence structures (Liu
et al. 2017; Assia et al., 2014; Yilmaz and Perera, 2014). Chebana et al. (2013) argued
that it was necessary to determine a multivariate distribution model quantifying the





time-varying dependence structure of various kinds of hydrological variables. Bender
et al. (2014) used a bivariate nonstationary multivariate model with a 50-year moving
time window to investigate the time-dependent behavior in bivariate case. Their results
showed that the joint probability varied significantly over time for different non-
stationary models. Jiang et al. (2015) also did a multivariate risk analysis using the
time-varying copula method incorporating time and reservoir index as covariates for
low-flow series extracted from two neighboring observed stations.

Traditional solutions of hydrological extreme events involve return period-based

methods, which are usually calculated as the inverse of annual exceedance probability
for a given magnitude under stationary conditions in a univariate case. In a multivariate
case, the univariate return period can be extended to joint return periods of hydrological
variables. There are three kinds of joint return period methods to quantify the
exceedance probability of a multivariate extreme event: the OR method that at least one
extreme attribute is larger than the specified threshold; the AND method that all the
attributes are larger than the specified threshold; and the Kendall method that the
univariate value derived from the Kendall distribution function according to a specified
value (Jiang et al., 2015; Salvadori and Michele, 2010; Salvadori et al., 2013). While
non-stationary distribution models provide flexibility to analyze the variability of a
hydrological variable, they are also incongruent with many of the traditional metrics
used in water resources planning. For example, the development of drainage standards
are vulnerable to the standard of extreme rainfall return period, which means drainage





facilities have been designed to withstand the extreme rainfall event of a specified
return period. The multivariate hydrologic and hydraulic design can be influenced by
the existence of nonstationarity in both the marginal and joint distributions. The
exceedance probability of a given extreme event would be different from year to year,
leading to a nonconstant and non-unique value of the conventional return period. Thus,
the notion of static return period of an extreme rainfall event (e.g., 100-year extreme
rainfall event, 200-year extreme rainfall event) is no longer reliable for hydraulic design
under nonstationary conditions (Salas and Obeysekera, 2014; Yan et al., 2017). As a
result, Rootzén and Katz (2013) first mentioned the concept of design life level (DLL)
to quantify the risk of a given extreme rainfall magnitude over the hydrological
structure's life time (Note that following the idea of Rootzén and Katz (2013) we regard
the term hydrological risk as the possibility of a certain extreme event occurring and
not as a quantification of expected losses). It is a logical extension to handle the
nonstationarity of the concept of "risk of failure" (Jakob, 2013), which is more
frequently used to quantify the risk of hydrologic extremes under stationarity. Read and
Vogel (2015) extended the DLL method to average annual reliability (AAR) method to
estimate the hydrologic design value considering nonstationarity. In general, these risk-
based methods can provide similar results of hydraulic design for hydrological
infrastucture (Yan et al., 2017). However, the cases of multivariate hydrologic designs,
especially under nonstationary conditions using the time-varying copula, and design
life level-based risk methods have great potentiality in future studies.



106  Therefore, the objective of this study is to do risk analysis of multivariate extreme

107 rainfall events involving the following steps. First, a series of statistical tests, such as

108 Ljung-Box test, and univariate and multivariate Mann-Kendall and Pettist tests are used

109 for both the marginal distributions and the dependence structures to determine different

110 forms of nonstationarity (sudden jump, periodicity, and trend). Second, a nonstationary

111 multivariate probability distribution is developed using a time-varying GEV and

112 copula-based model, which can encompass the nonstationarities probably existed in

113 marginal and joint distributions. Finally, design life level-based risk analysis is

114 extended to multivariate cases through Kendall's joint return period and AND's joint

115 return period methods. In this paper, we investigated two kinds of extreme rainfall

116 attributes: (1) annual extreme rainfall volume ($Ps$: Annual total precipitation of the

117 daily precipitation more than the 95th percentile threshold) and intensity ($Im$: Annual

118 maximum daily precipitation), through the nonstationary multivariate risk analysis

119 method. The remainder of this paper is organized as follows. The next section presents

120 the methodology adopted in this study. Section 3 discusses the results of proposed

121 model applied to Haihe River basin, China. Section 4 presents the final conclusion

122 through the proposed model.

123 **2. Methodology**

124  Copulas are tools to build multivariate distribution models of dependence

125 structures between random variables regardless of their marginal distribution types.

126 Detailed information about copulas can be found in Nelson (2007). The present copula-





based methods to solve the multivariate risk analysis mostly adopt static parameters for
whether the marginal distribution or joint distribution. The changing climate has led to
nonstationarity of individual hydrological series or the dependence between
hydrological variables. To realize this situation, a time-varying copula-based model can
describe the time dependent characteristics for dependence structure of hydrological
variables, as inspired by Patton (2006) from the financial field.

Let $(x, y)$ represent a hydrological pair. The joint probability distribution of

multivariables through time-varying copula model can then be presented as:
$$F_{X,Y}(x, y) = C[F_X(x|\theta_X^t), F_Y(y|\theta_Y^t)|\theta_C^t] = C(u, v|\theta_C^t) \tag{1}$$

where $C(\cdot)$ denotes the copula function; $F_{X,Y}(\cdot)$ denotes the joint function; $F_X(\cdot)$
and $F_Y(\cdot)$ represent the marginal functions of hydrological variable (*Ps* and *Im* in this
study); $\theta_X^t$ and $\theta_Y^t$ represent the time-varying marginal distribution parameters; $\theta_C^t$ is
the dynamic copula parameter which is a linear function of time; and $u$ and $v$ are the
marginal probabilities in the time-varying copula in the hypercube unit.

In the framework of multivariate risk analysis (**Figure 1**), the property of

nonstationarity can be determined not only by one or two marginal variables but also
in the dependence structure or vice versa. It is however possible that the nonstationary
behavior may exist in both the marginal and joint distribution function. To determine
the nonstationarity (mutation, cyclicity and trend) in the synthetized extreme rainfall
attribute series, statistical tests, such as Ljung-Box test, and univariate and multivariate
Mann-Kendall and Pettist tests are used for both the marginals and the dependence





structure. Details of these tests can be found in the references due to (Serinaldi and
Kilsby, 2016; Chebana et al., 2013; Rizzo and Székely, 2010).

As shown in **Figure 1**, the time-varying copula-based risk analysis model can be

decomposed into three main phases: (1) detection of nonstationarity in the marginal
variables and dependence structure through a series of nonparametric tests; (2)
estimation of the time-varying parameter for the marginal and joint probabilty
distributions; and (3) joint return period and risk analysis by design life level-based risk
methodology from the perspectives of Kendall's and AND's return period methods
(detailed information can be found in Section 2.3).

Insert **Figure 1** here.


*2.1. Time-varying marginal distribution*

In this part, the Generalized Extreme Value (GEV) distribution was used to

establish time-varying marginal distribution model for the extreme rainfall attributes
because it is a good aggregation of the Gumbel, Fréchet, and Weibull distributions and
is especially suitable for extreme data sets (Cheng and AghaKouchak, 2014). Let $F(x)$
be the cumulative probability distribution function (CDF) of the quantity of interest, $Ps$
or $Im$, in this study. The GEV distribution consists of three control parameters, the
location, the scale, and the shape, which describe mean value of the sample series,
amplitude near the location, and the tail of the distribution, respectively. The cumulative

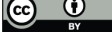



distribution of GEV model under stationary conditions can be expressed as follows:

$$
F(x) = \begin{cases} \exp\left\{ -\left[ 1 + \kappa \left( \dfrac{x-\mu}{\sigma} \right) \right]_+^{-\frac{1}{\kappa}} \right\} & \text{if } \kappa \neq 0 \\[2mm] \exp\left\{ -exp\left( -\dfrac{x-\mu}{\sigma} \right)_+ \right\} & \text{if } \kappa \rightarrow 0 \end{cases}
\tag{2}
$$

where $z_+$=max{y,0} and
$x\epsilon[(\mu - \sigma)/\kappa, +\infty)$ when $\kappa > 0,$
$x\epsilon(-\infty, (\mu - \sigma)/\kappa]$ when $\kappa < 0,$ and
$x\epsilon(-\infty, +\infty)$ when $\kappa = 0.$
where $\mu$ denotes the location parameter, $\sigma$ is the scale parameter and $\kappa$ is the shape
parameter. In this study, two kinds of nonstationary GEV models (GEVns-1 and
GEVns-2) are developed with the shape parameter being constant. It should be
emphasized that modelling the time variance in shape parameter needs long-term
observations, which are often not available in practice (Cheng et al., 2014). GEVns-1
model considers the time-varying characteristic of the location parameter only, while
GEVns-2 model incorporates the time varying features of both location and scale
parameter. These two nonstationary models regard significant trends as a linear function
of time (in years):
$$\mu(t) = \mu_o + \mu_1 t \tag{3}$$
$$\sigma(t) = \exp(\sigma_o + \sigma_1 t) \tag{4}$$
where the scale parameter is always positive throughout, it is usually calculated on the
basis of a log link function.

In this study, the Bayesian method through the Markov chain Monte Carlo

(MCMC) approach (Cheng et al., 2014) was used to estimate the nonstationary GEV
model. Simultaneously, the Deviance Information Criterion (*DIC*) and Bayes factors
(*BF*) for different stationary and nonstationary models were calculated to select the best
fitted marginal model. The minimum *DIC* value yielded the best performance, while
BF smaller than 1 indicated the best fitting.
*2.2. Time-varying Copula*

To model the dependence structure between annual total precipitation (*Ps*) and

annual daily maximum precipitation (*Im*) under nonstationary conditions, a time-
varying copula was developed. In multivariate hydrological frequency analysis, two
kinds of copulas, named elliptical and Archimedean copulas are widely used in
hydrological applications. In this study, time-varying elliptical copulas, Student t (St)
copula, as well as the widely-used time-varying Archimedean copulas, time-varying
Clayton, Gumbel and Frank copula, were selected as candidate models to simulate the
time-varying dependence between two extreme rainfall attributes. The Gaussian copula
was not used in this study because of its deficiency in describing dependencies of
extremes (Renard and Lang, 2007).

The copula parameter $\theta_C^t$ can be assumed as a linear function of the time ("year"

in this study) and can be defined as follows:
$$\theta_C^t = \begin{cases} \exp(\beta_o + \beta_1 t) & \theta_c > 0 \\ \beta_o + \beta_1 t & \theta_c \in R \end{cases} \tag{5}$$

where $\theta_c > 0$ denotes the Student t (St), Clayton and Gumbel copula, while $\theta_c \in R$
represents the Frank copula.



The maximum pseudo-likelihood (MPL) method was adopted to estimate the time-
varying copula parameter (Genest et al., 1995). The Corrected Akaike Information
Criterion (AICc; Hurvich and Tsai, 1989) was employed to make a goodness-of-fit,
which is a modified version of AIC for small samples. Obviously, the presence of
nonstationarity in the copula parameter was determined by comparison of the AICc
value.
*2.3. Joint return period and risk analysis based on KEN's and AND's methods*
For hydrological management, engineering administrators focus more on the
return period and risk of failure during the design life of hydraulic structures (Condon
et al., 2015). Inspired by design life level (DLL) method to present the risk proposed
by Rootzén and Katzs (2013), we would like to expand the DLL-based risk to the
multivariate case.
Let *F(X)* be the cumulative probability distribution function (CDF) of the quantity
of interest, in this study, maximum daily precipitation in a year (*Im*). Conventionally,
the *T*-year return level for certain daily precipitation $x_T$ is equal to the (1-1/*T*)-th
quantile of the marginal distribution of *Im* (The probability distribution is the same for
all years in a stationary situation.). Equivalently, on average, one out of *T* years has at
least one daily rainfall that exceeds $x_T$, so that $T(1 - F(x_T)) = 1$(Serinaldi and Kilsby,
2015), and the probability of annual maximum daily rainfall exceeds $x_T$ is 1/*T*.
Then, the hydrological risk *R* (i.e. risk of failure) of a certain hydraulic structure
for a design life of *n* years can be expressed as the probability that at least one rainfall

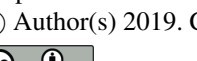



extreme exceeds the design level $x_T$ in a period of $n$ years. Under stationary conditions,
the probability of annual maximum daily rainfall exceeding $x_T$ in every year is the
same as $1/T$. In a univariate context, hydrological stationary risk can be defined as
(Fernandez and Salas, 1999; Serinaldi and Kilsby, 2015):
$R_s = 1 - F(x_T)^n = 1 - (1 - 1/T)^n$                    (6)

Considering time-varying exceedance probabilities, the probability of annual

maximum daily rainfall exceeding $x_T$ in each year is different. So here we use $F_t(x_T)$
to represent the probability of daily rainfall exceeding design level $x_T$ in the $t$-th year.
So the design life level-based nonstationary risk for the univariate case is:
$R_{ns} = 1 - \prod_{t=1}^{n} F_t(x_T)$                    (7)

From the perspective of bivariate case, the joint return period (JRP) of extreme

rainfall events can be calculated through three methods in a stationary situation
(Salvadori et al., 2011). They are AND method corresponding to the probability of
$P(X \geq x \cap Y \geq y)$, OR method corresponding to $P(X \geq x \cup Y \geq y)$, and Kendall
return period method (KEN). Details of the Kendall return period can be found in
Salvadori and De Michele (2004). Since the AND method is widely used and the
Kendall method is of great potentiality, we expanded the AND method and the Kendall
return period method to the nonstationary case here. Let $JRP_{s-and}$ and $JRP_{s-ken}$
represent the three types of return period in the stationary case; they can be calculated
as follows:





$$JRP_{s-and} = \frac{1}{P((X \geq x \cap Y \geq y))} = \frac{1}{1 - F_X(x) - F_Y(y) + C[F_X(x), F_Y(y)]} \qquad (8)$$
$$JRP_{s-ken} = \frac{1}{P\{C[F_X(x), F_Y(y)] \geq p_{ken}\}} = \frac{1}{1 - K_c(p_{ken})} \qquad (9)$$
where $K_c(\cdot)$ is the Kendall distribution function which can be defined as:
$$K_c(p_{ken}) = P\{C[F_X(x), F_Y(y)] \leq p_{ken}\} \qquad (10)$$
Here, $F_X(x)$ and $F_Y(y)$ are the marginal cumulative probability distribution functions
(CDF) for $Ps$ and $Im$, respectively, while $C[F_X(x), F_Y(y)]$ is the bivariate copula
function connecting these two extreme attributes. $p_{ken}$ is just the critical probability
level corresponding to $K_c(p_{ken})$.
Similar to the JRPs of extreme rainfall events under stationary case, the JPRs of
AND and KEN in nonstationary situations can be achieved by:
$$JRP_{ns-and} = \frac{1}{1 - F_X(x|\theta_X^t) - F_Y(y|\theta_Y^t) + C[F_X(x), F_Y(y)|\theta_C^t]} \qquad (11)$$
$$JRP_{ns-ken} = \frac{1}{1 - K_c^t(p_{ken})} \qquad (12)$$
where $\theta_X^t$, $\theta_Y^t$ and $\theta_C^t$ represent the time variant parameters of the marginal and
copula distributions; and $K_c^t(p_{ken})$ is the time-varying Kendall distribution function
corresponding to the time-varying copula.
Multivariate extreme value analysis should be focused on the most likely extreme
event with the largest copula density. The most likely event at the $T_0$-year level can be
calculated as (Graler et al., 2013):
$$(u_m, v_m) = \underset{T_0}{\operatorname{argmax}} c(u, v) \qquad (13)$$
The most likely design combinations $(x_m, y_m)$ can be computed according to the
inverse of marginal cumulative distribution function:



$\quad x_m = F_X^{-1}(u_m)$ and $y_m = F_Y^{-1}(v_m)$ $\hspace{3cm}$ (14)
where $u, v$ are the marginal distribution functions of $X$ and $Y$. Let two pairs of extreme
rainfall attributes $(x_{m_1}, y_{m_1})_{T_0^{and}}$ and $(x_{m_2}, y_{m_2})_{T_0^{ken}}$ be the most likely design
combinations of $Ps$ and $Im$ at the $T_0$-year level for $JRP_{s-and}$ and $JRP_{s-ken}$. Similar
to the nonstationary risk calculation in the univariate case, the hydrological
nonstationary DLL-based risk in the bivariate case can be calculated from two
circumstances:
$\quad R_{ns-and} = 1 - \displaystyle\prod_{t=1}^{n} \left\{ F_X\left(x_{m_1}|\theta_X^t\right) + F_Y\left(y_{m_1}|\theta_Y^t\right) - C\left[F_X\left(x_{m_1}\right), F_Y\left(y_{m_1}\right)|\theta_C^t\right] \right\}$ $\quad$ (15)
$\quad R_{ns-ken} = 1 - \displaystyle\prod_{t=1}^{n} K_C^t(p_{ken})_{(x_{m_2}, y_{m_2})}$ $\hspace{3cm}$ (16)
where $R_{ns-and}, R_{ns-ken}$ indicate the nonstationary risk for a design life level of $n$
years in the bivariate case corresponding to two types of joint return period. The
stationary risk can be calculated in the same way with marginal and copula distribution
parameters being constant.
$\quad$ In this study, comparison of hydrological risk for the bivariate case between
stationary and nonstationary models can be quantified by the risk changing rate $\Delta R_{T_0}^n$
which can be calculated as:
$\quad \Delta R_{T_0}^n = \dfrac{1}{n} \displaystyle\sum_{i=1}^{n} \dfrac{|R_i^{ns} - R_i^s|}{R_i^s}$ $\hspace{3cm}$ (17)
where $R_i^{ns}$ and $R_i^s$ are nonstationary risk and stationary risk of a certain hydraulic
structure for a design life of $i$ years. $\Delta R_{T_0}^n$ helps quantify the difference in risk between
stationary and nonstationary models.



**3.  Application**
*3.1.  Study area and data collection*

The area selected for the study is Haihe River basin, China, which belongs to the

temperate East Asian monsoon climate zone (**Figure 2**). In summer, heavy rains take
place and temperature and humidity are high caused by marine air masses. The annual
rainfall has a great spatial and temporal variability across the basin due to the
inconsistency of intensity, retreat time and influence of the Pacific subtropical high over
the years. Natural disasters, such as urban floods and mountain torrents induced by
extreme rainfall events in the basin have caused huge losses to the social economy and
people's lives and property, and have been highly valued by decision-making authorities.
As a result, time-varying copula-based multivariate risk analysis of this basin is
conducive to providing reliable strategies and alternative options for water resources
risk-based decision making.

Daily rainfall data from Haihe River basin observed at Wutaishan, Fengning,

Zhangjiakou, Beijing, Tianjin, and Nangon were analyzed for the proposed
nonstationary model. Detailed information on these six gauges is presented in Table 1.
According to various data ranges shown in **Table 1**, the rainfall series from 1958-2017
was selected as the final version.

Insert **Figure 2** Here.

Insert **Table 1** Here.






*4.2. Preprocessing Analysis*


Before developing a nonstationary frequency analysis model, it is essential to


examine nonstationarities of extreme precipitation attributes (*Ps* and *Im*) as well as the


structure of dependence between these two attributes. A series of statistical tests (i.e.


Ljung-Box test, univariate and multivariate Man-Kendall tests, and univariate and


multivariate Pettitt tests) were performed to detect the nonstationarity in extreme


precipitation time series. Trends in the time series can be evaluated using various tests


(Lima et al., 2016; Yilmaz et al., 2017; Sarhadi and Soulis, 2017). **Table 2** shows results


of tests detecting nonstationarity, while **Figure 3** shows the spatial distribution of trends


and change points for two attributes of rainfall extremes (*Ps* and *Im*) as well as the


dependence structure between them. First, time series of these two rainfall extremes (*Ps*


and *Im*) for all 6 stations can pass the Ljung-Box test with 20 lags (p.value>0.05 in


**Table 2**). Extreme observations are mutually independent with no serial autocorrelation,


so it is appropriate to apply the standardized Mann-Kendall test to evaluate the


statistical significance of trend without any modification (Serinaldi and Kilsby, 2016).


As shown in **Figure 3**, concurrences of univariate and bivariate trends, the


nonstationarities in rainfall extremes can be detected at several stations (stations 2, 3,


and 4). Station 1 exhibits a significant nonstationarity for extreme attribute *Ps*, while


extreme attribute *Im* and dependence structure show an insignificant decreasing trend.


On the other hand, stations 5 and 6 show a weak decreasing trend. The above tests




totally recommend the presence of nonstationarities in extreme series as well as the
dependence pattern across three out of 6 sites. According to Porporato and Ridolfi
(1998), an insignificant trend should not be ignored because of its effect on the results
of hydrological risk analysis. Hence, even if precipitation extremes at a certain station
may recommend statistically weak trends, both the nonstationary and stationary models
are established for each station in the following section.

Insert **Table 2** Here.

Insert **Figure 3** Here.


*4.3. Marginal distribution fitting*

The nonstationarity can appear either in univariate variables or in dependence

structure in the multivariate framework (Bender et al., 2014). Results of trend and
change point tests performed in Section 4.1 pointed out the necessity to take the
nonstationarity of marginal distributions into consideration. In this study, the
Generalized Extreme Value (GEV) distribution which is a good hybrid of the Gumbel,
Fréchet, and Weibull distributions fits the block or annual maximum time series better
(Cheng et al., 2014). **Table 3(a)-(b)** shows performances of nonstationary vs. stationary
models for these six stations. The location parameter ($\mu$) and scale parameter ($\sigma$) are
regarded as time variant, while the shape parameter $\kappa$ is time invariant; it should be
noted that modeling of time-varying $\kappa$ requires a sufficiently long record of



observations (Cheng et al., 2014). Despite the exception of *Im* for station 4, the shape
parameter $\kappa$ for most fitted models was in the interval of [-0.3,0.3] which is in
accordance with the previous study (Martins and Stedinger, 2000; Ganguli and
Coulibaly, 2017). The best fitted model was selected by performing the minimum *DIC*
criterion combined with the Bayes factor (*BF*) test. For instance, the GEV$_{ns}$-2 model
(nonstationary GEV model with time varying location and scale parameters) was the
best selected model for the extreme attribute *Im* extracted from station 1. That was
because the BF values of GEV$_{ns}$-2 and GEV$_{ns}$-1 were both smaller than 1 which meant
that these two nonstationary models passed the BF test. Then, the best fitted
nonstationary model GEV$_{ns}$-2 for *Im* of station 1 was achieved following the *DIC* test.
Similarly, the best fitted marginal distribution of two extreme rainfall attributes for all
these six stations was selected. Except for stations 4 and 5, the best distributions for the
other stations were parallel for nonstationarity tests shown in Section 4.1.

Insert **Table 3(a)-(b)** here.

*4.4. Copula fitting*
Elliptical and Archimedean (Clayton, Gumbel, and Frank) copulas have been
widely applied in hydrological practice. In this study, time-varying elliptical copulas,
Student t (St) copula, as well as Clayton, Gumbel and Frank copulas were selected as
alternative models to simulate the dependence structures of extreme attributes. The





Gaussian copula was not used in this study because of its deficiency in describing
dependencies of extremes (Renard and Lang, 2007). Once a marginal distribution was
estimated based on test statistics, the dependence structure for *Im* and *Ps* was described
by the time-varying or time invariant copula functions. **Table 4(a)-(b)** illustrates the
results of best fitted copula, based on the minimum AICc and maximum log-likelihood
value (LL). The time-varying Student t (St) copula exhibited the best performance
among the eight candidate copulas (four stationary copulas as well as the corresponding
nonstationary copulas) for stations 1, 2, 3, 4, and 6, while the stationary St copula was
the best one for station 5 which meant that results of dependence structure modelling
for station 5 did not indicate any nonstationarity signal which was reasonable, according
to multivariate MK and Pettit tests of station 5 (**Table 2**). Contrary to station 5, the
nonstationary St copula fitted better than did the stationary model for stations 1 and 6
which was not in accordance with the nonstationarity tests for these two stations (**Table**
**2**). Based on the above results, an insignificant trend or weak change point would lead
to a nonstationary probability function of dependence patterns to some extent
(Porporato and Ridolfi, 1998) which should be dealt with cautiously.
With the best fitted marginal distributions and the best copula, the quantiles of
extreme rainfall attributes (*Ps* and *Im*) were derived from the pseudo-observations
ranging from 0 to 1 in order to provide a benchmark for return period and risk analysis
for hydrological and hydraulic design. The method of analysis is presented in the
following section.




Insert **Table 4(a)-(b)** here.

*4.5. Nonstationary return period and risk analysis for univariate and bivariate cases*

(1) Univariate return period: Once parameters of the best fitted models for
univariate and bivariate cases have been estimated, the extreme rainfall quantiles for
certain return levels (T) can be simulated. In this section, return period and risk analysis
was performed by comparing stationary and nonstationary models. The estimated
rainfall quantiles (*Ps* and *Im*) versus time in the univariate case are shown in Figures 4
for the six stations of Haihe River. *Im* and *Ps* for stations 4 and 6 are not provided,
because the best marginal model for the extreme attributes of these two stations was the
stationary GEV model (**Table 3**). In the case of *Ps* for station 1 shown in **Figure 3**, a
100-year *Ps* quantile under stationary circumstances (GEV$_s$ model with dashed red line
in **Figure 3**) (355 mm) corresponded to a 35-year *Ps* under nonstationary conditions
(GEV$_{ns-2}$) in the year 1960 and a 60-year *Ps* in the year 1970. In other words, an
exceedance probability of 0.01 increased to 0.028 and 0.017. On the other hand, the
return period associated with a given quantile decreased from 1960 to 2020 for *Im* of
station 4 and *Ps* of station 5, while the return period increased for extreme attributes of
other stations. Interestingly, the temporal variability between different stations
corresponding to the best selected nonstationary model exhibited a significant
difference. For example, the nonstationary GEV$_{ns-2}$ model fitted to *Ps* of stations 1, 2,



3, and 5 showed a significant upward or downward trend of extreme quantiles with
years. Compared to the temporal variability, the attributes of stations 4 and 6 with
GEV$_{ns-1}$ model showed a weaker trend which demonstrated the time variability of scale
parameter of the GEV distribution. Finally, it is noteworthy that nonstationary isolines
were over stationary isolines for *Ps* of stations 1 and 5 as well as *Im* of station 3 (marked
in blue star in Figure 4) which meant that the stationary model would underestimate the
risk for a certain return period.
Insert **Figure 3** here**.**
Insert **Figure 4** here**.**

(2) Joint return period (JRP) based on AND and KEN method:
After the nonstationary copula and GEV distribution models were selected
according to several goodness-of-fit tests, the design values characterizing annual
extreme rainfall events were determined through Kendall's ($JRP_{ken}$) or AND's joint
return period ($JRP_{and}$) expressed by Equations (8)-(10). Although the copula model for
station 5 was stationary, it was regarded as a nonstationary model because of the
marginal nonstationary GEV$_{ns-2}$ model for *Ps* or *Im*, which existed at other stations.
Since events with lower exceedance probabilities are of interest for hydrological
practice and the joint return period of 50-year level is able to minimize the uncertainties
of extrapolation. In this study we focused typically on events with a joint return period
of $JRP_{ken}$ ($JRP_{and}$)=50 which means the exceedance probability was equal to 0.02.





**Figure 5** shows isolines of Kendall return period and AND-based return period at the
50-year level for both stationary and nonstationary models. Since the number of isolines
of the nonstationary model were 60 (sample size) which might show certain
overlapping areas in the isoline map, four isolines corresponding to the year 1960, 1980,
2000, and 2020 are presented for simplicity. For comparison, the $JRP_{ken}(JRP_{and})$-
isolines derived from the corresponding stationary model, which was composed of
stationary GEV and stationary copula model, are also shown. Observations belonging
to each station are also presented. Although the plots for all the years are not shown,
the variability of design quantiles over time showed the nonstationary behavior of the
dependence structure.

From **Figure 5**, $JRP_{ken}$ was larger than $JRP_{and}$ for the dependence structure of

the same extreme rainfall attributes, which was caused by Kendall's return period
method of generating the same dangerous region, regardless of different realizations
(Salvadori et al. 2011). Focusing on $JRP_{ken}$ and $JRP_{and}$ for station 1, the design
values of *Ps* varied over time, while the design values of *Im* did not vary with time.
From the horizontal direction, both the $JRP_{ken}$-isolines and $JRP_{and}$-isolines exhibited
a left-moving trend, recommending a descending trend for *Ps*. The maximum *Ps* values
for the year 1960 were measured as 341.6 mm and 371.5 mm corresponding to $JRP_{ken}$
and $JRP_{and}$=50, respectively, while 246.4 mm and 264.8 mm is calculated as the
minimum marginal values. The gap between them reached 100 mm. On the other hand,
none of the $JRP_{ken}$ and $JRP_{and}$-isolines exhibited a variation trend of *Im* values for





station 1 from the vertical perspective, which can be attributed to the stationary GEV
model for *Im* of station 1 (**Table 3**). Due to sudden changes in the magnitudes of
marginal values, the Kendall isolines also crossed each other. In a similar way, the
nonstationary behavior of the variables was detected from the variation of design values
of extreme attributes at the other five stations compared to the isolines derived from the
stationary model (denoted as black line). **Figure 5**). It is noteworthy that the stationary
copula model for station 5 also exhibited the variation of design values derived from
both $JRP_{ken}$ and $JRP_{and}$. A weak variation of 22.3 mm for *Ps* and 4.9 mm for *Im* was
detected because of the corresponding nonstationary GEV model (**Table 3**).

Insert **Figure 5** here**.**

(3) Univariate risk
The hydrological risk of a certain design extreme attribute quantile $x_{T_0}$ can be
computed using Equations. (6) and (7) on the basis of the initial return period $T_0$ and
design life *n*. The best marginal distribution model for *Im* of station 1 as well as *Ps* of
stations 4 and 6 were the stationary GEV model, so these three scenarios were not taken
into consideration in this part. Except for the results of Ps of station 5, the risk results
of extreme attributes of other five stations were very similar (**Figure 6**). Here, we
considered the risk result of the attribute *Im* of station 2 for detailed illustration.
Comparing the risk of stationary and nonstationary models, a definite conclusion can



be addressed: risk can increase from the stationary condition to nonstationary condition.
For example, when $T_0$= 50 and n=20, the risks for the stationary and nonstationary
conditions were 33.24% and 46.8%, respectively. That is to say an unjustified
assumption would lead to an overestimation of the risk under a certain return period
and design life. For $Im$ of station 2, the nonstationary risk was higher than the stationary
risk when $n \leq 53$. On the other hand, the nonstationary risk was smaller than the
stationary risk when $n \geq 53$. This conclusion can be detected from the $Ps$ of station 1.

Once $T_0$ was decided, the risk changing rate was calculated by equation (17).

Here, $T_0$ was set as 50 for illustration. For attribute $Im$, the risk changing rate $\Delta R_{T_0}^n$
corresponding to $T_0$=50 was 45.93%, 5.31%, 18.25%, 39.44% and 37.10% for stations
2, 3, 4, 5 and 6, respectively. For attribute $Ps$, $\Delta R_{T_0}^n$ was 61.26%, 22.47%, 59.51% and
20.53% for stations 1, 2, 3 and 5. Generally, $Im$ of station 2 and $Ps$ of station 1 should
be paid more attention with the highest risk changing rate in hydrological practice.

Insert **Figures 6** here**.**


(3) Bivariate risk based on $JRP_{ken}$ and $JRP_{and}$

The hydrological nonstationary risk in the bivariate case cannot be calculated until

the most likely event at $T_0$-year level is generated. In this part, we first focused on the
development of the most likely design events where the joint probability density
functions had their maximum values on the 50-year level. **Figures 7(a)** and **(b)** illustrate





the time dependent development of both variables $Ps$ (upper panel) and $Im$ (lower panel)
through the $JRP_{ken}$ and $JRP_{and}$ methods. The attribute $Ps$ for stations 1, 2, and 3
showed a positive trend, while attribute $Im$ for stations 4 and 5 exhibited a negative
trend through the $JRP_{ken}$ and $JRP_{and}$ methods. On the other hand, the trend of the
design value of Im was not significant.
The hydrological nonstationary risk based on $JRP_{ken}$ and $JRP_{and}$ was
computed using equations (15)-(16). **Figures 8(a)** and **(b)** show the nonstationary risk
$R$ of the most likely design combinations of $Ps$ and $Im$ at the $T_0$-year level. The risk
results of extreme attributes of stations 1, 3, 4, and 6 were very similar, while the results
of stations 2 and 5 exhibited a similar pattern. For stations 1, 3, 4, and 6, the risk
increased with design life $n$ under both stationary and nonstationary conditions, but for
any $T_0$, the nonstationary risk was higher than stationary risk from both $JRP_{ken}$ and
$JRP_{and}$ methods. For stations 2 and 5, the nonstationary risk was lower than stationary
risk from both the $JRP_{ken}$ and $JRP_{and}$ methods. The corresponding changing rate to
quantify the differences in hydrological risk for the bivariate case between stationary
and nonstationary models was also calculated by equation (17). Whether it was
calculated through the $JRP_{ken}$ and $JRP_{and}$ methods, the changing risk rate increased
as $T_0$ increased, which meant that nonstationarity influenced the risk of lower
exceedance probabilities more than that of higher exceedance probabilities.
Since a 50-year level with lower exceedance probabilities (0.02) is of great interest
in hydrological practice and necessary to control the uncertainties of extrapolation


(Bender et al., 2014), in this part, we focused on the risk changing rate under the 50
year-level for each station. For $JRP_{and}$, the risk changing rate $\Delta R_{T_0}^n$ corresponding to
$T_0$=50 was 41.83%, 7.96%, 24.27%, 6.94%, 9.21%, and 70.63% for stations 1, 2, 3, 4,
5, and 6, respectively. For $JRP_{ken}$, the risk changing rate $\Delta R_{T_0}^n$ corresponding to
$T_0$=50 was 59.93%, 8.44%, 44.19%, 10.69%, 11.96%, and 75.29% for stations 1, 2, 3,
4, 5, and 6, respectively. According to the above results of risk changing rate, changing
risk rates based on the $JRP_{ken}$ method were higher than those through the $JRP_{and}$
method, which indicated that the $JRP_{ken}$-based risk was more sensitive to the
nonstationarity of marginal and bivariate distribution models.

Insert **Figures 7(a)-(b)** here**.**

Insert **Figures 8(a)-(b)** here**.**


*4.6. Further discussion*

Based on the above analysis, the nonstationary risk analysis over extreme rainfall

events using the time-varying GEV and copula-based distribution models were
distinguished from those where the partial assumption of stationarity was employed
(**Figures 4-8**). These results showed the significance of considering nonstationarity
when calculating return period and hydrological risk both in univariate and bivariate
cases. There were also certain differences between the results using Kendall's joint
return period method and AND's return period method. *Im* of station 2 and *Ps* of station



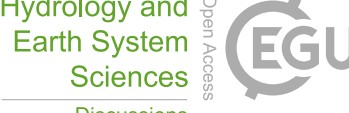

1 should be concerned with the highest risk changing rate from the perspective of
univariate case, while the dependence structure of station 6 should be paid more
attention with the highest risk changing rate from the perspective of bivariate case.

According to the results performed by the proposed time-varying models, the

following points should be emphasized:
✓    It is necessary to use statistical tests, such as the Ljung-Box test, univariate

and multivariate Mann-Kendall test, and univariate and multivariate Pettitt

test to evaluate nonstationarities of extreme rainfall attributes ($Ps$ and $Im$) as

well as the dependence between these two attributes. These two attributes

corresponding to six stations showed no serial correlation which rationalized

the implementation of traditional multivariate Mann-Kandall tests without

any modification.

✓    Nonstationarity in the dependence structure and marginal variable was non-

ignorable. The nonstationary (time-varying) GEV and copula-based model

not only addressed the abrupt changes and significant trends existed in the

marginal variables, but also evaluated the dependence of multivariate

hydrological series, which led to the reliable estimation of hydraulic design

quantiles.

✓    The traditional hydrological risk under nonstationary conditions in the

univariate case was expanded to the bivariate case through the Kendall joint

return period method and AND return period method. According to the return



period analysis in the univariate case, the scale parameter of the nonstationary
GEV distribution demonstrated a significant time variability for uncertainty.
The joint return period and risk analysis also showed that the $JRP_{ken}$-based
risk was more sensitive to the nonstationarity of marginal and bivariate
distribution models.
Moreover, the two indexes used in this study, revealing the characteristics of
extreme rainfall events, i.e., *Ps* and *Im*, representing rainfall volume and intensity,
respectively were extracted from observed daily precipitation datasets. Risk
analysis based on these two attributes helped understand extreme rainfall patterns,
especially storm events lasting several days, which would be devastating to urban
infrastructure and farmlands. In addition, the duration which is another meaningful
extreme rainfall attribute should also be incorporated into multivariate risk
analysis.
**5. Conclusions**
In this paper, a nonstationary risk analysis through the time-varying Generalized
Extreme Value (GEV) and copula-based distribution model is performed over the
extreme rainfall events in Haihe River Basin. The time-dependent copula and GEV
models are applied to these two attributes (*Ps* and *Im*) extracted from daily rainfall data
of six stations in Haihe River basin, China. Nonstationarity and trends in the attribute
series were investigated through multivariate Mann-Kendall test and multivariate
Pettist test. The best nonstationary GEV model was selected for the attribute of each





station through the minimum *DIC* criterion combined with the Bayes factor (*BF*) test,
while the best-fitted time-varying copula was selected through the minimum Corrected
Akaike Information Criterion (AICc). Based on frequency analysis by the Kendall joint
return period method and the AND return period method, the design values of the two
indexes were computed and shown by the $JRP_{ken}$-isolines and $JRP_{and}$-isolines. The
extended bivariate nonstationary DLL-based risk was calculated through the estimated
most likely event (combinations of *Ps* and *Im*) to quantify the risk of each station under
nonstationary conditions. Analysis of extreme rainfall occurrence risk based on the
observed index series demonstrated that station 6 should be paid more attention with
the highest risk changing rate. The following conclusions can be drawn from this study:
1.   A 100-year Ps quantile under stationary conditions (355 mm) can correspond to a
35-year Ps under nonstationary conditions. In other words, an exceedance probability
of 0.01 can increase to 0.028 and 0.017. On the other hand, the return period associated
with a given quantile can decrease for Im of some stations but can increase for other
stations.
2.  The stationary model would underestimate the risk for a certain return period.
3.  From the marginal return period to the joint return period, there can be a significant

upward or downward trend in extreme quantiles in the univariate case which can

change into a weak trend in the joint return period.

4.  Nonstationarity influences the risk of lower exceedance probabilities more than that

of higher exceedance probabilities.



5. Changing risk rates based on the $JRP_{ken}$ are higher than those based on the

$JRP_{and}$ method, which indicated that the $JRP_{ken}$-based risk is more sensitive to

the nonstationarity of marginal and bivariate distribution models.

This study emphasizes the significance of incorporating nonstationarity into

multivariate risk analysis through the investigation of univariate and multivariate trend
and change points in the attribute series. The Kendall return period is justified as more
practical method for hydraulic design than the AND return period method according to
the calculation of the design quantiles for the extreme rainfall. The extended bivariate
nonstationary DLL-based risk method was applied to both stationary and nonstationary
conditions.

**Acknowledgments**
This study was supported by the National Key Research and Development Program of
China (2017YFC1502704,2016YFC0401501), and the National Natural Science Fund
of China (41571017, 51679118 and 91647203), and Jiangsu Province"333 Project"
(BRA2018060).



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





**Lists of Tables**

**Lists of Figures**

**Figure 1.** Flowchart of this study

**Figure 2.** Selected meteorological stations in Haihe River basin

**Figure 3.** Spatial distribution of trend and changing points according to univariate and multivariate Mann-Kendall and Pettitt tests shown in Table 2; All above tests were performed at a significance level of 10%, i.e. p value< 0.10.

**Figure 4**. Estimated extreme rainfall attribute (*Ps* and *Im*) quantiles for stationary and selected nonstationary model at six stations.

**Figure 5.** Isolines of Kendall return period and AND-based return period at the 50 year level for both stationary and nonstationary models. (1)-(6) represent the station number.

**Figure 6(a).** The most likely design event of Ps and Im with $JRP_{and} = 50$ for six stations

**Figure 6(b).** The most likely design event of Ps and Im with $JRP_{ken} = 50$ for six stations

**Figure 7.** Nonstationary risk $R$ of the Haihe River design extreme rainfall quantile $x_{T_0}$ under univariate case. The nonstationary design life level-based hydrological risk $R$ is regarded as a function of design life $n$ for $x_{T_0}$ with an initial return period $T_0$.

**Figure 8(a).** Nonstationary risk $R$ of the most likely design combinations of *Ps* and *Im* at $T_0$-year level based on $JRP_{s-and}$ and $JRP_{s-ken}$.

**Figure 8(b).** Nonstationary risk $R$ of the most likely design combinations of *Ps* and *Im* at $T_0$-year level based on $JRP_{s-and}$ and $JRP_{s-ken}$.





**Table 1.** Information on meteorological gauges of Haihe River basin

| | Station ID | Station name | Location | | Data range |
|---|---|---|---|---|---|
| | | | Longtitude | Latitude | |
| 1 | 53588 | Wutaishan | 113 °32′ | 39 °02′ | 1952-2017 |
| 2 | 54308 | Fengning | 116 °32′ | 41 °12′ | 1957-2017 |
| 3 | 54401 | Zhangjiakou | 115 °11′ | 40 °50′ | 1958-2017 |
| 4 | 54511 | Beijing | 116 °19′ | 39 °57′ | 1958-2017 |
| 5 | 54527 | Tianjin | 117 °10′ | 39 °06′ | 1958-2017 |
| 6 | 54705 | Nangon | 115 °23′ | 37 °22′ | 1956-2017 |





**Table 2.** Detection of trends and change points in extreme rainfall attributes collected from six stations

| Station No. | Attribute | L-jung-Box Test p.value | Univariate MK p.value | Univariate MK Z statistics | Univariate Pettitt Test p.value | Multivariate MK p.value | Multivariate MK Z tatistics | Multivariate Pettitt Test p.value |
|---|---|---|---|---|---|---|---|---|
| 1 | Ps | 0.927 | 0.063 | -1.856* | 0.048 | 0.450 | -0.755 | 0.089 |
|   | Im | 0.798 | 0.674 | 0.421 | 0.708 | | | |
| 2 | Ps | 0.307 | 0.105 | -1.652* | 0.089 | 0.098 | -1.645* | 0.095 |
|   | Im | 0.462 | 0.132 | -1.649* | 0.091 | | | |
| 3 | Ps | 0.986 | 0.345 | -0.944 | 0.131 | 0.099* | -1.641 | 0.078 |
|   | Im | 0.575 | 0.051 | -1.963** | 0.012 | | | |
| 4 | Ps | 0.981 | 0.072 | -1.799* | 0.098 | 0.055 | -1.922* | 0.039 |
|   | Im | 0.971 | 0.054 | -1.926* | 0.089 | | | |
| 5 | Ps | 0.051 | 0.524 | -0.638 | 0.801 | 0.606 | 0.516 | 0.589 |
|   | Im | 0.747 | 0.214 | -1.244 | 0.678 | | | |
| 6 | Ps | 0.815 | 0.226 | -1.212 | 0.454 | 0.115 | -1.575 | 0.359 |
|   | Im | 0.923 | 0.067 | -1.831* | 0.024 | | | |

**and * represent statistically significant at 5% and 10% significance levels; The standardized Mann-Kendall test statistic (Z statistics) in univariate and multivariate cases indicates positive (negative) with an increasing (decreasing) trend, and statistically significant at 5% and 10% significance levels when |Z|>1.96 and |Z|>1.64 respectively; Change point tests in univariate and multivariate cases are performed at 10% significance level.





**Table 3(a)** Performance of stationary and nonstationary GEV models fitted to for the marginal distribution corresponding to each attribute (stations 1-3)

| Station | Attribute | Model | μ | σ | κ | DIC | BF |
|---|---|---|---|---|---|---|---|
| 1 | Ps | GEV$_s$ | 144.37 | 45.32 | 0.058 | 1944.47 | - |
| | | GEV$_{ns}$-1 | 147.82+0.0014t | 46.67 | 0.033 | 1943.03 | 0.9999 |
| | | **GEV$_{ns}$-2** | **151.85-0.000042t** | **exp(3.92-0.000038t)** | **0.019** | **1942.08** | **0.9991** |
| | Im | **GEV$_s$** | **50.64** | **18.13** | **0.012** | **1593.71** | **-** |
| | | GEV$_{ns}$-1 | 46.69+0.0013t | 17.81 | 0.035 | 1597.58 | 1.0011 |
| | | GEV$_{ns}$-2 | 51.7-0.00062t | exp(2.96-0.000037t) | 0.019 | 1596.53 | 1.004 |
| 2 | Ps | GEV$_s$ | 106.25 | 28.84 | -0.12 | 1740.92 | - |
| | | GEV$_{ns}$-1 | 104.08-0.00018t | 28.83 | -0.12 | 1742.66 | 0.998 |
| | | **GEV$_{ns}$-2** | **99.92+0.0028t** | **exp(3.25+0.000057t)** | **-0.13** | **1742.29** | **0.996** |
| | Im | GEV$_s$ | 40.71 | 14.01 | -0.053 | 1497.533 | - |
| | | **GEV$_{ns}$-1** | **42.02+0.00026t** | **14.30** | **-0.099** | **1493.47** | **0.997** |
| | | GEV$_{ns}$-2 | 42.90-0.00031t | exp(2.39+0.00012t) | -0.082 | 1492.82 | 0.996 |
| 3 | Ps | GEV$_s$ | 86.92 | 26.92 | -0.077 | 1724.35 | - |
| | | GEV$_{ns}$-1 | 88.48+0.000096t | 27.16 | -0.085 | 1724.306 | 1.0002 |
| | | **GEV$_{ns}$-2** | **85.78+0.0014t** | **exp(3.57-0.00014t)** | **-0.091** | **1723.76** | **0.999** |
| | Im | GEV$_s$ | 34.31 | 11.00 | 0.13 | 1455.2 | - |
| | | **GEV$_{ns}$-1** | **37.51-0.00067t** | **11.38** | **0.10** | **1448.41** | **0.995** |
| | | GEV$_{ns}$-2 | 41.76-0.0030t | exp(2.37+0.000019t) | 0.11 | 1449.82 | 0.996 |

**Table 3(b)** Performance of stationary and nonstationary GEV models fitted for marginal distribution corresponding to each attribute (Station 2-6)

| Station | Attribute | Model | μ | σ | κ | DIC | BF |
|---|---|---|---|---|---|---|---|





|  |  |  | | | | | |
|---|---|---|---|---|---|---|---|
| 4 | | GEV$_s$ | **141.69** | **55.87** | **0.19** | **2034.265** | **-** |
|  | $Ps$ | GEV$_{ns}$-1 | 144.73-0.0016t | 55.99 | 0.19 | 2036.30 | 1.0014 |
|  |  | GEV$_{ns}$-2 | 139.80-0.00014t | exp(3.88+0.000060t) | 0.21 | 2037.05 | 1.0019 |
|  |  | GEV$_s$ | 56.36 | 22.29 | 0.34 | 1751.59 | - |
|  | $Im$ | GEV$_{ns}$-1 | **65.73-0.00035t** | **23.17** | **0.31** | **1749.31** | **0.9995** |
|  |  | GEV$_{ns}$-2 | 60.33-0.00055t | exp(3.10+0.000026t) | 0.31 | 1749.48 | 0.9998 |
| 5 | | GEV$_s$ | 160.34 | 56.07 | -0.21 | 1963.47 | |
|  | $Ps$ | GEV$_{ns}$-1 | 148.22+0.0016t | 54.74 | -0.18 | 1962.35 | 1.0002 |
|  |  | GEV$_{ns}$-2 | **150.25+0.0011t** | **exp(4.07-0.000038t)** | **-0.19** | **1962.12** | **0.9999** |
|  |  | GEV$_s$ | 67.46 | 26.37 | 0.072 | 1752.36 | |
|  | $Im$ | GEV$_{ns}$-1 | **77.72-0.0041t** | **26.88** | **0.047** | **1747.30** | **0.9976** |
|  |  | GEV$_{ns}$-2 | 70.19-0.00072t | exp(3.66-0.00019t) | 0.067 | 1749.53 | 0.9979 |
| 6 | | GEV$_s$ | **137.56** | **57.44** | **-0.24** | **1965.79** | **-** |
|  | $Ps$ | GEV$_{ns}$-1 | 133.69+0.00084t | 58.31 | -0.25 | 1968.09 | 1.0017 |
|  |  | GEV$_{ns}$-2 | 138.26-0.00087t | exp(4.25-0.0001t) | -0.25 | 1967.70 | 1.0015 |
|  |  | GEV$_s$ | 63.02 | 29.43 | 0.033 | 1767.72 | - |
|  | $Im$ | GEV$_{ns}$-1 | 69.26-0.0053t | 27.93 | 0.059 | 1767.24 | 1.0001 |
|  |  | GEV$_{ns}$-2 | **57.05+0.000083t** | **exp(3.45-0.000071t)** | **0.072** | **1766.92** | **0.9999** |





**Table 4(a)** Performance of stationary and nonstationary copula models fitted to the dependence structure of two attributes (stations 1-3)

| Station | Model | Copula | $\theta$ | LL | AICc |
|---|---|---|---|---|---|
| 1 | S[a] | St[c] | 0.8066 | 28.83 | -55.59 |
| | | Clayton | 1.9725 | 23.81 | -45.55 |
| | | Gumbel | 2.2677 | 25.08 | -48.08 |
| | | Frank | 8.2702 | 24.88 | -46.55 |
| | NS[b] | **St** | **exp(0.2998-0.0002t)** | **31.42** | **-58.71** |
| | | Clayton | exp(-4.8096+0.0023t) | 22.73 | -43.39 |
| | | Gumbel | exp(-5.268+0.0256t) | 23.63 | -46.49 |
| | | Frank | 7.677+0.00028t | 21.14 | -42.45 |
| 2 | S | St | 0.9000 | 51.55 | -101.05 |
| | | Clayton | 3.7056 | 42.85 | -83.63 |
| | | Gumbel | 3.4208 | 52.42 | -102.77 |
| | | Frank | 13.592 | 45.88 | -89.68 |
| | NS | **St** | **exp(0.4247-0.0003t)** | **53.46** | **-102.786** |
| | | Clayton | exp(-3.289+0.0017t) | 41.38 | -80.59 |
| | | Gumbel | exp(-4.344+0.0356t) | 45.57 | -88.57 |
| | | Frank | 10.592+0.0018t | 46.07 | -93.09 |
| 3 | S | St | 0.8491 | 38.25 | -74.44 |
| | | Clayton | 3.0403 | 35.57 | -69.06 |
| | | Gumbel | 2.6696 | 33.21 | -64.35 |
| | | Frank | 9.4212 | 34.89 | -67.15 |
| | NS | **St** | **exp(-1.6982+0.0008t)** | **41.49** | **-78.84** |
| | | Clayton | *NaN* | *NaN* | *NaN* |
| | | Gumbel | *NaN* | *NaN* | *NaN* |
| | | Frank | 11.385-0.0011t | 35.18 | -68.59 |

[a]S is stationary copula model; [b]NS represents the time-varying copula model; [c]St represents Student's t copula.





**Table 4(b)** Performance of stationary and nonstationary copula models fitted to the dependence structure of two attributes (stations 2-6)

| Station | Model | Copula | $\theta$ | LL | AICc |
|---|---|---|---|---|---|
| 4 | S | St | 0.9001 | 43.10 | -87.13 |
| | | Clayton | 2.944 | 36.68 | -71.30 |
| | | Gumbel | 3.273 | 46.82 | -91.57 |
| | | Frank | 11.462 | 44.68 | -87.67 |
| | NS | **St** | **exp(-1.6101+0.0008t)** | **51.79** | **-99.46** |
| | | Clayton | exp(-3.369+0.0027t) | 38.97 | -75.89 |
| | | Gumbel | exp(-4.564+0.0289t) | 44.68 | -85.98 |
| | | Frank | 12.398-0.00081t | 44.68 | -87.67 |
| 5 | S | **St** | **0.9000** | **47.977** | **-93.886** |
| | | Clayton | 3.2678 | 40.62 | -79.17 |
| | | Gumbel | 3.1054 | 42.94 | -83.82 |
| | | Frank | 11.023 | 44.68 | -87.67 |
| | NS | St | exp(0.5868-0.0003t) | 46.00 | -91.359 |
| | | Clayton | exp(-2.987+0.0037t) | 42.56 | -82.69 |
| | | Gumbel | exp(-3.898+0.0252t) | 41.69 | -81.59 |
| | | Frank | 12.589-0.00036t | 42.57 | -83.79 |
| 6 | S | St | 0.8889 | 43.85 | -83.55 |
| | | Clayton | 3.4955 | 44.77 | -84.47 |
| | | Gumbel | 2.8959 | 37.84 | -73.61 |
| | | Frank | 11.227 | 41.89 | -81.79 |
| | NS | **St** | **exp(0.7846-0.0005t)** | **46.02** | **-87.90** |
| | | Clayton | exp(-3.876+0.071t) | 39.59 | -78.89 |
| | | Gumbel | *NaN* | *NaN* | *NaN* |
| | | Frank | 11.462 | 44.68 | -87.67 |

Inf: infinite number, or out of scope of computation





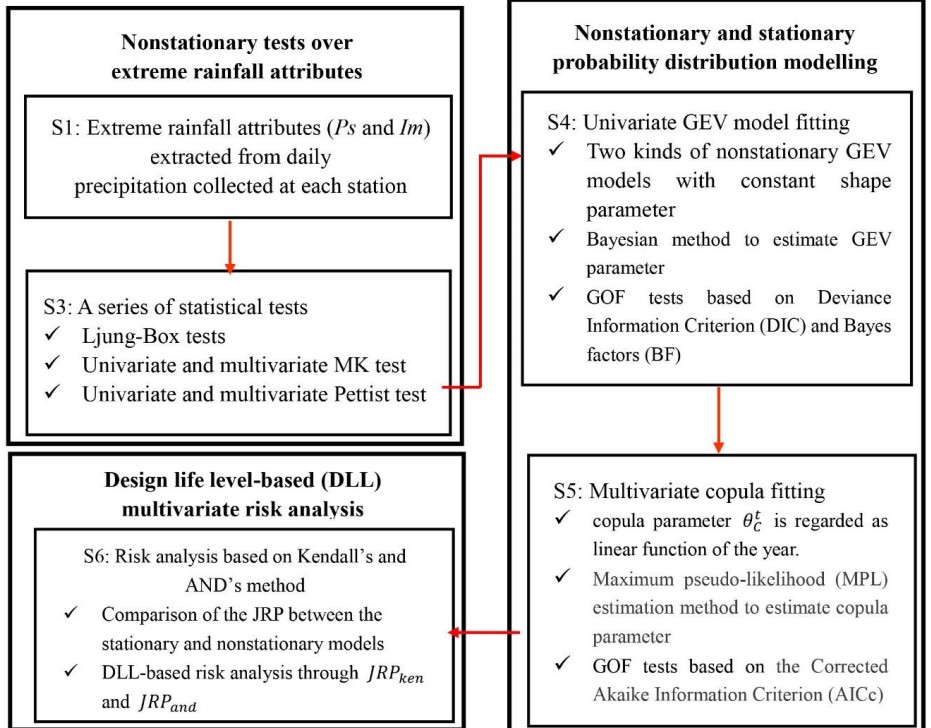

**Figure 1.** Flowchart of this study

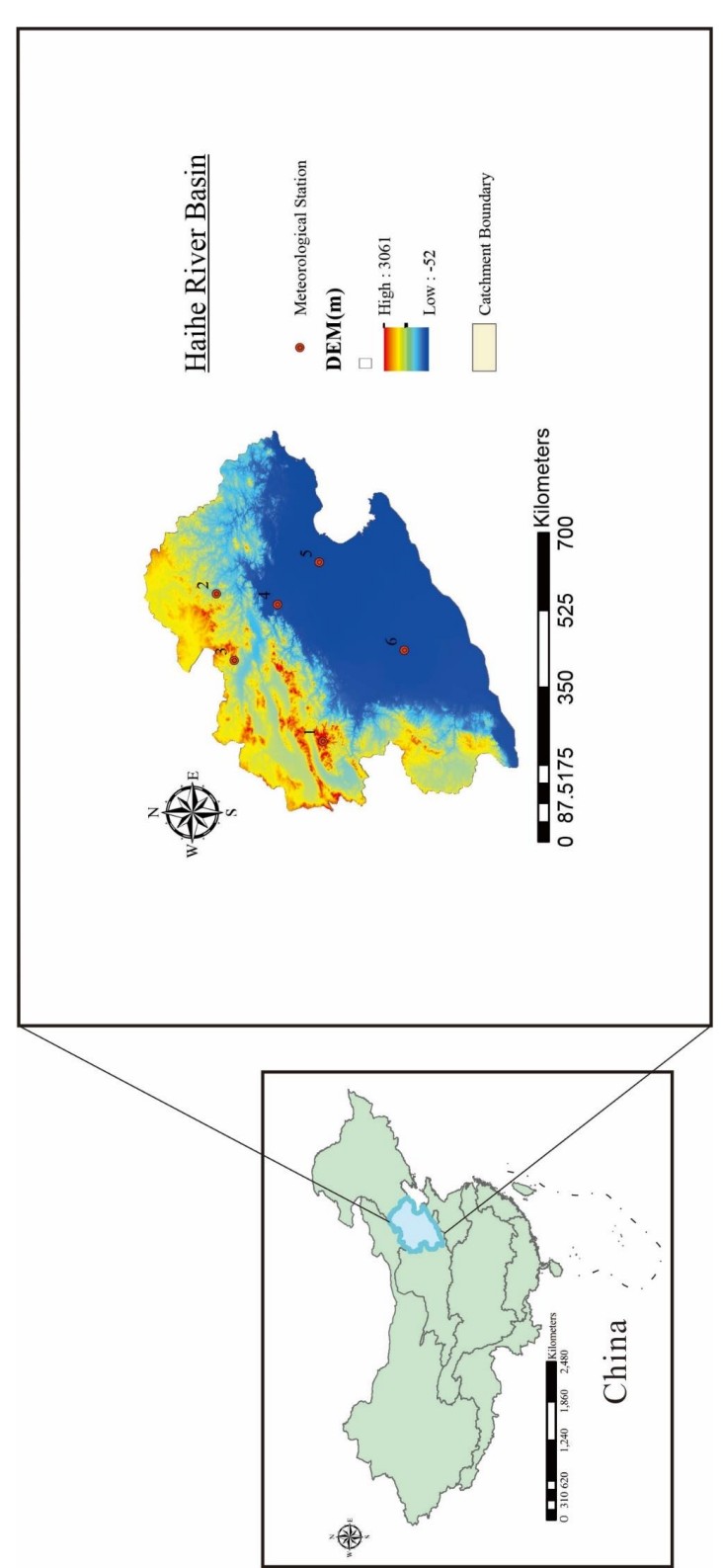

**Figure 2.** Selected meteorological stations in Haihe River basin

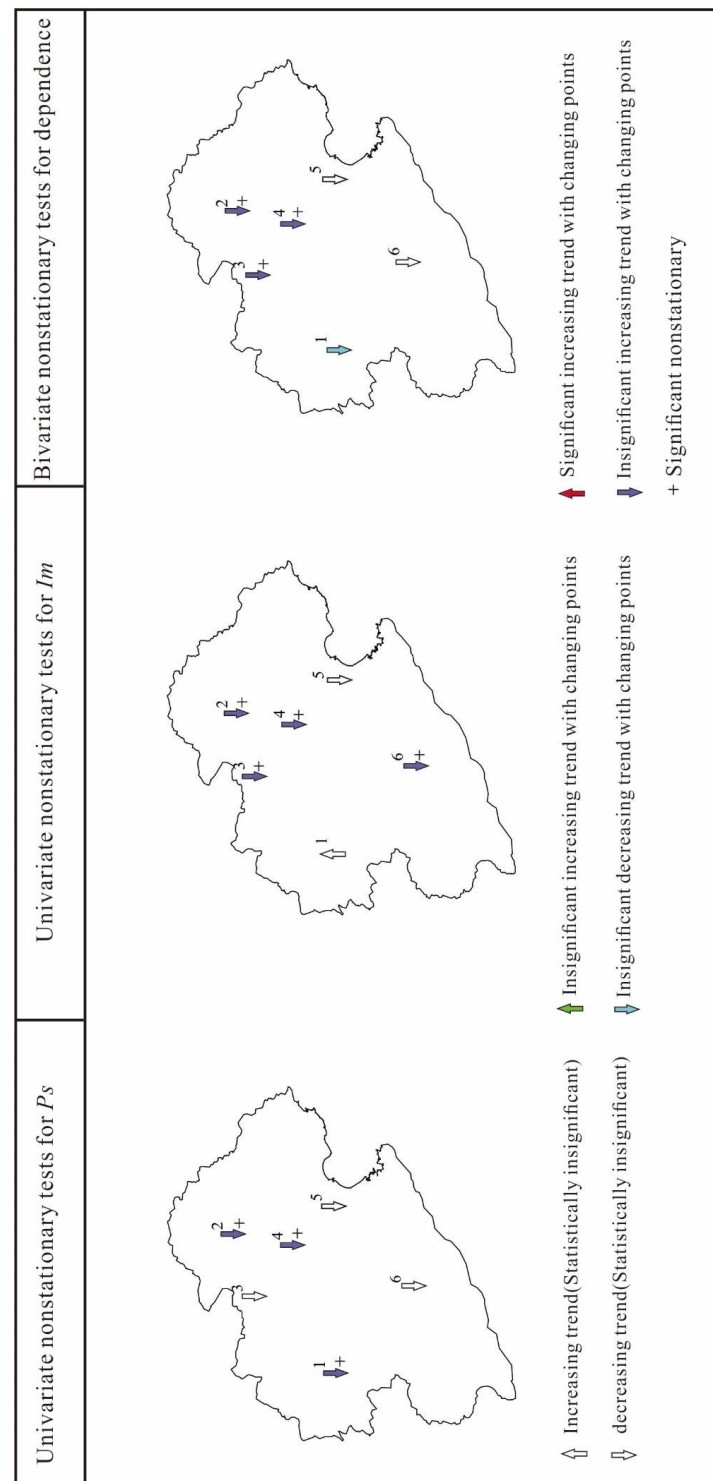

**Figure 3.** Spatial distribution of the trend and changing points according to univariate and multivariate Mann-Kandalld and Pettitt tests shown in Table 2; All above tests are performed at a significance level of 10%, i.e. p value< 0.10.





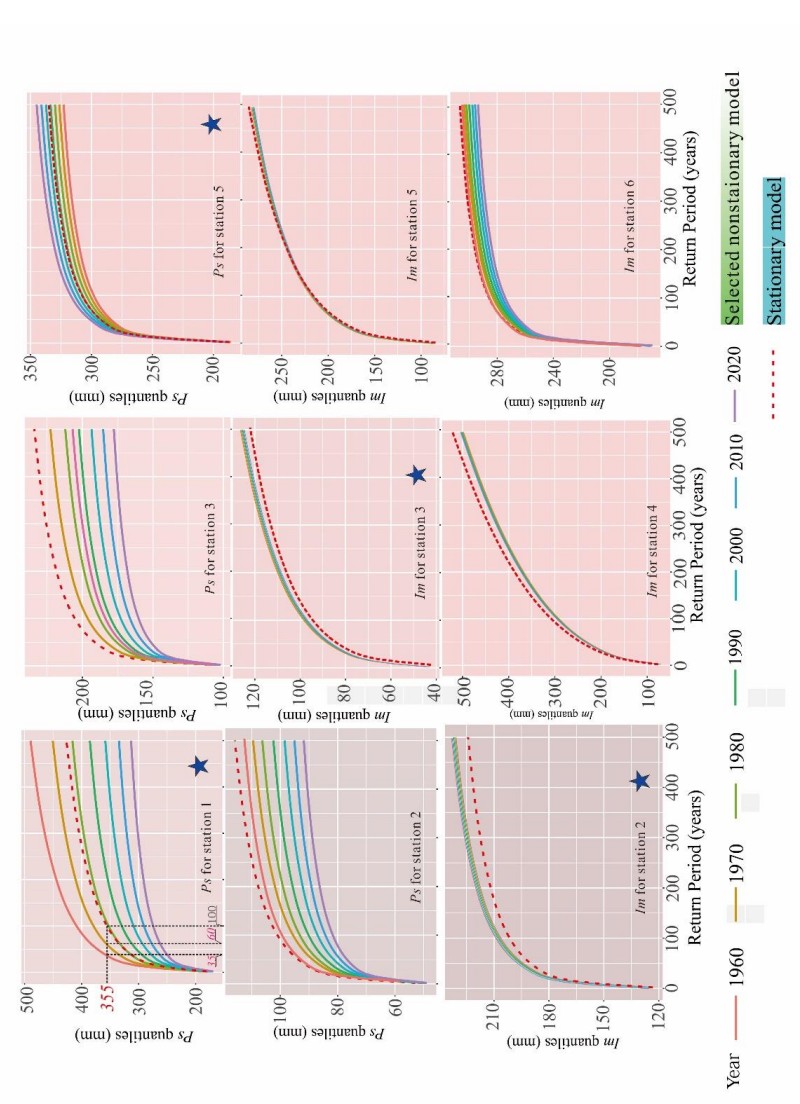

**Figure 4.** Estimated extreme rainfall attribute (*Ps* and *Im*) quantiles for stationary and selected nonstationary models at six stations.

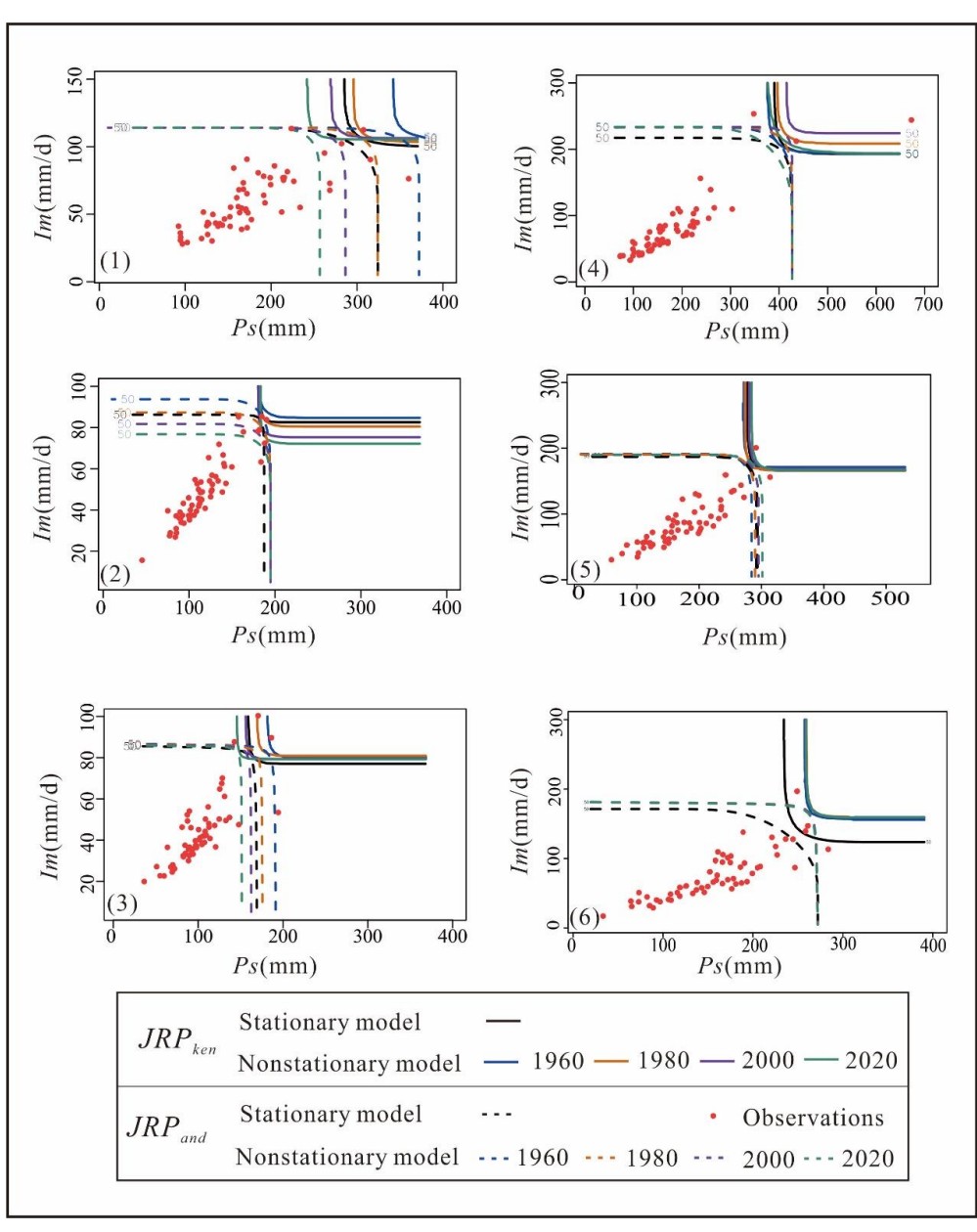

**Figure 5.** Isolines of Kendall return period and AND-based return period at the 50-year level for both stationary and nonstationary models. (1)-(6) represent the station number.



**Figure 6(a).** The most likely design event of Ps and Im with $JRP_{and} = 50$ for six stations



**Figure 6(b).** The most likely design event of Ps and Im with $JRP_{ken} = 50$ for six stations





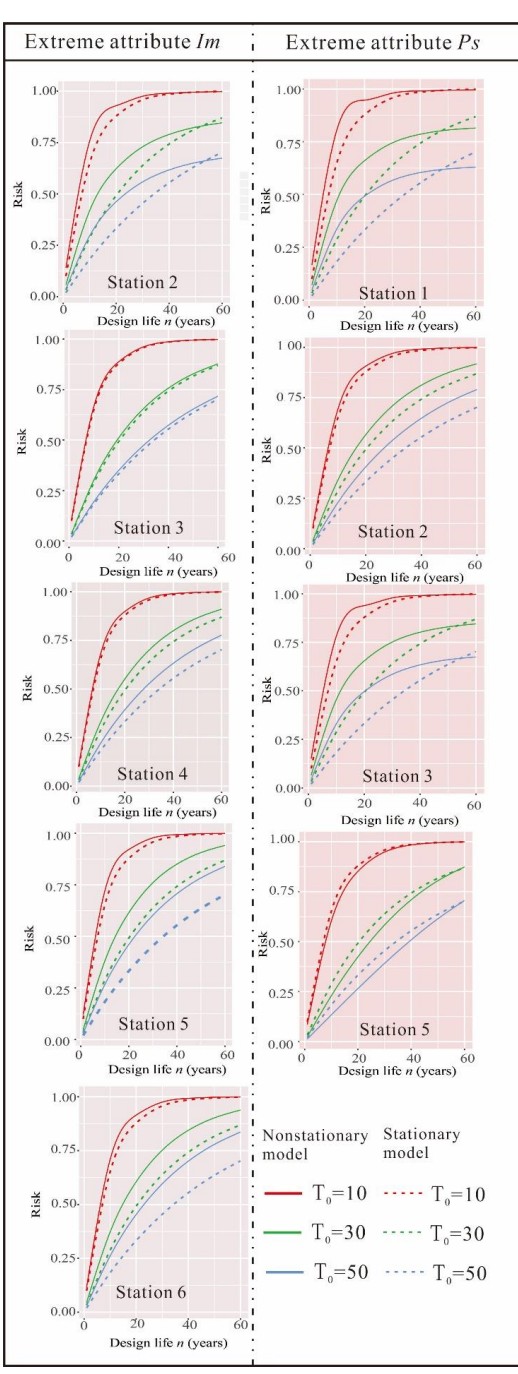

**Figure 7.** Nonstationary risk $R$ of the Haihe River design extreme rainfall quantile $x_{T_0}$ under th univariate case. The nonstationary design life level-based hydrological risk $R$ is regarded as a function of design life $n$ for $x_{T_0}$ with an initial return period $T_0$.





**Figure 8(a).** Nonstationary risk $R$ of the most likely design combinations of $Ps$ and $Im$ at $T_0$-year level based on $JRP_{s-and}$ and $JRP_{s-ken}$.





**Figure 8(b).** Nonstationary risk $R$ of the most likely design combinations of $Ps$ and $Im$ at $T_0$-year level based on $JRP_{s-and}$ and $JRP_{s-ken}$.