# Peer review of "Time-varying copula and design life level-based nonstationary risk analysis of extreme rainfall events"

_Hydrology and Earth System Sciences, 2019_

## Referee Comment (RC1) · Anonymous Referee #1 · 11 Nov 2019

The Authors present a methodology to estimate a methodology for a design life level-based risk analysis that takes into account uni- e bi-variate non-stationarity. The methodology presented is organized in 3 steps:

- The presence of trends and changing points for uni- e bi-variate datasets is tested using Mann-Kendall and Pettitt tests and their statistical significance assessed. - Independently to the stationarity test's results "According to Porporato and Ridolfi (1998), an insignificant trend should not be ignored because of its effect on the results of hydrological risk analysis. Hence, even if precipitation extremes at a certain station may recommend statistically weak trends, both the nonstationary and stationary models are established for each station in the following section", both stationary and nonstationary distribution functions are used to describe uni- e bi-variate data. - Design life level-

based risk analysis is applied to bi-variate data through Kendall's joint return period and AND's joint return period. By doing so the Authors the values to be assigned to the hydrological variables to design the infrastructures including bivariate nonstationarity.

The hydrological univariate variables considered are $P_s$ (annual cumulated precipitation above a threshold) and $I_m$ (annual maximum daily precipitation) extracted from daily precipitation time series of six meteorological stations located in the Haihe River basin (China).

For univariate variables the GEV distribution is assumed a priori and one stationary and two non-stationary (1: location parameter is time-varying; 2: both location and scale parameters are time-varying) parameters sets are evaluated for each variable. Once the best GEV model (i.e stationary or not) to describe each variable was identified the bi-variate analysis was performed.

For bivariate analysis four Copulas are considered: Student's t, Clayton, Gumbel and Frank with stationary and time-varying parameter. The best copula model is chosen according to AICc criterion results and the nonstationary "Kendall" and "AND" joint return periods computed.

The comments in the manuscript are coherent with the Figures; unfortunately, I was not able to reproduce Figure 4 using Table 3 values (see point 5 in General comments). For this reason, I would like to suggest the Authors to revise Table 3 and/or Figure 4 and the text related in the manuscript. Not being able to reproduce the results of Figure 4 I do not try to reproduce the other results in the manuscript I accept the Authors's analysis but I assume that it is based on data different from those presented in the manuscript.

The research presented is of potential interest for the readers of HESS but there some points need to be revised by the Authors. I suggest to accept the manuscript with major review.

[Figure]

############################################################################

General comments

There are some points that I would like the Authors to address in the manuscript

1) For the design of which structure are the variables Im and Ps significant? 2) In the Introduction climate change is indicated as one of the motivation to propose a nonstationary design life level-based risk analysis, but there is any attempt to project the results of the manuscript in the future. Shall the Authors provide some indication of what shall we expect in the future? Do the Authors compare their projection of hydrological extremes with the trends that can be derived from climate models?

3) Why do the Authors limits their analysis to six rain-gauges when there are several more in the area (e.g. https://doi.org/10.1002/hyp.9607)?

4) Which are the limits/problems relate to use an upper bounded distribution (i.e. GEV when $\kappa<0$) to describe variables that potentially range between [0 +∞).

5) There is a significant difference between quantiles reported in Figure 4 and those computed using equation 2 and parameters reported in bold in Tables 3(a) and 3(b) with t=1960, 1970, 1980, 1990, 2000, 2010, 2020. (see Fig1)

Here I show my attempt to reproduce the panels of Figure 4 using parameters as in the Table below vs screenshots of Figure 4 from the manuscript. Differences are quite evident, I see a similarity only for Im at station 4, for all the other cases it was impossible to reproduce the fan of non stationary GEVs at the different years. I would appreciate if the Authors could spend some of their time to investigate the reasons their and my outputs are so different. (see Fig2-Fig4)

6) In the presence of a statistically significant change point, is it correct to use the same parameter formulation to describe data "before" and "after" the abrupt change point? Moreover, which would be the results of trend analysis if the time series are split as "before" and "after" the change point. Do the Mann-Kendall test' results change

considering the "before" and "after" segment of the time series separately?

7) As last I would like to suggest the Author to add: a. A section on climate change projection and analysis that can be of interest for future infrastructure design b. one table reporting the basic statistics (min/max/mean/standard deviation) of the Ps and Im variables and the values of the 95-th percentile threshold to help understanding the variability of datasets; c. one figure showing the timeseries with the indication of the change point year of occurrence according to Pettitt test.

################################################################################

Specific comments

Line 116 The definition given of Ps variable recall me the index R95pTOT used in climate change studies (http://etccdi.pacificclimate.org/list_27_indices.shtml). Is it the same index? In addition, could the Authors specificy the period of observation they used to set the 95-th percentile threshold? According to R95pTOT index the reference period to set the 95-th percentile threshold is 1961-1990.

Lines 290-291 The Authors write "where $R\_i\hat{\ }ns$ and $R\_i\hat{\ }s$ are nonstationary risk and stationary risk of a certain hydraulic structure for a design life of i years", but 'i' goes from 1 to n. I would expect that 'n' indicates the design life and 'i' indicates the i-th year from now (i.e. the year the project "starts") to n-th year (end of the project's life).

Lines 367-368 The Authors write "Except for stations 4 and 5, the best distributions for the other stations were parallel for nonstationarity tests shown in Section 4.1". Is it possible that the mismatch between the nonstationarity test results and the best fitting distribution for Im (station 4 and 5) and Ps (station 5) was to the choice of the Author to ignore the test's results?

Lines 387-390 The Authors write "Contrary to station 5, the nonstationary St copula fitted better than did the stationary model for stations 1 and 6 which was not in accordance with the nonstationarity tests for these two stations (Table 2)." It is true that

according to bivariate MK test results station 1 and 6 should stationary, but at station 1 bivariate Pettitt test shows the presence of a change point; the presence of a change point could have influenced the results of LL and AICc ? What will happen if Im and Ps time series are "broken" before and after the change point to LL and AICc estimates?

Line 411 (and Conclusions) The Authors report a value of 355 mm for the 100-year Ps quantile in station 1 under stationary circumstances, but using the parameters reported in Table 3(a) the 100-year Ps quantile in station 1 under stationary circumstances is about 383 mm. It is probably a matter of approximation in the parameters values (355 mm corresponds to a report period of about 62 yr) but I will suggest the Authors to check these values.

###########################################################################

Minor corrections

Around the manuscript there are some typos like "Pettist" instead of "Pettitt"; missing spaces and so on (e.g Lines 226, 228), please check the text.

Lines 172 and 173 Is the limit "$(\mu\text{-}\sigma)/\kappa$" for lower (upper) boundary of x value correct? According to parameter's estimates in Tables 3(a)-3(b), when $\kappa<0$, x can assume only negative values, that is non coherent with the variables Ps and Im that are positively defined.

Lines 249-250 The Authors write "Let JRPs-and and JPRs-ken represent the three types of return period in the stationary case", but the return periods presented are only 2.

Line 260 "JPRs" probably was "JRPs"

Line 343 and Line 426 check the correct location of Figure 3.

Lines 359-365 The Authors write "The best fitted model was selected by performing the minimum DIC criterion combined with the Bayes factor (BF) test", but looking at bold

rows in tables 3(a) and 3(b) the criterion of minimum DIC seems not be respected for Im at station 2 where GEVns-1 is in bold instead of GEVns-2 (minimum DIC value).

Line 360-365 Comparing these lines with Table 3(a), for station 1, the variable described as GEVns-2 appears to be Ps and not Im. BF for Im variable in station 1 is >1. Please clarify this point.

Line 387 Please define "MK"

Lines 409-411 Figure 3 illustrates the results of nonstationary tests. Figure 4 reports the extreme rainfall quantiles. Please check the text.

##############################################################################

References

Line 62 Does "Assia et al., 2014" refer to "Aissia, M.A.B., Chebana, F., Ouarda, T.B.M.J., Roy, L., Bruneau, P., and Barbet, M.: Dependence evolution of hydrological characteristics, applied to floods in a climatechange context in Quebec, J. Hydrol., 519, 148–163, https://doi.org/10.1016/j.jhydrol.2014.06.042, 2014" ?

Line 98 Does "(Jakob, 2013)" refer to "Jakob, D., AghaKouchak, A. Easterling, D., Hsu, K., Schubert, S., and Sorooshian, S. (Eds.): Nonstationarity in extremes and engineering design, Springer, New York, 2013." ?

Line 100 "Read and Vogel (2015)" there is no correspondence in the references

Line 126 Does "Nelson (2007)" refer to "Nelsen, R.B.: An introduction to copulas, Springer, New York, 2007."?

Line 212 "Genest et al., 1995" there is no correspondence in the references

Line 213 "Hurvich and Tsai, 1989" there is no correspondence in the references

Line 235 "Fernandez and Salas, 1999" there is no correspondence in the references

"Ghanbari, M., M. Arabi, J. Obeysekera, and Sweet, W.: A coherent statistical model for

coastal flood frequency analysis under nonstationary sea level conditions, Earth's Future, 7, 162-177, https://doi.org/10.1029/2018EF001089, 2017." The publication year is 2019

"Zhang, Q. , Gu, X. , Singh, V. P. , and Chen, X.: Evaluation of ecological instream flow using multiple ecological indicators with consideration of hydrological alterations, J. Hydrol., 529, 711-722, https://doi.org/10.1016/j.jhydrol.2015.08.066, 2015." should be moved at the end of the reference list

################################################################################

Table (1) I suggest the Authors to change Longitude and Latitude with Longitude E and Latitude N, respectively coherently with the choice of indicating geographical coordinates in degree/minutes format.

Table (2) "Ps" and "Im" should be in italic. For station 3 and multivariate MK test the "*" should be close to the Z-statistic value not to the p-value. I suggest the Authors to add the indication of the year at which the change point is detected for both univariate and bivariate Pettitt test.

Table 3(a) e 3(b) please specify the meaning of bold row, I guess that bold indicates the "best" fitting model, but in this case why for Im variable at station 2 the best model is GEVns-1 if GEVns-2 shows the minimum DIC?

Table 3(b) refers to (Station 4-6) not to (Station 2-6) and the '-' symbol is missing for BF values of stationary GEV in station 5

Table 4(a) and 4(b) reports the meaning of bold and underlined text. Infinity symbol cited in caption does not appear in the table, probably substituted by "NaN".

################################################################################

Figure (1) Step S2 is omitted.

Figure (2) I would like to suggest the Authors to add the Haine river to the map.

Figure (3) In the caption there is a typo "Mann-Kendalld" instead of "Mann-Kendall". Please check the legend, the description of the last item (purple backward arrow) is equal to the one of the third one (green upward arrow). The "+" symbol is redundant with the test that already specify if the trend/change point is statistically significant.

Figure (4) the "star" symbol is not defined.

Figures (4), (6), (7) I would like to suggest the Authors to improve the quality of these figures. They seems to be a collection of screenshots with different size and background colour. Figure 4, in particular, seems to lack of organization in the sub-figures arrangement.

Please also note the supplement to this comment:
https://www.hydrol-earth-syst-sci-discuss.net/hess-2019-358/hess-2019-358-RC1-supplement.zip

———————————————

| Station | Attribute | Model | μ | σ | κ |
|---------|-----------|-------|---|---|---|
| 1 | Ps | GEVs | 144.37 | 45.32 | 0.058 |
| | | GEVns-2 | 151.85-0.00042t | exp(3.92-0.000038t) | 0.019 |
| | Im | GEVs | 50.64 | 18.13 | 0.012 |
| | | GEVs | 50.64 | 18.13 | 0.012 |
| 2 | Ps | GEVs | 106.25 | 28.84 | -0.12 |
| | | GEVns-2 | 99.92+0.0028t | exp(3.25+0.000057t) | -0.13 |
| | Im | GEVs | 40.71 | 14.01 | -0.053 |
| | | GEVns-1 | 42.02+0.00026t | 14.30 | -0.099 |
| 3 | Ps | GEVs | 86.92 | 26.92 | -0.077 |
| | | GEVns-2 | 85.78+0.0014t | exp(3.57-0.00014t) | -0.091 |
| | Im | GEVs | 34.31 | 11.00 | 0.13 |
| | | GEVns-1 | 37.51-0.00067t | 11.38 | 0.10 |
| 4 | Ps | GEVs | 141.69 | 55.87 | 0.19 |
| | | GEVs | 141.69 | 55.87 | 0.19 |
| | Im | GEVs | 56.36 | 22.29 | 0.34 |
| | | GEVns-1 | 65.73-0.0035t | 23.17 | 0.31 |
| 5 | Ps | GEVs | 160.34 | 56.07 | -0.21 |
| | | GEVns-2 | 150.25+0.0011t | exp(4.07-0.000038t) | -0.19 |
| | Im | GEVs | 67.46 | 26.37 | 0.072 |
| | | GEVns-1 | 77.72-0.0041t | 26.88 | 0.047 |
| 6 | Ps | GEVs | 137.56 | 57.44 | -0.24 |
| | | GEVs | 137.56 | 57.44 | -0.24 |
| | Im | GEVs | 63.02 | 29.43 | 0.033 |
| | | GEVns-2 | 57.05+0.00083t | exp(3.45-0.000071t) | 0.072 |

**Fig. 1.**

[Figure]

**Fig. 2.**

[Figure]

**Fig. 3.**

[Figure]

**Fig. 4.**

---

## Referee Comment (RC2) · Anonymous Referee #2 · 11 Nov 2019

**REVIEW REPORT**

Journal: HESS

**Paper:** HESS-2019-358

**Title:** Time-varying copula and design life level-based nonstationary risk analysis of extreme rainfall events

Author(s): Pengcheng Xu, Dong Wang, Vijay P. Singh, Yuankun Wang, Jichun Wu, Huayu Lu, Lachun Wang, Jiufu Liu, Jianyun Zhang

**GENERAL COMMENTS.**

In my opinion, essentially this paper only adds "noise" to the existing Literature: the techniques used have already been published in other works, the only novelty (clearly, not a methodological one) could be the case study, but any new case study must represent a newness over previous ones (otherwise it would be a replica). Most importantly, the work is in general statistically weak, and affected and flawed by fatal errors: the conclusions of the Authors may not be supported by the analyses they carried out. Apparently, the Authors (incorrectly) interpret the results according to their convenience, in order to prove what they want to prove, as shown below. In addition, referencing is often imprecise and/or improper and/or missing: always give credits to whom deserve credits. My recommendation is: REJECTION.

**SPECIFIC COMMENTS.**

**Line(s) 49–54.**

**Authors.** Copulas, a useful tool for modelling the structure of dependence between hydrological variables regardless of the types of marginal distributions, have been widely used for multivariate frequency analysis + references...

**Referee.** Historically, the paper by Salvadori and De Michele (2004) was the first one to deal with (copula) multivariate frequency analysis—later works are copies or small variants: this paper is not cited. Please, always give credits to whom deserve credits.

**Line(s) 75-ff.**

Authors. There are three kinds of joint return period methods...

**Referee.** NO. In Literature there are, at least, four kinds of joint return periods. The references given are incorrect. In Salvadori and De Michele (2004) the OR, AND and Kendall cases were first introduced. In Salvadori et al. (2013) a further survival-Kendall approach (not mentioned by the Authors) was outlined. Referencing is often imprecise, almost random: for instance, why citing Jiang et al. (2015) here? It has nothing to do with the original formalization of the four return periods mentioned above. Incidentally, the reference "Salvadori and Michele, 2010" is "Salvadori and De Michele, 2010" (it seems that the Authors wrote the references by hand, instead of using some suitable software...)

**Line(s) 95–97.**

**Authors.** Note that following the idea of Rootzén and Katz (2013) we regard the term hydrological risk as the possibility of a certain extreme event occurring and not as a quantification of expected losses.

**Referee.** Then, probabilistically and statistically speaking (and hydrologically as well!), you should better use the term "hazard" instead of "risk".

**Line(s) 126.**

Authors. Detailed information about copulas can be found in Nelson (2007).

**Referee.** NO. It is Nelsen (2006), not Nelson. For an engineering approach, you may also cite Salvadori et al. (2007). As a strong suggestion, the Authors should carefully check the correctness of all the references (it is easy to do it on the Internet), and add the missing ones.

**Line(s) 138-139.**

Authors. ...  $\theta_C^t$  is the dynamic copula parameter which is a linear function of time.

**Referee.** The Authors must justify this choice. Please do not reply that "the model was taken from this or that paper": it is not a scientific reason, for a model must be validated on the available data. Also, the results of suitable Goodness-of-Fit statistical tests must be shown.

**Line(s) 143-144.**

**Authors.** It is however possible that the nonstationary behavior may exist in both the marginal and joint distribution function.

**Referee.** Such an issue was already clearly pointed out and discussed in Salvadori et al. (2018), where a similar case study was investigated, and a thorough statistical analysis was carried out. The Authors must mention this fact, and follow the (proper statistical) guidelines outlined in that paper.

**Line(s) 158-Figure 1.**

The flow-chart shown in Figure 1 provides wrong indications (see also later comments). In fact, the Authors confuse GoF tests with selection criteria. The flow-chart must be rewritten.

**Line(s) 161-164.**

Authors. In this part, the Generalized Extreme Value (GEV) distribution was used to... (Cheng and AghaKouchak, 2014).

**Referee.** This reference makes little sense: the features of the GEV have already been stated and described since decades in other (seminal) works. Please use proper references.

**Line(s) 166-ff.**

Authors. The GEV distribution consists of three control parameters...

**Referee.** The GEV distribution is well known to hydrologists, there is no need to tell again a story that everybody knows.

**Line(s) 176–179.**

**Authors.** In this study, two kinds of nonstationary GEV models (GEVns-1 and GEVns-2) are developed with the shape parameter being constant. It should be emphasized that modelling the time variance in shape parameter needs long-term observations, which are often not available in practice (Cheng et al., 2014).

**Referee.** I recently rejected a paper very similar to the present one, where the GEV shape parameter was kept constant. The shape parameter is the most important one, for it rules the generation of extremes. The assumption adopted is definitely questionable: what (extreme) climate change could you really hope to model with a constant shape parameter? Practically, you are trying to model climate changes where the statistics of the extremes do not change with time: it makes little sense.

In addition (see also later comments), some estimates of the GEV shape parameter are positive and other negative (Table 3). This entails that, in some cases, the corresponding GEV law is upperbounded, i.e. unable to model an extreme behavior: this is a well known feature of the GEV. I agree that the GEV is the right distribution to be used in your analysis (Block Maxima), but the question is: how can you claim that the phenomenon you are modeling is an extreme one when upper-bounded GEV's are involved? The statistical results seem to tell another story...

**Line(s) 184–186, Eq.s (3)–(4).**

You must justify the assumptions/relations implicit in these equations. Why should the position and scale parameters change according to Eq.s (3)–(4)? Did you carry out any valuable/reliable fit? What are the p-Values? And, again, why should the shape parameter be constant instead? Incidentally, these are the same equations used in the paper I recently rejected...

**Line(s) 191-194.**

**Authors.** Simultaneously, the Deviance Information Criterion (DIC) and Bayes factors (BF) for different stationary and nonstationary models were calculated to select the best fitted marginal model. The minimum DIC value yielded the best performance, while BF smaller than 1 indicated the best fitting.

**Referee.** This is a typical fatal error of practitioners. These are only selection criteria, not Goodnessof-Fit tests. You must first use (non-stationary) GoF tests to check whether a model is admissible! Otherwise, without first checking the models via suitable GoF tests, you may end up choosing nonadmissible ones. This work has no statistical bases.

**Line(s) 191-194.**

**Authors.** In multivariate hydrological frequency analysis, two kinds of copulas, named elliptical and Archimedean copulas are widely used in hydrological applications.

**Referee.** So what? The fact that these copulas were used in other works is not, and cannot be, a scientific justification. This is the usual approach of practitioners that use the copulas provided by Matlab. Given my experience, I do not really think that Nature (especially considering the generation of Extremes) gets stick to just these dependence structures—see also later comments. And, worst of all, you did not even check these copula models via suitable multivariate GoF tests (which are available in Literature, and some certified software is even for free—see below): this work has no statistical bases.

**Line(s) 203-205.**

**Authors.** The Gaussian copula was not used in this study because of its deficiency in describing dependencies of extremes (Renard and Lang, 2007).

**Referee.** The Authors are clearly considering the concept of Tail Dependence. Well, also the Frank family has no tail dependence, while the Clayton family only has lower tail dependence (possibly, of no interest here), the Gumbel family only has upper tail dependence, and the Student family has both lower and upper tail dependence (but they must be equal, and, most of all, they both must exist at the same time!). There are more suitable families of copulas for modeling extremes: again, the ones used by the Authors are simply those provided by Matlab, as (unfortunately, too) many practitioners do, preventing a reliable/valuable investigation and modeling of the phenomenon of interest.

**Line(s) 208, Eq. (5).**

Again, as above, you must justify the assumptions/relations shown in this equation. Why should the

copula parameter change according to Eq. (5)? Did you carry out any investigation? What are the p-Values?

**Line(s) 212-214.**

Authors. The Corrected Akaike Information Criterion (AICc; Hurvich and Tsai, 1989) was employed to make a goodness-of-fit...

**Referee.** NO. This a typical fatal error of practitioners. The AIC (corrected or not) is only a selection criterion, not a GoF procedure. You must first show that a copula is statistically admissible, e.g. via suitable Monte Carlo Cramer-von Mises or Kolmogorov-Smirnov tests, as in the R package "copula". Then, and only then, you may compare (only) the admissible copulas (if any) and select the "best" one according to some suitable criterion (e.g., the AICc, the BIC, the NLL, etc...).

**Line(s) 214-216.**

**Authors.** Obviously, the presence of nonstationarity in the copula parameter was determined by comparison of the AICc value.

**Referee.** This sentence is obscure. Are you saying that, since the non-stationary model performs better, then the phenomenon is non-stationary? If so, this makes no statistical and philosophical sense. It looks like you are using your models to "decide" how the real world should work: this is contrary to every scientific principle. This work is also bugged from an epistemological perspective.

**Line(s) 217-ff., Sec. 2.3.**

Authors. "2.3. Joint return period and risk analysis based on KEN's and AND's methods"

**Referee.** Multivariate failure probabilities have been well mathematically formalized in Salvadori et al. (2016), by originally defining and exploiting suitable Hazard Scenarios and copulas' relations. The Authors must take this work into serious account, and mention it.

**Line(s) 241, Eq. (7).**

See the more general approach and discussion in (Salvadori et al., 2016, Eq.s (33)-(35)).

**Line(s) 262, Eq. (11).**

Why in Eq. (11) the parameters of the marginals  $F_X$ ,  $F_Y$ , used as arguments in the copula C, do not vary with time?

**Line(s) 268-269.**

Authors. The most likely event at the  $T_0$ -year level can be calculated as (Graler et al., 2013)...

**Referee.** NO. The Most Likely technique was first introduced in Salvadori et al. (2011): always give credits to whom deserve credits. In addition, it is not the only possible one, as shown in the same paper (viz., the Component-wise Excess method). Moreover, further approaches are outlined in Corbella and Stretch (2012) and Salvadori et al. (2014). Why was the Most Likely approach chosen in this work?

**Line(s) 289, Eq. (17).**

In Eq. (17), why is the modulus used? Obviously  $\Delta R$  will always be positive. And even in this latter case, there is no quantification of any "scale" on which  $\Delta R$  should be evaluated (when is it large? when is it small?). Such a number tells nothing to me.

**Line(s) 330-331.**

**Authors.** As shown in Figure 3, concurrences of univariate and bivariate trends, the nonstationarities in rainfall extremes can be detected at several stations...

**Referee.** This is simply because you use a 10% critical  $\alpha$ -level, entailing a large probability of rejecting the Null Hypothesis of non-stationarity. For instance, at a standard 5% level, no one of the Univariate and Multivariate MK tests would fail, only two (at most three) out of 12 of the Univariate Pettitt tests would fail, and only one out of 6 of the Multivariate Pettitt tests would fail. In turn, the conclusions of the Authors are definitely questionable: in my opinion, in general, there is no clear statistical evidence of non-stationarity (not to say if the standard 1% level were used, for in this case stationarity would be fully supported). Apparently, the Authors manipulate statistics according to their convenience, in order to show what they want to show.

**Line(s) 353-354.**

Authors. The location parameter ( $\mu$ ) and scale parameter ( $\sigma$ ) are regarded as time variant, while the shape parameter  $\kappa$  is time invariant...

**Referee.** As above, it is a dream to try and model time-variation of extremes using a constant shape parameter: it is the only one that matters in these kind of analyses. In addition, why should the other parameters vary according to Eq.s (3)–(5)? Simply because the same relations were used in other papers (again, without justification)? This paper has no scientific objective grounds.

**Line(s) 356-358.**

Authors. Despite the exception of Im for station 4, the shape parameter  $\kappa$  for most fitted models was in the interval of [-0.3,0.3]...

**Referee.** Tables 3 provide little statistical information, for no suitable confidence intervals are shown: this may have considerable consequences regarding the conclusions drawn by the Authors in later sections. In fact, they did not carry out any Monte Carlo analysis, and hence their results do not take into account the estimates' uncertainties (as if the Authors were stating the absolute Truth). To be clear, no confidence bands are plotted in later figures. This is not a scientific way of proceeding: the Authors must provide plots such as the ones shown in Salvadori et al. (2018), which may give an idea of the uncertainties at play (which may be huge, especially when a GEV is used, and may completely change the interpretation of the results, as I suspect).

In addition, as above, some of the fitted values of the shape parameter would imply that the corresponding GEV is Upper Bounded, entailing that the corresponding variable cannot be an Extreme one. Furthermore, the fact that the range of the shape parameter is "in accordance with previous studies" is not significant and relevant at all (also given the fact that the range is quite large).

**Line(s) 359-360.**

**Authors.** The best fitted model was selected by performing the minimum DIC criterion combined with the Bayes factor (BF) test.

Referee. Again, you did not show that it is an admissible one! This work has no statistical bases.

**Line(s) 380-382.**

**Authors.** Table 4(a)-(b) illustrates the results of best fitted copula, based on the minimum AICc and maximum logllikelihood value (LL).

**Referee.** Again, AIC and LL are not GoF criteria: the chosen models can be non-admissible! This work has no statistical bases.

**Line(s) 433-435.**

**Authors.** Although the copula model for station 5 was stationary, it was regarded as a nonstationary model because of the marginal nonstationary GEVns-2 model for Ps or Im, which existed at other stations.

**Referee.** This makes no sense. The Authors do not understand the basic fact that the dependence structure is independent of the marginals (as stated by Sklar's representation Theorem): even if the marginals are non-stationary, the copula may be stationary. The introduction of non-stationary copulas is arbitrary, without any justification: you cannot manipulate the results in this way!

**Line(s) 440-ff.**

Authors. Figure 5 shows isolines of Kendall return period and AND-based return period...

**Referee.** Given the uncertainties mentioned above (not considered by the Authors), I strongly suspect that the interpretation of the results shown in Figure 5 could be quite different if suitable confidence bands were plotted. This work lacks of elementary statistical bases.

**Line(s) 537-ff., Sec. 4.6.**

In the light of the objections given above, the "Further discussion" section (4.6) makes no sense.

**References**

- Corbella, S., Stretch, D. D., 2012. Multivariate return periods of sea storms for coastal erosion risk assessment. Nat. Hazards Earth Syst. Sci. 12, 2699–2708.
- Salvadori, G., De Michele, C., 2004. Frequency analysis via copulas: theoretical aspects and applications to hydrological events. Water Resour. Res. 40, W12511, doi: 10.1029/2004WR003133.
- Salvadori, G., De Michele, C., Durante, F., 2011. On the return period and design in a multivariate framework. Hydrol. Earth Syst. Sci. 15, 3293—3305.
- Salvadori, G., De Michele, C., Kottegoda, N., Rosso, R., 2007. Extremes in Nature. An approach using Copulas. Vol. 56 of Water Science and Technology Library Series. Springer, Dordrecht, ISBN: 978-1-4020-4415-1.
- Salvadori, G., Durante, F., De Michele, C., 2013. Multivariate return period calculation via survival functions. Water Resour. Res. 49, 2308–2311, doi: 10.1002/wrcr.20204.
- Salvadori, G., Durante, F., De Michele, C., Bernardi, M., Petrella, L., 2016. A multivariate Copula-based framework for dealing with Hazard Scenarios and Failure Probabilities. Water Resources Research 52 (5), 3701–3721, doi: 10.1002/2015WR017225.
- Salvadori, G., Durante, F., Michele, C. D., Bernardi, M., 2018. Hazard assessment under multivariate distributional change-points: Guidelines and a flood case study. Water 10 (6), 751–765.
- Salvadori, G., Tomasicchio, G. R., D'Alessandro, F., 2014. Practical guidelines for multivariate analysis and design in coastal and off-shore engineering. Coastal Engineering 88, 1–14, doi: 10.1016/j.coastaleng.2014.01.011.

---

## Author Comment (AC1) · 20 Dec 2019

General comments 1) For the design of which structure are the variables Im and Ps significant? Response: Copulas have been widely used as for multivariate frequency analysis of extreme rainfall events (Zhang and Singh, 2007; Kao and Govindaraju, 2008; Rauf and Zeephongsekul, 2014; Vandenberghe et al., 2010). In this study, the proposed nonstationary model can not only used to make hazard assessment for extreme rainfall events but also for flood and drought events. We focus more on the proposed method and show how to implement it in the multivariate hazard assessment. Ps denotes annual extreme rainfall volume while Im (annual maximum daily precipitation) is just the annual rainfall intensity. In my opinion, variables Im and Ps would be significant for the urban flood control and drainage facilities. Not like the flood

event, the volume and peak attributes has direct relation with the dam spillway. Here, variables Im and Ps would be indirectly related to the urban drainage facilities. If the Ps and Im intensify, the pressure of the drainage facilities would be larger.

2) In the Introduction climate change is indicated as one of the motivation to propose a nonstationary design life level-based risk analysis, but there is any attempt to project the results of the manuscript in the future. Shall the Authors provide some indication of what shall we expect in the future? Do the Authors compare their projection of hydrological extremes with the trends that can be derived from climate models? Response: In the revised manuscript, the average annual reliability (AAR) method to quantify the probability that a hydraulic system would be safe over its planning period in univariate case (Salas and Obeysekera, 2014; Read and Vogel, 2015) is adopted because of its agreement with nonstationarity. We have accepted this good suggestions of predicting the future extremes in the future. Since the nonstationary models with time as covariates, it can be used to predict the extreme value in the future 5 years (2018-2022). Considering new dataset being added to the original data, the parameters of the time-varying or stationary models would change. We limited the years of future to 5 years. In the same way, the ending year of the design life period is set as 2022 (Figure 5 in manscript). In this study, the time-varying marginal and copula models are established on the observed dataset (1958-2017). It would show different trend analysis results with the synthetized time series from the seven GCM simulations (BCC-CSM 1.1, CanESM2, CNRM-CM5, CSIRO-Mk3.6.0, IPSL-CM5A-LR, IPSL-CM5A-MR, NorESM1-M) involved in CMIP5 under RCP4.5 scenarios as shown in Table R1. Based on the analysis of manuscript, the extreme values or their dependence structure showed a significant trend. So we compared the series from these 4 stations. The difference between the observed extreme series and the simulated series are obvious. And if we make the trend analysis based on the predicted data set from the GCM models for the period of 1961-2100, the trend is significant (Z for Ps of station 1 is -5.82 while Im of station 1 is -4.32) at 5% level. And the results of Mann-Kendall trend analysis is bound to berelected by the length of extreme series. And the above

of trend comparison is beyond our scope of this paper, we focus more on the guide-lines of how to establish the nonstationary models and the importance of considering the nonstationarity in hydrologic design by quantile estimation based a certain AAR level. And in the future study of mine, we should focus more on the trend analysis and nonstationary analysis based on the synthetized time series from climate model.

(3)Why do the Authors limits their analysis to six rain-gauges when there are several more in the area (e.g. https://doi.org/10.1002/hyp.9607)? Response: Very thanks to this suggestion. We have also taken the suggestions of the comment suggested by Reviewer 2. Also, in the following comment, the selected extreme series did not show a significant trend at 5% level. So we considered more stations in the Haihe River Basin. In addition, the original 95-th percentile threshold for Ps has changed to 0.90. Because of the limitation data length and missing data, the daily precipitation with period 1958-2017 was available at 16 stations. Fortunately, the absolute value of Z statistics of the Ps of station 2 and dependence structure of station 6 is larger than 1.96 which the threshold value of 5% significance level. And also, the change-point (CP) tests recommend change point existed in the Ps and dependence structure of Ps and Im from station 7. Based on the trend and change-point analysis, the single extremes or their dependence structure of station 2, 6, 7 showed a monotonic trend or change point. And based on LR statistic tests which are used to test whether linear or nonlinear trend existed in parameter, only 1,5, 6 and 8 showed trend in parameters. For the space limitations, we only show eight stations which contains all the nonstationary series in these stations. And this study mainly focus on how to establish the nonstationary copula models to make the hazard assessment from the hydrologic design. Based on the LR and nonparametric tests (Mann-Kendall and change point), there are 11 stations out of these 16 stations did not have stationarity.

4) Which are the limits/problems relate to use an upper bounded distribution (i.e. GEV when $\kappa<0$) to describe variables that potentially range between [0 +∞). Response: This is a formula expression error. In order to make the fit of marginal distribution fit

more objectively, we take five kinds of extreme value distribution into consideration: two kinds of 3-parameter distribution (Generalized Extreme Value, GEV; Pearson type III, PIII) and three kinds of 2-parameter distributions (Gamma, Weibull and Lognormal).

5). There is a significant difference between quantiles reported in Figure 4 and those computed using equation 2 and parameters reported in bold in Tables 3(a) and 3(b) with t=1960, 1970, 1980, 1990, 2000, 2010, 2020. Response: As stated in response to comment 4, the selected extreme series did not show a significant trend at 5% level. On the one hand, we considered more stations in the Haihe River Basin. On the other hand, the original 95-th percentile threshold for Ps has changed to 0.90. And potential marginal models are shown in supplementary information Table S1 (a)-(d). Finally, based on the LR tests, GOF tests and AICc criterion. For marginal distribution, the extreme attributes (Ps or Im) extracted from station 1, 5, 6, 7 and 8 showed nonstationarity. The nonstationarity for station 1, 5, 6 and 8 was because of existed trend in parameter while the existed change point (year 1979) was the reason why incorporate the nonstationarity in station 7.

6) In the presence of a statistically significant change point, is it correct to use the same parameter formulation to describe data "before" and "after" the abrupt change point? Moreover, which would be the results of trend analysis if the time series are split as before" and "after" the change point. Do the Mann-Kendall test' results change considering the before" and "after" segment of the time series separately? Response: Firstly, thanks a lot for this suggestion by reviewer 1. Your idea is just in accordance with the paper by Salvadori et al. (2018), which is just the correct and objective procedure when the change point did exist in the extreme series. Following this suggestions, we made univariate and multivariate change point analysis for these 8 stations with results in Table 3 (in revised manuscript). The Ps and dependence structure of Ps and Im of station 7 showed change point at year 1979. Also the Mann-Kendall (MK) trend analysis was also implemented for these two spilt series. For station 7, Before change point (1979), Z=0.26 for univariate MK test for Ps, Z=0.99 for Multivariate MK tests;

After change point (1979) Z=0.45 for univariate MK for Ps and Z=0.62 for Multivariate MK tests. Compared to the MK tests for the whole data, the statistics (Z) changed a lot from 1.54 to 0.26 and 0.45 for Ps. The same situation existed in Multivariate MK tests. For the sake of showing more visually, the change point at year 1979 is plotted in Figure 3 (in revised manuscript). In the following analysis of ARR quantiles, the quantile of each AAR quantile was calculated for these two spilt series no matter in univariate or bivariate case.

7) As last I would like to suggest the Author to add: a. A section on climate change projection and analysis that can be of interest for future infrastructure design b. one table reporting the basic statistics (min/max/mean/standard deviation) of the Ps and Im variables and the values of the 95-th percentile threshold to help understanding the variability of datasets; c. one figure showing the time series with the indication of the change point year of occurrence according to Pettitt test. Response: for suggestion (a), we have add this suggestion in section 3.3 as follows: "Since the nonstationary models with time as covariates, it can be used to predict the extreme value in the future. Considering new dataset being added to the original data, the parameters of the time-varying models would change. In this study, we assume the parameters of the selected time-varying models did not change a lot in the future 5 years. So the estimated time varying models from the original extreme series were used to predict year 2018-2022 (t=61-65). Based on the same assumption, the ending year of design life period in the following AAR-based quantiles calculation is set as 2022. As shown in Figure 5 (in revised manuscript), mean value of Ps from station 1 and station 6 exhibited a downward trend while mean value of Im from station 5 and Ps from station 8 exhibited a NMT trend. For the predicted period, the predicted nonstationary marginal distributions for the extremes extracted from these 4 stations presented smaller mean values than those of the stationary distributions. Furthermore, the divergence of mean values between them are becoming larger as time goes on for station 1, 5 and 6. But for Ps from station 8, the divergence of mean values between them are becoming smaller as time goes on." And in section 3.5-"Nonstationary hazard assessment based on AAR

metrics for univariate and bivariate cases"ïïjŇthe design life period is set as 1983-2022. The future 5 years from 2018 to 2022. Considering new dataset being added to the original data, the parameters of the time-varying or stationary models would change. We limited the years of future to 5 years for cautiousness. These two adding parts are of great interest for the future hydrologic design. For suggestion (b), we add Table 2 (in revised manuscript). to show the basic statistics (min/max/mean/standard deviation) of the Ps and Im variables and the values of the 90-th percentile threshold to help understanding the variability of datasets.

For suggestion (c), we plot the change point for the Ps extreme series from station 7 (Figure 3 in revised manuscript).

Specific comments

Line 116 The definition given of Ps variable recall me the index R95pTOT used in climate change studies (http://etccdi.pacificclimate.org/list_27_indices.shtml). Is it the same index? In addition, could the Authors specificy the period of observation they used to set the 95-th percentile threshold? According to R95pTOT index the reference period to set the 95-th percentile threshold is 1961-1990. Response: In this study, the Ps is different from the R95pTOT index. It is just annual total precipitation of the daily precipitation more than the 90th percentile threshold for each year. That means, each year has a unique threshold value. We have made its definition more readable and clear.

Line 290 -291 The Authors write where $R\_i\hat{\ }ns$ and $R\_i\hat{\ }s$ are nonstationary risk and stationary risk of a certain hydraulic structure for a design life of i years", but 'i' goes from 1 to n. I would expect that 'n' indicates the design life and 'i' indicates the i-th year from now (i.e. the year the project "starts") to n-th year (end of the project's life). Response: This risk quantified metric is deleted from the manuscript. We accept the average annual reliability (AAR) method to quantify the probability that a hydraulic system would be safe over its planning period in univariate or bivariate case (Salas and

Obeysekera, 2014; Read and Vogel, 2015).

Lines 367-368 The Authors write "Except for stations 4 and 5, the best distributions for the other stations were parallel for nonstationarity tests shown in Section 4.1". Is it possible that the mismatch between the nonstationarity test results and the best fitting distribution for Im (station 4 and 5) and Ps (station 5) was to the choice of the Author to ignore the test's results? Response: as stated by the reviewer, it is of great possibility the mismatch between the nonstationarity test results and the best fitting distribution for Im (station 4 and 5) and Ps (station 5) was to the choice of the Author to ignore the test's results. And we use 10% significance level which entailing a large probability of rejecting Null Hypothesis of non-stationarity. So in revised forms, we take a careful statistical tests with 5% significance level for Mann-Kendall (MK), change point tests. We also proposed the Log Likelihood ratio (LR) tests (Coles (2001)), which is more rigorous trend detection methods than nonparametric methods (MK). For most cases, the results of trend analysis from LR tests are consistent with that by MK tests. However, as shown in Table 3(b), the trend in the parameter existed in Im of station 5, Ps of station 6 and 8 based on the LR tests at the 5% significance level which recommends another situation different from the previous MK tests. Through analysis, it would be caused by the he opposing trend in the location and scale parameters.

Lines 387-390 The Authors write "Contrary to station 5, the nonstationary St copula fitted better than did the stationary model for stations 1 and 6 which was not in accordance with the nonstationarity tests for these two stations (Table 2)." It is true that according to bivariate MK test results station 1 and 6 should stationary, but at station 1 bivariate Pettitt test shows the presence of a change point; the presence of a change point could have influenced the results of LL and AICc ? What will happen if Im and Ps time series are "broken" before and after the change point to LL and AICc estimates? Response: we have taken this suggestion into considerations. The extreme series from station 7 in the revised manuscript showed change point. The Ps and dependence structure of Ps and Im of station 7 showed change point at year 1979. Also the

Mann-Kendall (MK) trend analysis was also implemented for these two spilt series. For subseries, no trend can be detected based on Mann-Kendall and LR tests. The best marginal distribution and copula is shown in Table 4(a) and Table 5(a).

Line 411 (and Conclusions) The Authors report a value of 355 mm for the 100-year Ps quantile in station 1 under stationary circumstances, but using the parameters reported in Table 3(a) the 100-year Ps quantile in station 1 under stationary circumstances is about 383 mm. It is probably a matter of approximation in the parameters values (355 mm corresponds to a report period of about 62 yr) but I will suggest the Authors to check these values. Response: Since average annual reliability (AAR) method to quantify the probability of the hydraulic structure is of great potentiality in communicating hazard of failure under both stationary and nonstationary conditions (Read and Vogel, 2015), we adopt ARR method to estimate the quantiles for a design life period including future 5 years.

Minor corrections Around the manuscript there are some typos like "Pettist" instead of "Pettitt"; missing spaces and so on (e.g Lines 226, 228), please check the text. Response: Following the study of Salvadori et al. (2018), the non-parametric change-point statistical tests were implemented to check that whether the marginal or joint distributions are sensitive to changes. These tests can be manipulated in the R package npcp (Kojadinovic, 2017). We replaced Pettitt tests with the above statistical change point tests as suggested by Review #2.

Lines 172 and 173 Is the limit "$(\mu\text{-}\sigma)/\kappa$" for lower (upper) boundary of x value correct? According to parameter's estimates in Tables 3(a)-3(b), when $\kappa<0$, x can assume only negative values, that is non coherent with the variables Ps and Im that are positively defined. Response: It is an error of formula definition.

Lines 249-250 The Authors write "Let ãĂŰJRPãĂŮ_(s-and) and ãĂŰJRPãĂŮ_(s-ken) represent the three types of return period in the stationary case", but the return periods presented are only 2. Response: it is an error of station. It should be stated as follows:

[Figure]

Let we calculate the joint return period from the AND and Kendall Scenario.

Line 260 "JPRs" probably was "JRPs" Response: we have made deep self-checking of the notations.

Line 343 and Line 426 check the correct location of Figure 3 Response: we have checked this figure's location.

Lines 359-365 The Authors write "The best fitted model was selected by performing the minimum DIC criterion combined with the Bayes factor (BF) test", but looking at bold rows in tables 3(a) and 3(b) the criterion of minimum DIC seems not be respected for Im at station 2 where EVns-1 is in bold instead of GEVns-2 (minimum DIC value). Response: The reviewers' suggestions and views are right. The selected extreme series from the manuscript did not show a significant trend at 5% level. We have checked the process of extreme value extraction. We have changed the original 95-th percentile threshold for Ps has changed to 0.90-th percentile for Ps. And we take data from more stations into consideration. And based on the analyzed results, the nonparametric tests were consistent with the LR tests in most cases.

Line 360-365 Comparing these lines with Table 3(a), for station 1, the variable described as GEVns-2 appears to be Ps and not Im. BF for Im variable in station 1 is >1. Please clarify this point. Response: in the revised manuscript, we use the LR tests to select the nonstationary models with trend in parameter. Detailed information can be found in lines 210-227 of revised manuscript.

Line 387 Please define "MK" Response: we have taken this suggestion to show the full definition of the Abbreviation.

Lines 409-411 Figure 3 illustrates the results of nonstationary tests. Figure 4 reports the extreme rainfall quantiles. Please check the text. Response: We have checked the text.

References Line 62 Does "Assia et al., 2014" refer to "Aissia, M.A.B., Chebana, F.,
Ouarda, T.B.M.J., Roy, L., Bruneau, P., and Barbet, M.: Dependence evolution of hydrological characteristics, applied to floods in a climatechange context in Quebec, J. Hydrol., 519, 148–163, https://doi.org/10.1016/j.jhydrol.2014.06.042, 2014" ? Response: We have made revision for the reference quotation.

Line 98 Does "(Jakob, 2013)" refer to "Jakob, D., AghaKouchak, A. Easterling, D., Hsu, K., Schubert, S., and Sorooshian, S. (Eds.): Nonstationarity in extremes and engineering design, Springer, New York, 2013" ? Response: We have made revision for the reference quotation.

Line 100 "Read and Vogel (2015)" there is no correspondence in the references Response: we have added the "Read, L.K., and Vogel, R.M.: Reliability, return periods, and risk under nonstationarity, Water Resour. Res., 51, 6381-6398, https://doi.org/10.1002/2015WR017089, 2015." to reference list.

Line 126 Does "Nelson (2007)" refer to "Nelsen, R.B.: An introduction to copulas, Springer, New York, 2007."? Response: We have made revision for the reference quotation.

Line 212 "Genest et al., 1995" there is no correspondence in the references Response: In revised manuscript, this reference is deleted

Line 213 "Hurvich and Tsai, 1989" there is no correspondence in the references Response: we have added the "Hurvich, C. M. and Tsai, C. L.: Regression and time series model selection in small samples, Biometrika, 76, 297–307, https://doi.org/10.2307/1271469, 1989." to the reference list

Line 235 "Fernandez and Salas, 1999" there is no correspondence in the references. Response: we have used AAR metrics which did not contain the reference.

Ghanbari, M., M. Arabi, J. Obeysekera, and Sweet, W.: A coherent statistical model for coastal flood frequency analysis under nonstationary sea level conditions, Earth's Future, 7, 162-177, https://doi.org/10.1029/2018EF001089, 2017." The publication year

is 2019. Response: we have changed the publication year.

"Zhang, Q. , Gu, X. , Singh, V. P. , and Chen, X.: Evaluation of ecological instream flow using multiple ecological indicators with consideration of hydrological alterations, J. Hydrol., 529, 711-722, https://doi.org/10.1016/j.jhydrol.2015.08.066, 2015 . " should be moved at the end of the reference list. Response: we have deleted the reference which is not included in the manuscript.

Table (1) I suggest the Authors to change Longitude and Latitude with Longitude E and Latitude N, respectively coherently with the choice of indicating geographical coordinates in degree/minutes format. Response: we have accepted the suggestion and made revision to Table 1.

Table (2) "Ps" and "Im" should be in italic. For station 3 and multivariate MK test the "*" should be close to the Z-statistic value not to the p-value. I suggest the Authors to add the indication of the year at which the change point is detected for both univariate and bivariate Pettitt test. Response: we have accepted the suggestion and showed the change point if it exists by change point tests. "Ps" and "Im" have be made in italic form.

Table 3(a) e 3(b) please specify the meaning of bold row, I guess that bold indicates the "best" fitting model, but in this case why for Im variable at station 2 the best model is GEVns-1 if GEVns-2 shows the minimum DIC? Response: we have taken this suggestion in the revised manuscript.

Table 3(b) refers to (Station 4-6) not to (Station 2-6) and the '-' symbol is missing for BF values of stationary GEV in station 5. Response: we have taken this suggestion in the revised manuscript.

Table 4(a) and 4(b) reports the meaning of bold and underlined text. Infinity symbol cited in caption does not appear in the table, probably substituted by "NaN". Response: we have taken this suggestion in the revised manuscript and improved the definition of

table symbols.

##############################################################################

Figure (1) Step S2 is omitted. Response: we have taken this suggestion in the revised manuscript and improved the logical order of the flowchart.

Figure (2) I would like to suggest the Authors to add the Haihe river to the map. Response: we have add the main stream of Haihe River to the map.

Figure (3) In the caption there is a typo "Mann-Kendalld" instead of "Mann-Kendall". Please check the legend, the description of the last item (purple backward arrow) is equal to the one of the third one (green upward arrow). The "+" symbol is redundant with the test that already specify if the trend/change point is statistically significant. Response: we have taken this suggestion in the revised manuscript

Figure (4) the "star" symbol is not defined. Response: we have checked clearly in revised manuscript.

Figures (4), (6), (7) I would like to suggest the Authors to improve the quality of these figures.They seems to be a collection of screenshots with different size and background colour. Figure 4, in particular, seems to lack of organization in the sub-figures arrangement. Response: we have taken this suggestion to improve the quality of the figures.

Please also note the supplement to this comment:
https://www.hydrol-earth-syst-sci-discuss.net/hess-2019-358/hess-2019-358-AC1-supplement.pdf

**Supplement:**

**Response to reviews**

**Reviewer #1:**

The Authors present a methodology to estimate a methodology for a design life level-based risk analysis that takes into account univariate and bi-variate non-stationarity. The methodology presented is organized in 3 steps. The research presented is of potential interest for the readers of HESS but there some points need to be revised by the Authors. I suggest to accept the manuscript with major review.

**Response to Reviewer 1:**

Great appreciation for this comment!

In order to make the description of the methodology more readable and easier to comprehend, considerable revision has been made to the original manuscript. We have taken the review's suggestions into consideration. A detailed point-by-point reply has been made as follows:

General comments

There are some points that I would like the Authors to address in the manuscript:

1)  For the design of which structure are the variables *Im* and *Ps* significant?

**Response:** Copulas have been widely used as for multivariate frequency analysis of extreme rainfall events (Zhang and Singh, 2007; Kao and Govindaraju, 2008; Rauf and Zeephongsekul, 2014; Vandenberghe et al., 2010). In this study, the proposed nonstationary model can not only used to make hazard assessment for extreme rainfall events but also for flood and drought events. We focus more on the proposed method and show how to implement it in the multivariate hazard assessment. *Ps* denotes annual extreme rainfall volume while *Im* (annual maximum daily precipitation) is just the annual rainfall intensity. In my opinion, variables *Im* and *Ps* would be significant for the urban flood control and drainage facilities. Not like the flood event, the volume and peak attributes has direct relation with the dam spillway. Here, variables *Im* and *Ps* would be indirectly related to the urban drainage facilities. If the *Ps* and *Im* intensify,

```
```

the pressure of the drainage facilities would be larger.

2) In the Introduction climate change is indicated as one of the motivation to propose a nonstationary design life level-based risk analysis, but there is any attempt to project the results of the manuscript in the future. Shall the Authors provide some indication of what shall we expect in the future? Do the Authors compare their projection of hydrological extremes with the trends that can be derived from climate models?

**Response:** In the revised manuscript, the average annual reliability (AAR) method to quantify the probability that a hydraulic system would be safe over its planning period in univariate case (Salas and Obeysekera, 2014; Read and Vogel, 2015) is adopted because of its agreement with nonstationarity. We have accepted this good suggestions of predicting the future extremes in the future. Since the nonstationary models with time as covariates, it can be used to predict the extreme value in the future 5 years (2018-2022). Considering new dataset being added to the original data, the parameters of the time-varying or stationary models would change. We limited the years of future to 5 years. In the same way, the ending year of the design life period is set as 2022 (**Figure 5**).

In this study, the time-varying marginal and copula models are established on the observed dataset (1958-2017). It would show different trend analysis results with the synthetized time series from the seven GCM simulations (BCC-CSM 1.1, CanESM2, CNRM-CM5, CSIRO-Mk3.6.0, IPSL-CM5A-LR, IPSL-CM5A-MR, NorESM1-M) involved in CMIP5 under RCP4.5 scenarios as shown in Table R1. Based on the analysis of manuscript, the extreme values or their dependence structure showed a significant trend. So we compared the series from these 4 stations. The difference between the observed extreme series and the simulated series are obvious. And if we make the trend analysis based on the predicted data set from the GCM models for the period of 1961-2100, the trend is significant (Z for Ps of station 1 is -5.82 while Im of station 1 is -4.32) at 5% level. And the results of Mann-Kendall trend analysis is bound to be the length of extreme series. And the above of trend comparison is beyond our scope of this paper, we focus more on the guidelines of how to establish the nonstationary models and the importance of considering the nonstationarity in
```
```

hydrologic design by quantile estimation based a certain AAR level.

[Figure]

**Figure 5.** Nonstationary marginal distributions with trends in parameter during observed and predicted periods

**Table R1.** The trend analysis results of synthesized time series (1961-2017) from Seven GCM models

| Station No. | Attribute | AC Test | Univariate MK | Multivariate MK |
|---|---|---|---|---|
| | | p.value | Z | Z |
| 1 | *Ps* | 0.69(0.93) | -0.34(-1.72) | -0.197(-0.76) |
| | *Im* | 0.75(0.79) | 0.07(0.31) | |
| 5 | *Ps* | 0.67(0.89) | -1.32(-0.53) | -0.22(-1.25) |
| | *Im* | 0.65(0.57) | -0.81(-1.79) | |
| 6 | *Ps* | 0.57(0.99) | -0.98(-1.33) | -0.25(-1.32) |
| | *Im* | 0.79(0.97) | -0.68(-1.16) | |
| 8 | *Ps* | 0.67(0.53) | -0.69(-1.35) | -1.12(-1.38) |
| | *Im* | 0.58(0.52) | -0.58(-1.22) | |

Note: data in the bracket is corresponding value calculated on the observed dataset from 1961-2017.

```

3) Why do the Authors limits their analysis to six rain-gauges when there are several more in the area (e.g. https://doi.org/10.1002/hyp.9607)?

**Response:** We have taken the suggestions of the comment suggested by Reviewer 2. Also, in the following comment, the selected extreme series did not show a significant trend at 5% level. So we considered more stations in the Haihe River Basin. In addition, the original 95-th percentile threshold for $Ps$ has changed to 0.90. Because of the limitation data length and missing data, the daily precipitation with period 1958-2017 was available at 16 stations. Fortunately, the absolute value of Z statistics of the Ps of station 2 and dependence structure of station 6 is larger than 1.96 which the threshold value of 5% significance level. And also, the change-point (CP) tests recommend change point existed in the $Ps$ and dependence structure of $Ps$ and $Im$ from station 7.

Based on the trend and change-point analysis, the single extremes or their dependence structure of station 2, 6, 7 showed a monotonic trend or change point. And based on LR statistic tests which are used to test whether linear or nonlinear trend existed in parameter, only 1,5, 6 and 8 showed trend in parameters. For the space limitations, we only show eight stations which contains all the nonstationary series in these stations. And this study mainly focus on how to establish the nonstationary copula models to make the hazard assessment from the hydrologic design. Based on the LR and nonparametric tests (Mann-Kendall and change point), there are 11 stations out of these 16 stations did not have stationarity.

```

**Table R2** Trend analysis for the additional stations which are not shown in manuscript

| No. | Attribute | AC Test p.value | Univariate MK Z | Univariate CP Test p.value | Multivariate MK Z tatistics | Multivariate CP Test p.value |
|-----|-----------|------|------|------|------|------|
| 9 | Ps | 0.48 | -0.13 | 1.29 | -0.58 | 0.57 |
|   | Im | 0.76 | -0.94 | 0.56 | | |
| 10 | Ps | 0.70 | -0.93 | 0.61 | -0.86 | 0.40 |
|    | Im | 0.69 | -0.68 | 1.12 | | |
| 11 | Ps | 0.54 | -0.38 | 1.20 | -0.61 | 0.69 |
|    | Im | 0.17 | -0.76 | 0.88 | | |
| 12 | Ps | 0.14 | -1.71 | 0.09 | -1.58 | 0.11 |
|    | Im | 0.46 | -1.51 | 0.18 | | |
| 13 | Ps | 0.68 | -0.24 | 1.38 | -0.21 | 0.83 |
|    | Im | 0.85 | -0.13 | 1.41 | | |
| 14 | Ps | 0.54 | -0.41 | 0.75 | -0.31 | 0.43 |
|    | Im | 0.49 | -0.18 | 0.87 | | |
| 15 | Ps | 0.58 | -0.82 | 0.14 | -1.08 | 0.25 |
|    | Im | 0.94 | -1.25 | 0.93 | | |
| 16 | Ps | 0.53 | -1.47 | 0.32 | -0.91 | 0.35 |
|    | Im | 0.74 | -0.25 | 0.51 | | |

4) Which are the limits/problems relate to use an upper bounded distribution (i.e. GEV when κ <0) to describe variables that potentially range between [0 +∞).

**Response:** This is a formula expression error. In order to make the fit of marginal distribution fit more objectively, we take five kinds of extreme value distribution into consideration: two kinds of 3-parameter distribution (Generalized Extreme Value, GEV; Pearson type III, PIII) and three kinds of 2-parameter distributions (Gamma, Weibull and Lognormal).

```
```

5). There is a significant difference between quantiles reported in Figure 4 and those computed using equation 2 and parameters reported in bold in Tables 3(a) and 3(b) with t=1960, 1970, 1980, 1990, 2000, 2010, 2020.

**Response:** As stated in response to comment 4, the selected extreme series did not show a significant trend at 5% level. On the one hand, we considered more stations in the Haihe River Basin. On the other hand, the original 95-th percentile threshold for *Ps* has changed to 0.90. And potential marginal models are shown in **Table S1 (a)-(d)**. Finally, based on the LR tests, GOF tests and AICc criterion. For marginal distribution, the extreme attributes (*Ps* or *Im*) extracted from station 1, 5, 6, 7 and 8 showed nonstationarity. The nonstationarity for station 1 , 5, 6 and 8 was because of existed trend in parameter while the existed change point (year 1979) was the reason why incorporate the nonstationarity in station 7.

```

**S1 (a)** Patterns of the time-varying models for 3-parameter marginal distribution (GEV in this study)

| Model | $\mu$ | $\sigma$ | $\kappa$ |
|---|---|---|---|
| GEV0 | constant | constant | constant |
| GEV1 | $\mu = \mu_0 + \mu_1 t$ | constant | constant |
| GEV2 | $\mu = \mu_0 + \mu_1 t + \mu_2 t^2$ | constant | constant |
| GEV3 | constant | $ln\sigma = \sigma_0 + \sigma_1 t$ | constant |
| GEV4 | constant | $ln\sigma = \sigma_0 + \sigma_1 t + \sigma_2 t^2$ | constant |
| GEV5 | constant | constant | $\kappa = \kappa_0 + \kappa_1 t$ |
| GEV6 | constant | constant | $\kappa = \kappa_0 + \kappa_1 t + \kappa_2 t^2$ |
| GEV7 | $\mu = \mu_0 + \mu_1 t$ | $ln\sigma = \sigma_0 + \sigma_1 t$ | constant |
| GEV8 | $\mu = \mu_0 + \mu_1 t$ | constant | $\kappa = \kappa_0 + \kappa_1 t$ |
| GEV9 | constant | $ln\sigma = \sigma_0 + \sigma_1 t$ | $\kappa = \kappa_0 + \kappa_1 t$ |
| GEV10 | $\mu = \mu_0 + \mu_1 t + \mu_2 t^2$ | $ln\sigma = \sigma_0 + \sigma_1 t + \sigma_2 t^2$ | constant |
| GEV11 | constant | $ln\sigma = \sigma_0 + \sigma_1 t + \sigma_2 t^2$ | $\kappa = \kappa_0 + \kappa_1 t + \kappa_2 t^2$ |
| GEV12 | $\mu = \mu_0 + \mu_1 t + \mu_2 t^2$ | constant | $\kappa = \kappa_0 + \kappa_1 t + \kappa_2 t^2$ |
| GEV13 | $\mu = \mu_0 + \mu_1 t$ | $ln\sigma = \sigma_0 + \sigma_1 t + \sigma_2 t^2$ | constant |
| GEV14 | constant | $ln\sigma = \sigma_0 + \sigma_1 t$ | $\kappa = \kappa_0 + \kappa_1 t + \kappa_2 t^2$ |
| GEV15 | $\mu = \mu_0 + \mu_1 t$ | constant | $\kappa = \kappa_0 + \kappa_1 t + \kappa_2 t^2$ |
| GEV16 | $\mu = \mu_0 + \mu_1 t + \mu_2 t^2$ | $ln\sigma = \sigma_0 + \sigma_1 t$ | constant |
| GEV17 | constant | $ln\sigma = \sigma_0 + \sigma_1 t + \sigma_2 t^2$ | $\kappa = \kappa_0 + \kappa_1 t$ |
| GEV18 | $\mu = \mu_0 + \mu_1 t + \mu_2 t^2$ | constant | $\kappa = \kappa_0 + \kappa_1 t$ |
| GEV19 | $\mu = \mu_0 + \mu_1 t + \mu_2 t^2$ | $ln\sigma = \sigma_0 + \sigma_1 t$ | $\kappa = \kappa_0 + \kappa_1 t + \kappa_2 t^2$ |
| GEV20 | $\mu = \mu_0 + \mu_1 t + \mu_2 t^2$ | $ln\sigma = \sigma_0 + \sigma_1 t + \sigma_2 t^2$ | $\kappa = \kappa_0 + \kappa_1 t$ |
| GEV21 | $\mu = \mu_0 + \mu_1 t$ | $ln\sigma = \sigma_0 + \sigma_1 t + \sigma_2 t^2$ | $\kappa = \kappa_0 + \kappa_1 t + \kappa_2 t^2$ |
| GEV22 | $\mu = \mu_0 + \mu_1 t$ | $ln\sigma = \sigma_0 + \sigma_1 t$ | $\kappa = \kappa_0 + \kappa_1 t + \kappa_2 t^2$ |
| GEV23 | $\mu = \mu_0 + \mu_1 t + \mu_2 t^2$ | $ln\sigma = \sigma_0 + \sigma_1 t$ | $\kappa = \kappa_0 + \kappa_1 t$ |
| GEV24 | $\mu = \mu_0 + \mu_1 t$ | $ln\sigma = \sigma_0 + \sigma_1 t + \sigma_2 t^2$ | $\kappa = \kappa_0 + \kappa_1 t$ |
| GEV25 | $\mu = \mu_0 + \mu_1 t$ | $ln\sigma = \sigma_0 + \sigma_1 t$ | $\kappa = \kappa_0 + \kappa_1 t$ |
| GEV26 | $\mu = \mu_0 + \mu_1 t + \mu_2 t^2$ | $ln\sigma = \sigma_0 + \sigma_1 t + \sigma_2 t^2$ | $\kappa = \kappa_0 + \kappa_1 t + \kappa_2 t^2$ |

```

**S1 (b)** Patterns of the time-varying models for 3-parameter marginal distribution (PIII in this study)

| Model | $\mu$ | $\sigma$ | $\kappa$ |
|---|---|---|---|
| PIII0 | constant | constant | constant |
| PIII1 | $\mu = M * \sin(\mu_0 + \mu_1 t)$ | constant | constant |
| PIII2 | $\mu = M * \sin(\mu_0 + \mu_1 t + \mu_2 t^2)$ | constant | constant |
| PIII3 | constant | $\sigma = \sigma_0 + \sigma_1 t$ | constant |
| PIII4 | constant | $\sigma = \sigma_0 + \sigma_1 t + \sigma_2 t^2$ | constant |
| PIII5 | constant | constant | $ln\kappa = \kappa_0 + \kappa_1 t$ |
| PIII6 | constant | constant | $ln\kappa = \kappa_0 + \kappa_1 t + \kappa_2 t^2$ |
| PIII7 | $\mu = M * \sin(\mu_0 + \mu_1 t)$ | $\sigma = \sigma_0 + \sigma_1 t$ | constant |
| PIII8 | $\mu = M * \sin(\mu_0 + \mu_1 t)$ | constant | $ln\kappa = \kappa_0 + \kappa_1 t$ |
| PIII9 | constant | $\sigma = \sigma_0 + \sigma_1 t$ | $ln\kappa = \kappa_0 + \kappa_1 t$ |
| PIII10 | $\mu = M * \sin(\mu_0 + \mu_1 t + \mu_2 t^2)$ | $\sigma = \sigma_0 + \sigma_1 t + \sigma_2 t^2$ | constant |
| PIII11 | constant | $\sigma = \sigma_0 + \sigma_1 t + \sigma_2 t^2$ | $ln\kappa = \kappa_0 + \kappa_1 t + \kappa_2 t^2$ |
| PIII12 | $\mu = M * sin(\mu_0 + \mu_1 t + \mu_2 t^2)$ | constant | $ln\kappa = \kappa_0 + \kappa_1 t + \kappa_2 t^2$ |
| PIII13 | $\mu = M * \sin(\mu_0 + \mu_1 t)$ | $\sigma = \sigma_0 + \sigma_1 t + \sigma_2 t^2$ | constant |
| PIII14 | constant | $\sigma = \sigma_0 + \sigma_1 t$ | $ln\kappa = \kappa_0 + \kappa_1 t + \kappa_2 t^2$ |
| PIII15 | $\mu = M * \sin(\mu_0 + \mu_1 t)$ | constant | $ln\kappa = \kappa_0 + \kappa_1 t + \kappa_2 t^2$ |
| PIII16 | $\mu = M * \sin(\mu_0 + \mu_1 t + \mu_2 t^2)$ | $\sigma = \sigma_0 + \sigma_1 t$ | constant |
| PIII17 | constant | $\sigma = \sigma_0 + \sigma_1 t + \sigma_2 t^2$ | $ln\kappa = \kappa_0 + \kappa_1 t$ |
| PIII18 | $\mu = M * sin(\mu_0 + \mu_1 t + \mu_2 t^2)$ | constant | $ln\kappa = \kappa_0 + \kappa_1 t$ |
| PIII19 | $\mu = M * sin(\mu_0 + \mu_1 t + \mu_2 t^2)$ | $\sigma = \sigma_0 + \sigma_1 t$ | $ln\kappa = \kappa_0 + \kappa_1 t + \kappa_2 t^2$ |
| PIII20 | $\mu = M * sin(\mu_0 + \mu_1 t + \mu_2 t^2)$ | $\sigma = \sigma_0 + \sigma_1 t + \sigma_2 t^2$ | $ln\kappa = \kappa_0 + \kappa_1 t$ |
| PIII21 | $\mu = M * sin(\mu_0 + \mu_1 t)$ | $\sigma = \sigma_0 + \sigma_1 t + \sigma_2 t^2$ | $ln\kappa = \kappa_0 + \kappa_1 t + \kappa_2 t^2$ |
| PIII22 | $\mu = M * sin(\mu_0 + \mu_1 t)$ | $\sigma = \sigma_0 + \sigma_1 t$ | $ln\kappa = \kappa_0 + \kappa_1 t + \kappa_2 t^2$ |
| PIII23 | $\mu = M * \sin(\mu_0 + \mu_1 t + \mu_2 t^2)$ | $\sigma = \sigma_0 + \sigma_1 t$ | $ln\kappa = \kappa_0 + \kappa_1 t$ |
| PIII24 | $\mu = M * \sin(\mu_0 + \mu_1 t)$ | $\sigma = \sigma_0 + \sigma_1 t + \sigma_2 t^2$ | $ln\kappa = \kappa_0 + \kappa_1 t$ |
| PIII25 | $\mu = M * \sin(\mu_0 + \mu_1 t)$ | $\sigma = \sigma_0 + \sigma_1 t$ | $ln\kappa = \kappa_0 + \kappa_1 t$ |
| PIII26 | $\mu = M * \sin(\mu_0 + \mu_1 t + \mu_2 t^2)$ | $\sigma = \sigma_0 + \sigma_1 t + \sigma_2 t^2$ | $ln\kappa = \kappa_0 + \kappa_1 t + \kappa_2 t^2$ |

Note: M represents the minimum value of the observed time series.

```

**S1(c)** Patterns of the time-varying models for 2-parameter marginal distribution containing scale and shape parameter (Weibull, Gamma in this study).

| Model | $\sigma$ | $\kappa$ |
|---|---|---|
| WE0/GA0 | constant | constant |
| WE1/GA1 | $ln\sigma = \sigma_0 + \sigma_1 t$ | constant |
| WE2/GA2 | $ln\sigma = \sigma_0 + \sigma_1 t + \sigma_2 t^2$ | constant |
| WE3/GA3 | constant | $ln\kappa = \kappa_0 + \kappa_1 t$ |
| WE4/GA4 | constant | $ln\kappa = \kappa_0 + \kappa_1 t + \kappa_2 t^2$ |
| WE5/GA5 | $ln\sigma = \sigma_0 + \sigma_1 t$ | $ln\kappa = \kappa_0 + \kappa_1 t$ |
| WE6/GA6 | $ln\sigma = \sigma_0 + \sigma_1 t$ | $ln\kappa = \kappa_0 + \kappa_1 t + \kappa_2 t^2$ |
| WE7/GA7 | $ln\sigma = \sigma_0 + \sigma_1 t + \sigma_2 t^2$ | $ln\kappa = \kappa_0 + \kappa_1 t$ |
| WE8/GA8 | $ln\sigma = \sigma_0 + \sigma_1 t + \sigma_2 t^2$ | $ln\kappa = \kappa_0 + \kappa_1 t + \kappa_2 t^2$ |

**S1(d)** Patterns of the time-varying models for 2-parameter marginal distribution containing location and scale parameter (Lognormal function in this study).

| Model | $\mu$ | $\sigma$ |
|---|---|---|
| LOGN0 | constant | constant |
| LOGN1 | $\mu = \mu_0 + \mu_1 t$ | constant |
| LOGN2 | $\mu = \mu_0 + \mu_1 t + \mu_2 t^2$ | constant |
| LOGN3 | constant | $ln\sigma = \sigma_0 + \sigma_1 t$ |
| LOGN4 | constant | $ln\sigma = \sigma_0 + \sigma_1 t + \sigma_2 t^2$ |
| LOGN5 | $\mu = \mu_0 + \mu_1 t$ | $ln\sigma = \sigma_0 + \sigma_1 t$ |
| LOGN6 | $\mu = \mu_0 + \mu_1 t$ | $ln\sigma = \sigma_0 + \sigma_1 t + \sigma_2 t^2$ |
| LOGN7 | $\mu = \mu_0 + \mu_1 t + \mu_2 t^2$ | $ln\sigma = \sigma_0 + \sigma_1 t$ |
| LOGN8 | $\mu = \mu_0 + \mu_1 t + \mu_2 t^2$ | $n\sigma = \sigma_0 + \sigma_1 t + \sigma_2 t^2$ |

```
```

**Table 4(a).** Results of marginal models with no trend for the distribution parameters corresponding to both extreme attributes

| Station | Attribute | Model | $\mu$ | $\sigma$ | $\kappa$ | *AICc* | *KS* |
|---|---|---|---|---|---|---|---|
| 2 | *Ps* | LOGN0 | 5.49 [5.42,5.57][a] | -0.95 [-1.10,-0.81] | — | 720.40 | 0.88 |
| | *Im* | GEV0 | 61.11 [55.58,69.15] | 3.30 [3.11,3.44] | 0.24 [0.065,0.43] | 607.31 | 0.99 |
| 3 | *Ps* | GA0 | — | 2.17 [1.85,2.41] | 2.79 [2.55,3.13] | 599.85 | 0.95 |
| | *Im* | GEV0 | 35.08 [32.85,38.23] | 2.43 [2.61,2.23] | 0.25 [0.055,0.47] | 504.21 | 0.79 |
| 4 | *Ps* | GA0 | — | 2.64 [2.31,2.87] | 2.51 [2.26,2.82] | 638.59 | 0.84 |
| | *Im* | GA0 | — | 1.80 [1.44,2.04] | 2.06 [1.80,2.39] | 508.01 | 0.85 |
| 7 | *Ps*[b] | GA0 | — | 3.46 [2.67,3.93] | 2.12 [1.71,2.85] | 252.40 | 0.72 |
| | *Ps*[c] | LOGNO | 5.28 [5.20,5.39] | -1.03 [-1.24,-0.87] | — | 435.43 | 0.81 |
| | *Im* | GA0 | — | 2.44 [2.09,2.66] | 1.94 [1.68,2.34] | 577.74 | 0.99 |

[a]parameter uncertainties with 90% confidence bands; *Ps*[b] represents the *Ps* time series before change point 1979. *Ps*[c] represents the *Ps* time series after change point 1979.

```
```

**Table 4(b).** Results of nonstationary marginal models with a certain trend for the parameters

| Station | Attribute | Model | $\mu$ | | | $\sigma$ | | | $\kappa$ | $AIC_c$ | $LR^2$ | $KS$ |
|---|---|---|---|---|---|---|---|---|---|---|---|---|
| | | | $\mu_0$ | $\mu_1$ | | $\sigma_0$ | $\sigma_1$ | | | | | |
| 1 | *Ps* | LOGN5 | 5.64 | -0.0031 | | -1.07 | -0.0069 | | — | 691 | 0.021 | 0.75 |
| | | | [5.51,5.78] | [-0.0065, 0.0022] | | [-1.46,-0.84] | [-0.018,0.0022] | | | | | |
| | *Im* | LOGN0 | 4.03 | | | -1.04 | | | — | 533 | — | 0.66 |
| | | | [3.93, 4.09] | | | [-1.17,-0.93] | | | | | | |
| 5 | *Ps* | GAO | — | | | 2.45 | | | 2.57 | 620 | — | 0.45 |
| | | | | | | [2.12,2.72] | | | [2.34,2.89] | | | |
| | | | $\mu_0$ | $\mu_1$ | | $\sigma_0$ | $\sigma_1$ | $\sigma_2(*1e-4)$ | | | | |
| | *Im* | LOGN6 | 3.83 | -0.0041 | | -1.37 | 0.036 | -7.8 | — | 479 | 2.2e-4 | 0.73 |
| | | | [3.69,3.96] | [-0.0076,-0.0012] | | [-2.08,-1.02] | [0.0048,0.091] | [-15.6,-3.5] | | | | |
| 6 | | | $\mu_0$ | $\mu_1$ | | $\sigma_0$ | $\sigma_1$ | $\sigma_2(*1e-4)$ | | | | |
| | *Ps* | LOGN6 | 5.67 | -0.006 | | -0.57 | -0.04 | $5.8^2$ | — | 712 | 1.2e-5 | 0.68 |
| | | | [5.50,5.82] | [-0.0098,-0.0014] | | [-1.07,-0.34] | [-0.067,-0.031] | [-0.8,10.5] | | | | |
| | *Im* | GEV0 | 58.84 | | | 3.10 | | | 0.283 | 587 | — | 0.57 |
| | | | [53.92,65.44] | | | [2.91,3.29] | | | [0.078,0.43] | | | |
| 8 | | | $\mu_0$ | $\mu_1(*1e-2)$ | $\mu_2(*1e-4)$ | $\sigma_0$ | $\sigma_1$ | $\sigma_2(*1e-4)$ | | | | |
| | *Ps* | LOGN8 | 5.680 | -2.1 | $2.7^2$ | -1.07 | 0.032 | $-7.7^2$ | — | 702 | 8.5e-4 | 0.95 |
| | | | [5.40,5.93] | [-3.9,-0.5] | [0.3,5.3] | [-1.72,-0.80] | [-0.0016,0.071] | [-14.4,-3.2] | | | | |
| | *Im* | LOGN0 | 4.18 | | | -0.86 | | | — | 572 | — | 0.68 |
| | | | [4.10,4.27] | | | [-1.02,-0.74] | | | | | | |

[1]parameter uncertainties with 90% confidence bands;[2] this value is value in the*1e-4 because of the table space limitations;[2]p.value of LR tests.

```
```

6) In the presence of a statistically significant change point, is it correct to use the same parameter formulation to describe data "before" and "after" the abrupt change point? Moreover, which would be the results of trend analysis if the time series are split as before" and "after" the change point. Do the Mann-Kendall test' results change considering the before" and "after" segment of the time series separately?

**Response:** Firstly, thanks a lot for this suggestion by reviewer 1. Your idea is just in accordance with the paper by Salvadori et al. (2018), which is just the correct and objective procedure when the change point did exist in the extreme series. Following this suggestions, we made univariate and multivariate change point analysis for these 8 stations with results in **Table 3**. The *Ps* and dependence structure of *Ps* and *Im* of station 7 showed change point at year 1979. Also the Mann-Kendall (MK) trend analysis was also implemented for these two spilt series. For station 7, ***Before change point (1979),*** Z=0.26 for univariate MK test for *Ps*, Z=0.99 for Multivariate MK tests; ***After change point (1979)*** Z=0.45 for univariate MK for *Ps* and Z=0.62 for Multivariate MK tests. Compared to the MK tests for the whole data, the statistics (Z) changed a lot from 1.54 to 0.26 and 0.45 for *Ps*. The same situation existed in Multivariate MK tests. For the sake of showing more visually, the change point at year 1979 is plotted in **Figure 3**. In the following analysis of ARR quantiles, the quantile of each AAR quantile was calculated for these two spilt series no matter in univariate or bivariate case.

```
```

[Figure]

**Figure 3.** Plots of the change point according to the CP tests**.**

```
```

Table 3. Detection of trends and change points in extreme rainfall attributes collected from six stations

| Station No. | Attribute | AC Test | Univariate MK | Multivariate MK | Univariate CP Test | | Multivariate CP Test | |
|---|---|---|---|---|---|---|---|---|
| | | p.value | Z | Z | p.value | CP | p.value | CP |
| 1 | *Ps* | 0.93 | **-2.03**[*] | -0.86 | 0.069 | — | 0.062 | — |
| | *Im* | 0.79 | 0.42 | | 0.48 | — | | — |
| 2 | *Ps* | 0.79 | -0.63 | -0.43 | 0.67 | — | 0.70 | — |
| | *Im* | 0.62 | -0.16 | | 0.53 | — | | — |
| 3 | *Ps* | 0.19 | -0.20 | -0.72 | 0.96 | — | 0..66 | — |
| | *Im* | 0. 32 | -1.13 | | 0.33 | — | | — |
| 4 | *Ps* | 0.05 | 0.71 | 0.61 | 0.64 | — | 0.46 | — |
| | *Im* | 0.17 | -0.76 | | 0.43 | — | | — |
| 5 | *Ps* | 0.89 | -0.82 | -1.48 | 0.14 | — | 0.15 | — |
| | *Im* | 0.57 | -1.93 | | 0.071 | — | | — |
| 6 | *Ps* | 0.99 | -1.91 | **-2.02**[*] | 0.16 | — | 0.055 | — |
| | *Im* | 0.97 | -1.93 | | 0.12 | — | | — |
| 7 | *Ps* | 0.38 | -1.54 | -1.70 | **0.033** | 1979 | **0.025** | 1979 |
| | *Im* | 0.77 | -1.66 | | 0.11 | — | | — |
| 8 | *Ps* | 0.53 | -1.45 | -1.45 | 0.17 | — | 0.31 | — |
| | *Im* | 0.52 | -1.26 | | 0.33 | — | | — |

MK: Mann-Kendall tests; CP: change point tests; AC: Autocorrelation tests; For station 7, ***Before change point (1979),*** Z=0.26 for univariate MK test for *Ps*, Z=0.99 for Multivariate MK tests; ***After change point (1979)*** Z=0.45 for univariate MK for *Ps* and Z=0.62 for Multivariate MK tests.

```
```

7) As last I would like to suggest the Author to add:

a. A section on climate change projection and analysis that can be of interest for future infrastructure design

b. one table reporting the basic statistics (min/max/mean/standard deviation) of the Ps and Im variables and the values of the 95-th percentile threshold to help understanding the variability of datasets;

c. one figure showing the time series with the indication of the change point year of occurrence according to Pettitt test.

**Response:** for suggestion (a), we have add this suggestion in section 3.3 as follows: "Since the nonstationary models with time as covariates, it can be used to predict the extreme value in the future. Considering new dataset being added to the original data, the parameters of the time-varying models would change. In this study, we assume the parameters of the selected time-varying models did not change a lot in the future 5 years. So the estimated time varying models from the original extreme series were used to predict year 2018-2022 (t=61-65). Based on the same assumption, the ending year of design life period in the following AAR-based quantiles calculation is set as 2022. As shown in **Figure 5**, mean value of Ps from station 1 and station 6 exhibited a downward trend while mean value of Im from station 5 and Ps from station 8 exhibited a NMT trend. For the predicted period, the predicted nonstationary marginal distributions for the extremes extracted from these 4 stations presented smaller mean values than those of the stationary distributions. Furthermore, the divergence of mean values between them are becoming larger as time goes on for

```
```

station 1, 5 and 6. But for Ps from station 8, the divergence of mean values between them are becoming smaller as time goes on."

And in section 3.5-"Nonstationary hazard assessment based on AAR metrics for univariate and bivariate cases", the design life period is set as 1983-2022. The future 5 years from 2018 to 2022. Considering new dataset being added to the original data, the parameters of the time-varying or stationary models would change. We limited the years of future to 5 years for cautiousness. These two adding parts are of great interest for the future hydrologic design.

For suggestion (b), we add **Table 2.** to show the basic statistics (min/max/mean/standard deviation) of the *Ps* and *Im* variables and the values of the 90-th percentile threshold to help understanding the variability of datasets.

**Table 2.** Basic statistics of the extreme variables observed at the original 8 stations

| No. | ID | Ps(mm) | | | | | Im(mm) | | | |
|-----|-----|------|------|------|------|------|------|------|------|------|
| | | Min | Max | Mean | Sd | $\overline{Th}$ | Min | Max | Mean | Sd |
| 1 | 53588 | 127.9 | 549.3 | 269.0 | 81.2 | 22.0 | 28.0 | 113.4 | 59.6 | 21.2 |
| 2 | 53986 | 80.1 | 636.2 | 263.2 | 107.7 | 26.6 | 24.3 | 414.0 | 85.0 | 56.5 |
| 3 | 54208 | 70.1 | 229.6 | 143.0 | 35.5 | 16.4 | 20.3 | 154.7 | 45.2 | 22.3 |
| 4 | 54311 | 65.8 | 299.2 | 172.5 | 49.1 | 18.7 | 18.1 | 100.9 | 47.3 | 17.3 |
| 5 | 54401 | 65.9 | 273.4 | 152.7 | 42.1 | 18.1 | 19.9 | 100.4 | 43.3 | 15.8 |
| 6 | 54511 | 102.2 | 865.9 | 263.6 | 115.1 | 28.7 | 32.8 | 253.5 | 79.7 | 44.7 |
| 7 | 54518 | 84.1 | 437.3 | 231.8 | 87.6 | 26.5 | 29.8 | 181.4 | 79.8 | 30.6 |
| 8 | 54602 | 87.2 | 559.0 | 230.1 | 93.1 | 27.3 | 23.7 | 185.6 | 71.8 | 32.9 |

$\overline{Th}$: represents the mean value of the 90% percentile threshold value for the attribute *Ps*.

For suggestion (c), we plot the change point for the Ps extreme series from station 7 (**Figure 3**).

```
```

Specific comments

Line 116 The definition given of Ps variable recall me the index R95pTOT used in climate change studies (http://etccdi.pacificclimate.org/list_27_indices.shtml). Is it the same index? In addition, could the Authors specificy the period of observation they used to set the 95-th percentile threshold? According to R95pTOT index the reference period to set the 95-th percentile threshold is 1961-1990.

**Response:** In this study, the Ps is different from the R95pTOT index. It is just annual total precipitation of the daily precipitation more than the 90th percentile threshold for each year. That means, each year has a unique threshold value. We have made its definition more readable and clear.

Line 290 -291 The Authors write where $R_i^{ns}$ and $R_i^s$ are nonstationary risk and stationary risk of a certain hydraulic structure for a design life of i years", but 'i' goes from 1 to n. I would expect that 'n' indicates the design life and 'i' indicates the i-th year from now (i.e. the year the project "starts") to n-th year (end of the project's life).

**Response:** This risk quantified metric is deleted from the manuscript. We accept the average annual reliability (AAR) method to quantify the probability that a hydraulic system would be safe over its planning period in univariate or bivariate case (Salas and Obeysekera, 2014; Read and Vogel, 2015).

```

Lines 367-368 The Authors write "Except for stations 4 and 5, the best distributions for the other stations were parallel for nonstationarity tests shown in Section 4.1". Is it possible that the mismatch between the nonstationarity test results and the best fitting distribution for Im (station 4 and 5) and Ps (station 5) was to the choice of the Author to ignore the test's results?

**Response:** as stated by the reviewer, it is of great possibility the mismatch between the nonstationarity test results and the best fitting distribution for Im (station 4 and 5) and Ps (station 5) was to the choice of the Author to ignore the test's results. And we use 10% significance level which entailing a large probability of rejecting Null Hypothesis of non-stationarity. So in revised forms, we take a careful statistical tests with 5% significance level for Mann-Kendall (MK), change point tests. We also proposed the Log Likelihood ratio (LR) tests (Coles (2001)), which is more rigorous trend detection methods than nonparametric methods (MK). For most cases, the results of trend analysis from LR tests are consistent with that by MK tests. However, as shown in Table 3(b), the trend in the parameter existed in Im of station 5, Ps of station 6 and 8 based on the LR tests at the 5% significance level which recommends another situation different from the previous MK tests. Through analysis, it would be caused by the he opposing trend in the location and scale parameters.

Lines 387-390 The Authors write "Contrary to station 5, the nonstationary St copula fitted better than did the stationary model for stations 1 and 6 which was not in accordance with the nonstationarity tests for these two stations (Table 2)." It is true that

```

according to bivariate MK test results station 1 and 6 should stationary, but at station 1 bivariate Pettitt test shows the presence of a change point; the presence of a change point could have influenced the results of LL and AICc ? What will happen if Im and Ps time series are "broken" before and after the change point to LL and AICc estimates?

**Response:** we have taken this suggestion into considerations. The extreme series from station 7 in the revised manuscript showed change point. The *Ps* and dependence structure of *Ps* and *Im* of station 7 showed change point at year 1979. Also the Mann-Kendall (MK) trend analysis was also implemented for these two spilt series. For subseries, no trend can be detected based on Mann-Kendall and LR tests.  The best marginal distribution and copula is shown in **Table 4(a)** and **Table 5(a)**.

Line 411 (and Conclusions) The Authors report a value of 355 mm for the 100-year Ps quantile in station 1 under stationary circumstances, but using the parameters reported in Table 3(a) the 100-year Ps quantile in station 1 under stationary circumstances is about 383 mm. It is probably a matter of approximation in the parameters values (355 mm corresponds to a report period of about 62 yr) but I will suggest the Authors to check these values.

**Response:** Since average annual reliability (AAR) method to quantify the probability of the hydraulic structure is of great potentiality in communicating hazard of failure under both stationary and nonstationary conditions (Read and Vogel, 2015), we adopt ARR method to estimate the quantiles for a design life period including future 5 years.

```

**Table 5(a).** The stationary copula for the dependence structure of the *Ps* and *Im* for station 1-5 and 7

| Station | Model | Kendall's tau | $\theta$ | $\beta$ | *AICc* | $AD_{RT}$ |
|---------|-------|---------------|----------|---------|--------|-----------|
| 1 | FR0 | 0.55 | 6.65 [4.53,8.36][1] | — | -43.69 | 0.46[3] |
| 2 | JC0 | 0.65 | -0.22 [-0.55,0.18] | -1.22 [-1.29,-1.15] | -81.59 | 0.77 |
| 3 | SJC0 | 0.56 | -0.78 [-1.15,-0.52] | -1.23 [-1.29,-1.16] | -57.27 | 0.64 |
| 4 | SJC0 | 0.58 | -0.35 [-0.98,0.11] | -1.28 [-1.44,-1.18] | -63.46 | 0.76 |
| 5 | SJC0 | 0.55 | -0.26 [-0.61,0.01] | -1.34 [-1.46,-1.25] | -57.81 | 0.88 |
| 7[1] | FR0 | 0.4 | 4.43 [2.21,8.69] | — | -5.71 | 0.76 |
| 7[2] | SGU0 | 0.61 | 0.43 [-0.58,0.70] | — | -34.73 | 0.63 |
| 8 | SGU0 | 0.58 | 0.33 [0.021,0.61] | — | -56.63 | 0.59 |

```
```

Minor corrections

Around the manuscript there are some typos like "Pettist" instead of "Pettitt"; missing spaces and so on (e.g Lines 226, 228), please check the text.

**Response:** Following the study of Salvadori et al. (2018), the non-parametric change-point statistical tests were implemented to check that whether the marginal or joint distributions are sensitive to changes. These tests can be manipulated in the R package npcp (Kojadinovic, 2017). We replaced Pettitt tests with the above statistical change point tests as suggested by Review #2.

Lines 172 and 173 Is the limit "(μ-σ)/κ" for lower (upper) boundary of x value correct? According to parameter's estimates in Tables 3(a)-3(b), when κ<0, x can assume only negative values, that is non coherent with the variables Ps and Im that are positively defined.

**Response:** It is an error of formula definition.

Lines 249-250 The Authors write "Let $JRP_{s-and}$ and $JRP_{s-ken}$ represent the three types of return period in the stationary case", but the return periods presented are only 2.

**Response:** it is an error of station. It should be stated as follows: Let we calculate the joint return period from the AND and Kendall Scenario.

Line 260 "JPRs" probably was "JRPs"

**Response:** we have made deep self-checking of the notations.

```

Line 343 and Line 426 check the correct location of Figure 3

**Response:** we have checked this figure's location.

Lines 359-365 The Authors write "The best fitted model was selected by performing the minimum *DIC* criterion combined with the Bayes factor (*BF*) test", but looking at bold rows in tables 3(a) and 3(b) the criterion of minimum DIC seems not be respected for Im at station 2 where EVns-1 is in bold instead of GEVns-2 (minimum DIC value).

**Response:** The reviewers' suggestions and views are right. The selected extreme series from the manuscript did not show a significant trend at 5% level. We have checked the process of extreme value extraction. We have changed the original 95-th percentile threshold for *Ps* has changed to 0.90-th percentile for *Ps*. And we take data from more stations into consideration. And based on the analyzed results, the nonparametric tests were consistent with the LR tests in most cases.

Line 360-365 Comparing these lines with Table 3(a), for station 1, the variable described as GEVns-2 appears to be Ps and not Im. BF for Im variable in station 1 is >1. Please clarify this point.

**Response:** in the revised manuscript, we use the LR tests to select the nonstationary models with trend in parameter. The process can be defined as follws:

"Let ST represent the stationary model for the extreme attribute (Ps and Im in this study) and let NST be the time-varying model with trend existed in the parameter. In the same way, let $LL_1$ and $LL_0$ denote the log-likelihoods under model ST and NST.

```

The log likelihood ratio statistics (LR) would converge to the $\chi_\Delta^2$ distribution with $\Delta$ being the divergence of the parameter number between the ST and NST models. And LR can be formulated as follows (Coles, 2001):

$$LR = 2(LL_1 - LL_0) \tag{3}$$

The LR statistics can be regarded as a good criterion to check whether the Null trend hypothesis ( $\mathcal{H}_0$: there is no trend for the distribution parameter ) can be rejected or not. If the value of LR is more than the upper-$\alpha$ point of the $\chi_\Delta^2$ distribution, the above Null assumption can be rejected at a significance level of $\alpha$ (in this study $\alpha$ is equal to 5%)."

Line 387 Please define "MK"

**Response:** we have taken this suggestion to show the full definition of the Abbreviation.

Lines 409-411 Figure 3 illustrates the results of nonstationary tests. Figure 4 reports the extreme rainfall quantiles. Please check the text.

**Response:** We have checked the text.

###############################################################################

References

```

Line 62 Does "Assia et al., 2014" refer to "Aissia, M.A.B., Chebana, F., Ouarda, T.B.M.J., Roy, L., Bruneau, P., and Barbet, M.: Dependence evolution of hydrological characteristics, applied to floods in a climatechange context in Quebec, J. Hydrol., 519, 148–163, https://doi.org/10.1016/j.jhydrol.2014.06.042, 2014" ?

**Response**: We have made revision for the reference quotation.

Line 98 Does "(Jakob, 2013)" refer to "Jakob, D., AghaKouchak, A. Easterling, D., Hsu, K., Schubert, S., and Sorooshian, S. (Eds.): Nonstationarity in extremes and engineering design, Springer, New York, 2013" ?

**Response:** We have made revision for the reference quotation.

Line 100 "Read and Vogel (2015)" there is no correspondence in the references

**Response:** we have added the "Read, L.K., and Vogel, R.M.: Reliability, return periods, and risk under nonstationarity, Water Resour. Res., 51, 6381-6398, https://doi.org/10.1002/2015WR017089, 2015." to reference list.

Line 126 Does "Nelson (2007)" refer to "Nelsen, R.B.: An introduction to copulas, Springer, New York, 2007."?

**Response:** We have made revision for the reference quotation.

Line 212 "Genest et al., 1995" there is no correspondence in the references

**Response:** In revised manuscript, this reference is deleted

```

Line 213 "Hurvich and Tsai, 1989" there is no correspondence in the references

**Response:** we have added the "Hurvich, C. M. and Tsai, C. L.: Regression and time

series model selection in small samples, Biometrika, 76, 297–307,

https://doi.org/10.2307/1271469, 1989."   to the reference list

Line 235 "Fernandez and Salas, 1999" there is no correspondence in the references.

**Response**: we have used AAR metrics which did not contain the reference.

Ghanbari, M., M. Arabi, J. Obeysekera, and Sweet, W.: A coherent statistical model for

coastal flood frequency analysis under nonstationary sea level conditions, Earth's

Future, 7, 162-177, https://doi.org/10.1029/2018EF001089, 2017." The publication

year is 2019.

**Response:** we have changed the publication year.

"Zhang, Q. , Gu, X. , Singh, V. P. , and Chen, X.: Evaluation of ecological instream

flow using multiple ecological indicators with consideration of hydrological alterations,

J. Hydrol., 529, 711-722, https://doi.org/10.1016/j.jhydrol.2015.08.066, 2015 . " should

be moved at the end of the reference list.

**Response:** we have deleted the reference which is not included in the manuscript.

```

##################################################################

Table (1) I suggest the Authors to change Longitude and Latitude with Longitude E and Latitude N, respectively coherently with the choice of indicating geographical coordinates in degree/minutes format.

**Response:** we have accepted the suggestion and made revision to Table 1.

Table (2) "Ps" and "Im" should be in italic. For station 3 and multivariate MK test the "*" should be close to the Z-statistic value not to the p-value. I suggest the Authors to add the indication of the year at which the change point is detected for both univariate and bivariate Pettitt test.

**Response:** we have accepted the suggestion and showed the change point if it exists by change point tests. "Ps" and "Im" have be made in italic form.

Table 3(a) e 3(b) please specify the meaning of bold row, I guess that bold indicates the "best" fitting model, but in this case why for Im variable at station 2 the best model is GEVns-1 if GEVns-2 shows the minimum DIC?

**Response:** we have taken this suggestion in the revised manuscript.

Table 3(b) refers to (Station 4-6) not to (Station 2-6) and the '-' symbol is missing for BF values of stationary GEV in station 5.

**Response:** we have taken this suggestion in the revised manuscript.

```

Table 4(a) and 4(b) reports the meaning of bold and underlined text.

Infinity symbol cited in caption does not appear in the table, probably substituted by "NaN".

**Response:** we have taken this suggestion in the revised manuscript and improved the definition of table symbols.

####################################################################

Figure (1) Step S2 is omitted.

Response: we have taken this suggestion in the revised manuscript and improved the logical order of the flowchart.

Figure (2) I would like to suggest the Authors to add the Haihe river to the map.

**Response:** we have add the main stream of Haihe River to the map.

```

[Figure]

**Figure 2.** Selected meteorological stations in Haihe River basin

```
```

Figure (3) In the caption there is a typo "Mann-Kendalld" instead of "Mann-Kendall". Please check the legend, the description of the last item (purple backward arrow) is equal to the one of the third one (green upward arrow). The "+" symbol is redundant with the test that already specify if the trend/change point is statistically significant.

**Response**: we have taken this suggestion in the revised manuscript

Figure (4) the "star" symbol is not defined.

**Response:** we have checked clearly in revised manuscript.

Figures (4), (6), (7) I would like to suggest the Authors to improve the quality of these figures.They seems to be a collection of screenshots with different size and background colour. Figure 4, in particular, seems to lack of organization in the sub-figures arrangement.

**Response:** we have taken this suggestion to improve the quality of the figures.

```

---

## Author Comment (AC2) · 20 Dec 2019

SPECIFIC COMMENTS. Line(s) 49–54. Authors. Copulas, a useful tool for modelling the structure of dependence between hydrological variables regardless of the types of marginal distributions, have been widely used for multivariate frequency analysis + references... Referee. Historically, the paper by Salvadori and De Michele (2004) was the first one to deal with (copula) multivariate frequency analysis—later works are copies or small variants: this paper is not cited. Please, always give credits to whom deserve credits. Response: We have added this reference to the revised manuscript.

Line(s) 75–ff. Authors. There are three kinds of joint return period methods... Referee. NO. In Literature there are, at least, four kinds of joint return periods. The references

given are incorrect. In Salvadori and De Michele (2004) the OR, AND and Kendall cases were first introduced. In Salvadori et al. (2013) a further survival-Kendall approach (not mentioned by the Authors) was outlined. Referencing is often imprecise, almost random: for instance, why citing Jiang et al. (2015) here? It has nothing to do with the original formalization of the four return periods mentioned above. Incidentally, the reference "Salvadori and Michele, 2010" is "Salvadori and De Michele, 2010" (it seems that the Authors wrote the references by hand, instead of using some suitable software...). Response: we have modified the above statement about the joint return period. And give credits to whom deserve credits at right places. Lines 315-325 in revised manuscript.

Line(s) 95–97. Authors. Note that following the idea of Rootzén and Katz (2013) we regard the term hydrological risk as the possibility of a certain extreme event occurring and not as a quantification of expected losses. Referee. Then, probabilistically and statistically speaking (and hydrologically as well!), you should better use the term "hazard" instead of "risk". Response: We take the advice by reviewer. And we modified this statement. And also we have changed the title to "Time-varying copula and average annual reliability-based nonstationary hazard assessment of extreme rainfall events."

Line(s) 126. Authors. Detailed information about copulas can be found in Nelson (2007). Referee. NO. It is Nelsen (2006), not Nelson. For an engineering approach, you may also cite Salvadori et al. (2007). As a strong suggestion, the Authors should carefully check the correctness of all the references (it is easy to do it on the Internet), and add the missing ones. Response: we have revised the quotation of this reference. Add Salvadori et al. (2007) to revised manuscript as follows: "Detailed information of theoretical derivation about copulas can be found in Nelsen (2006). For the practical guidelines from hydrological point of view, it is recommended to refer to Salvadori et al. (2007)."

Line(s) 138–139. Authors. ... $\theta\_C\hat{}t$ is the dynamic copula parameter which is a linear function of time. Referee. The Authors must justify this choice. Please do not reply

that "the model was taken from this or that paper": it is not a scientific reason, for a model must be validated on the available data. Also, the results of suitable Goodness-of-Fit statistical tests must be shown. Response: As shown in revised manuscript (equation (2) in lines 256-257), both the linear and quadratic models are assumed as the possible function relation between parameter and time, which would consider the linear and nonlinear trend for parameter. We implemented the goodness of fit test for copulas based on Rosenblatt's transformation (Lines 271-283 in revised manuscript) (Rosenblatt, 1952). The detailed information about this can be found in supplement.

Line(s) 143–144. Authors. It is however possible that the nonstationary behavior may exist in both the marginal and joint distribution function. Referee. Such an issue was already clearly pointed out and discussed in Salvadori et al. (2018), where a similar case study was investigated, and a thorough statistical analysis was carried out. The Authors must mention this fact, and follow the (proper statistical) guidelines outlined in that paper. Response: We have revised this statement. And we mentioned this fact as: Lines 134-160. "According to Vezzoli et al. (2017) and Salvadori et al. (2018), a comprehensive statistical analysis which can check the presence of trend and change point should be carried out before we incorporate the nonstationarity into the multivariate hazard assessment. Following the study of Salvadori et al. (2018), the non-parametric change-point statistical tests were implemented to check that whether the marginal or joint distributions are sensitive to changes. These tests can be manipulated in the R package npcp (Kojadinovic, 2017)."

Line(s) 158-Figure 1. The flow-chart shown in Figure 1 provides wrong indications (see also later comments). In fact, the Authors confuse GoF tests with selection criteria. The flow-chart must be rewritten. Response: we have revised the flowchart of this study (Figure 1) in recised manuscript.

Line(s) 161–164. Authors. In this part, the Generalized Extreme Value (GEV) distribution was used to... (Cheng and AghaKouchak, 2014). Referee. This reference makes little sense: the features of the GEV have already been stated and described

since decades in other (seminal) works. Please use proper references. Response: In revised manuscript, five kinds of marginal distribution have been used as candidate distribution. For space limitations, we have deleted the above statement of GEV features.

Line(s) 166–ff. Authors. The GEV distribution consists of three control parameters... Referee. The GEV distribution is well known to hydrologists, there is no need to tell again a story that everybody knows. Response: we deleted this statement in revised manuscript.

Line(s) 176–179. Authors. In this study, two kinds of nonstationary GEV models (GEVns-1 and GEVns-2) are developed with the shape parameter being constant. It should be emphasized that modelling the time variance in shape parameter needs long-term observations, which are often not available in practice (Cheng et al., 2014). Referee. I recently rejected a paper very similar to the present one, where the GEV shape parameter was kept constant. The shape parameter is the most important one, for it rules the generation of extremes. The assumption adopted is definitely questionable: what (extreme) climate change could you really hope to model with a constant shape parameter? Practically, you are trying to model climate changes where the statistics of the extremes do not change with time: it makes little sense. In addition (see also later comments), some estimates of the GEV shape parameter are positive and other negative (Table 3). This entails that, in some cases, the corresponding GEV law is upper-bounded, i.e. unable to model an extreme behavior: this is a well known feature of the GEV. I agree that the GEV is the right distribution to be used in your analysis (Block Maxima), but the question is: how can you claim that the phenomenon you are modeling is an extreme one when upper-bounded GEV's are involved? The statistical results seem to tell another story... Response: Thanks a lot for this suggestion. We have consider the trend in shape parameter. For all the three parameter (location, scale, shape), three kinds of forms are considered. Here we take the location parameter as an example. In supplement information, we shown the candidate marginal and

copula models. In this study, When $\kappa<0$, it recommends the upper-bounded GEV distribution. We have checked the shape parameter of the best fitted GEV models is always positive (station 2,3,6 in Table 4(a)-(b)).

Line(s) 184–186, Eq.s (3)–(4). You must justify the assumptions/relations implicit in these equations. Why should the position and scale parameters change according to Eq.s(3)–(4)? Did you carry out any valuable/reliable fit? What are the p-Values? And, again, why should the shape parameter be constant instead? Incidentally, these are the same equations used in the paper I recently rejected. Response: We have taken this advice by considering trend in shape parameters as shown in the former response. Also we have taken the nonstationary K-S tests on the assumed linear and nonlinear trend (Table 4(a)-(b)).

Line(s) 191–194. Authors. Simultaneously, the Deviance Information Criterion (DIC) and Bayes factors (BF) for different stationary and nonstationary models were calculated to select the best fitted marginal model. The minimum DIC value yielded the best performance, while BF smaller than 1 indicated the best fitting. Referee. This is a typical fatal error of practitioners. These are only selection criteria, not Goodness-of-Fit tests. You must first use (non-stationary) GoF tests to check whether a model is admissible! Otherwise, without first checking the models via suitable GoF tests, you may end up choosing non-admissible ones. This work has no statistical bases. Response: We have taken this advice by investigating the univariate and bivariate GOF tests as presented in (lines 225-233 and lines 267- 283).

Line(s) 191–194. Authors. In multivariate hydrological frequency analysis, two kinds of copulas, named elliptical and Archimedean copulas are widely used in hydrological applications. Referee. So what? The fact that these copulas were used in other works is not, and cannot be, a scientific justification. This is the usual approach of practitioners that use the copulas provided by Matlab. Given my experience, I do not really think that Nature (especially considering the generation of Extremes) gets stick to just these dependence structures—see also later comments. And, worst of all, you

did not even check these copula models via suitable multivariate GoF tests (which are available in Literature, and some certified software is even for free—see below): this work has no statistical bases. Response: In the revised manuscript, we firstly enrich the kinds of candidate copulas: five kinds of 1-parameter copula (Joe, Frank, Gumbel, Clayton and Gaussian) and five kinds of 2-parameter copulas (Clayton-Gumbel (BB1), Student t, Joe-Gumbel (BB6), Joe-Clayton (BB7) and Joe-Frank (BB8)). In addition, each copula can be rotated at 90, 180 (Survival Copula), 270 degrees. For this study, the rotated copula at 90 and 270 degrees are not considered because of the Kendall's tau values corresponding to each dependence structure of Ps and Im for each stations are positive (Table 5(a)). Considering the trend forms for parameter, almost 100 kinds of copula models are considered. In particular, the BB1, BB6, BB7 and BB8 Copula parameter are set in different parameter intervals of the corresponding copula because of numerically instabilities for large parameters in the R package CDVine (Brechmann and Schepsmeier, 2013). Take the BB1 copula as an example, the parameter interval for two parameters are $(0,\infty)$ and $[1,\infty)$ as usual while in the CDVine package the parameter interval would be $(0,7)$ and $[1,7]$. As a result, it is necessary to add constraint functions (shown in the supplement).

Line(s) 203–205. Authors. The Gaussian copula was not used in this study because of its deficiency in describing dependencies of extremes (Renard and Lang, 2007). Referee. The Authors are clearly considering the concept of Tail Dependence. Well, also the Frank family has no tail dependence, while the Clayton family only has lower tail dependence (possibly, of no interest here), the Gumbel family only has upper tail dependence, and the Student family has both lower and upper tail dependence (but they must be equal, and, most of all, they both must exist at the same time!). There are more suitable families of copulas for modeling extremes: again, the ones used by the Authors are simply those provided by Matlab, as (unfortunately, too) many practitioners do, preventing a reliable/valuable investigation and modeling of the phenomenon of interest. Response: As shown in the former response, we have enriched the kinds of candidate copulas.

Line(s) 208, Eq. (5). Again, as above, you must justify the assumptions/relations shown in this equation. Why should the copula parameter change according to Eq. (5)? Did you carry out any investigation? What are the p-Values? Response: We implemented the goodness of fit test for copulas based on Rosenblatt's transformation (Rosenblatt, 1952).

Line(s) 212–214. Authors. The Corrected Akaike Information Criterion (AICc; Hurvich and Tsai, 1989) was employed to make a goodness-of-fit... Referee. NO.This a typical fatal error of practitioners. The AIC (corrected or not) is only a selection criterion, not a GoF procedure. You must first show that a copula is statistically admissible, e.g. via suitable Monte Carlo Cramer-von Mises or Kolmogorov-Smirnov tests, as in the R package"copula". Then, and only then, you may compare (only) the admissible copulas (if any) and select the "best" one according to some suitable criterion (e.g., the AICc, the BIC, the NLL, etc...). Response: We implemented the goodness of fit test for copulas based on Rosenblatt's transformation which can also be used to implement GOF tests for nonstationary copula.

Line(s) 214–216. Authors. Obviously, the presence of nonstationarity in the copula parameter was determined by comparison of the AICc value. Referee. This sentence is obscure. Are you saying that, since the non-stationary model performs better, then the phenomenon is non-stationary? If so, this makes no statistical and philosophical sense. It looks like you are using your models to "decide" how the real world should work: this is contrary to every scientific principle. This work is also bugged from an epistemological perspective. Response: We make the LR tests which is a statistical test to check whether the trend in the parameter. If value of the LR tests is smaller than 5%, it recommends the trend existed in the parameter at 5% significance level. And then AICc criterion is used as model selection criterion.

Line(s) 217–ff., Sec. 2.3. Authors. "2.3. Joint return period and risk analysis based on KEN's and AND's methods" Referee. Multivariate failure probabilities have been well mathematically formalized in Salvadori et al. (2016), by originally defining and exploiting suitable Hazard Scenarios and copulas' relations. The Authors must take this work into serious account, and mention it. Response: we have made multivariate hazard assessment based on the Average Annual Reliability (AAR) in the revised manuscript. And we also added the reference Salvadori et al. (2016) when we wanted to use the joint exceedance probability under AND scenario.

Line(s) 241, Eq. (7). See the more general approach and discussion in (Salvadori et al., 2016, Eq.s (33)-(35)). Response: we have made multivariate hazard assessment based on the Average Annual Reliability (AAR) in the revised manuscript. So this formula has been deleted.

Line(s) 262, Eq. (11). Why in Eq. (11) the parameters of the marginals $F_X$, $F_Y$, used as arguments in the copula C, do not vary with time? Response: it is an error of formula definition. We have taken this suggestion.

Line(s) 268–269. Authors. The most likely event at the $T_0$-year level can be calculated as (Graler et al., 2013)... Referee. NO. The Most Likely technique was first introduced in Salvadori et al. (2011): always give credits to whom deserve credits. In addition, it is not the only possible one, as shown in the same paper (viz., the Component-wise Excess method). Moreover, further approaches are outlined in Corbella and Stretch (2012) and Salvadori et al. (2014). Why was the Most Likely approach chosen in this work? Response: We have added the suggested reference to the manuscript. In order to simplify the process of generating the extreme rainfall quantiles at each ARR level, the most-likely technique was implemented by choosing a certain quantile pair which has the largest joint probability than other combinations at the same level (Salvadori et al., 2011). And we proposed the algorithm to capture the numeric solution for the bivariate quantiles (shown in 332-349).

Line(s) 289, Eq. (17). In Eq. (17), why is the modulus used? Obviously $\Delta R$ will always be positive. And even in this latter case, there is no quantification of any "scale" on which $\Delta R$ should be evaluated (when is it large? when is it small?). Such a number

tells nothing to me. Response: we deleted the formula because of adopting the ARR-based quantile estimation from hydrologic design.

Line(s) 330–331. Authors. As shown in Figure 3, concurrences of univariate and bivariate trends, the nonstationarities in rainfall extremes can be detected at several stations... Referee. This is simply because you use a 10% critical $\alpha$-level, entailing a large probability of rejecting the Null Hypothesis of non-stationarity. For instance, at a standard 5% level, no one of the Univariate and Multivariate MK tests would fail, only two (at most three) out of 12 of the Univariate Pettitt tests would fail, and only one out of 6 of the Multivariate Pettitt tests would fail. In turn, the conclusions of the Authors are definitely questionable: in my opinion, in general, there is no clear statistical evidence of non-stationarity (not to say if the standard 1% level were used, for in this case stationarity would be fully supported). Apparently, the Authors manipulate statistics according to their convenience, in order to show what they want to show. Response: We considered more extreme series from more stations in the study area. Because of the insignificant trend denoted by reviewer #1 in the original manuscript, we changed the original 95-th percentile threshold for Ps to 0.90. After above modification of extreme values, the significant trend and change point at 5% significance level could be detected in 3 stations by nonparametric tests. And there is no station which can exhibit concurrences of univariate and bivariate trends, the nonstationarities according to the results in revised manuscript. The significance level of 5% is just the minimum standard. And it is not right and statistical objective to manipulate statistics according to our convenience, in order to show what we want to show. We have deeply realized the seriousness of the problem and corrected it in revised manuscript.

Line(s) 353–354. Authors. The location parameter ($\mu$) and scale parameter ($\sigma$) are regarded as time variant, while the shape parameter $\kappa$ is time invariant... Referee. As above, it is a dream to try and model time-variation of extremes using a constant shape parameter: it is the only one that matters in these kind of analyses. In addition, why should the other parameters vary according to Eq.s (3)–(5)? Simply because the

same relations were used in other papers (again, without justification)? This paper has no scientific objective grounds. Response: as shown above, we have incorporate the potential trends into shape parameter. And K-S based GOF tests were conducted to verify its rationality.

Line(s) 356–358. Authors. Despite the exception of Im for station 4, the shape parameter $\kappa$ for most fitted models was in the interval of [-0.3,0.3]... Referee. Tables 3 provide little statistical information, for no suitable confidence intervals are shown: this may have considerable consequences regarding the conclusions drawn by the Authors in later sections. In fact, they did not carry out any Monte Carlo analysis, and hence their results do not take into account the estimates' uncertainties (as if the Authors were stating the absolute Truth). To be clear, no confidence bands are plotted in later figures. This is not a scientific way of proceeding: the Authors must provide plots such as the ones shown in Salvadori et al. (2018), which may give an idea of the uncertainties at play (which may be huge, especially when a GEV is used, and may completely change the interpretation of the results, as I suspect). In addition, as above, some of the fitted values of the shape parameter would imply that the corresponding GEV is Upper Bounded, entailing that the corresponding variable cannot be an Extreme one. Furthermore, the fact that the range of the shape parameter is "in accordance with previous studies" is not significant and relevant at all (also given the fact that the range is quite large). Response: The parametric bootstrap (Efron and Tibshirani, 1993) method was implemented for parameter estimation and quantile estimation based on ARR level. It can provide a confidence bands (90% in this study) of parameter or the quantile estimation.

Line(s) 359–360. Authors. The best fitted model was selected by performing the minimum DIC criterion combined with the Bayes factor (BF) test. Referee. Again, you did not show that it is an admissible one! This work has no statistical bases. Line(s) 380–382. Authors. Table 4(a)-(b) illustrates the results of best fitted copula, based on the minimum AICc and maximum logllikelihood value (LL). Referee. Again, AIC and

LL are not GoF criteria: the chosen models can be non-admissible! This work has no statistical bases. Response: These two comments are replied by taking K-S based GOF tests for marginal distribution and A-D with Rosenblatt's transformation for copula models.

Line(s) 433–435. Authors. Although the copula model for station 5 was stationary, it was regarded as a nonstationary model because of the marginal nonstationary GEVns-2 model for Ps or Im, which existed at other stations. Referee. This makes no sense. The Authors do not understand the basic fact that the dependence structure is independent of the marginals (as stated by Sklar's representation Theorem): even if the marginals are non-stationary, the copula may be stationary. The introduction of non-stationary copulas is arbitrary, without any justification: you cannot manipulate the results in this way! Response: we have deleted this nonsense statement as suggestion.

Line(s) 440–ff. Authors. Figure 5 shows isolines of Kendall return period and AND-based return period... Referee. Given the uncertainties mentioned above (not considered by the Authors), I strongly suspect that the interpretation of the results shown in Figure 5 could be quite different if suitable confidence bands were plotted. This work lacks of elementary statistical bases. Response: The parametric bootstrap (Efron and Tibshirani, 1993) method was implemented for parameter estimation and quantile estimation based on ARR level. It can provide a confidence bands (90% in this study) of parameter or the quantile estimation.

Line(s) 537–ff., Sec. 4.6. In the light of the objections given above, the "Further discussion" section (4.6) makes no sense. Response: we have modified the further discussion after we revised the manuscript as reviewer 2 suggested.

Please also note the supplement to this comment: https://www.hydrol-earth-syst-sci-discuss.net/hess-2019-358/hess-2019-358-AC2-supplement.pdf

[Figure]

**Supplement:**

**Reviewer #2:**

**GENERAL COMMENTS.**

In my opinion, essentially this paper only adds "noise" to the existing Literature: the techniques used have already been published in other works, the only novelty (clearly, not a methodological one) could be the case study, but any new case study must represent a newness over previous ones (otherwise it would be a replica). Most importantly, the work is in general statistically weak, and affected and flawed by fatal errors: the conclusions of the Authors may not be supported by the analyses they carried out. Apparently, the Authors (incorrectly) interpret the results according to their convenience, in order to prove what they want to prove, as shown below. In addition, referencing is often imprecise and/or improper and/or missing: always give credits to whom deserve credits. My recommendation is: REJECTION.

**Response to Reviewer 2:**
Great appreciation for this comment!

We have taken the review's suggestions into consideration. Firstly, a nonstationary and stationary GOF tests were implemented for marginal distribution and copula models. In addition, we implement the log likelihood ratio (LR) statistics to check the trend in parameters of distribution, which is more rigorous trend detection method than Mann-Kendall tests (Coles, 2001). We considered more extreme series from more stations in the study area. Because of the insignificant trend denoted by reviewer #1 in the original

```

manuscript, we changed the original 95-th percentile threshold for *Ps* to 0.90. After above modification of extreme values, the significant trend and change point at 5% significance level could be detected in 3 stations by nonparametric tests. Here, we proposed the LR tests to detect the trend in parameter after the Mann-Kendall tests. Some interesting findings are captured. A detailed point-by-point reply has been made as follows.

**SPECIFIC COMMENTS.**

**Line(s) 49–54.**

**Authors.** Copulas, a useful tool for modelling the structure of dependence between hydrological variables regardless of the types of marginal distributions, have been widely used for multivariate frequency analysis + references...

**Referee.** Historically, the paper by Salvadori and De Michele (2004) was the first one to deal with (copula) multivariate frequency analysis—later works are copies or small variants: this paper is not cited. Please, always give credits to whom deserve credits.

**Response:** We have added this reference to the revised manuscript.

**Line(s) 75–ff.**

**Authors.** There are three kinds of joint return period methods...

**Referee.** NO. In Literature there are, at least, four kinds of joint return periods. The references given are incorrect. In Salvadori and De Michele (2004) the OR, AND and Kendall cases were first introduced. In Salvadori et al. (2013) a further survival-Kendall

```
```

approach (not mentioned by the Authors) was outlined. Referencing is often imprecise, almost random: for instance, why citing Jiang et al. (2015) here? It has nothing to do with the original formalization of the four return periods mentioned above. Incidentally, the reference "Salvadori and Michele, 2010" is "Salvadori and De Michele, 2010" (it seems that the Authors wrote the references by hand, instead of using some suitable software...).

**Response:** we have modified the above statement about the joint return period. And give credits to whom deserve credits at right places. Lines 315-325 in revised manuscript.

**Line(s) 95–97.**

**Authors.** Note that following the idea of Rootzén and Katz (2013) we regard the term hydrological risk as the possibility of a certain extreme event occurring and not as a quantification of expected losses.

**Referee.** Then, probabilistically and statistically speaking (and hydrologically as well!), you should better use the term "hazard" instead of "risk".

**Response:** We take the advice by reviewer. And we modified this statement. And also we have changed the title to "Time-varying copula and average annual reliability-based nonstationary hazard assessment of extreme rainfall events."

**Line(s) 126.**

```

**Authors.** Detailed information about copulas can be found in Nelson (2007).

**Referee.** NO. It is Nelsen (2006), not Nelson. For an engineering approach, you may also cite Salvadori et al. (2007). As a strong suggestion, the Authors should carefully check the correctness of all the references (it is easy to do it on the Internet), and add the missing ones.

**Response:** we have revised the quotation of this reference. Add Salvadori et al. (2007) to revised manuscript as follows: "Detailed information of theoretical derivation about copulas can be found in Nelsen (2006). For the practical guidelines from hydrological point of view, it is recommended to refer to Salvadori et al. (2007)."

Line(s) 138–139.

**Authors.** ... $\theta_C^t$ is the dynamic copula parameter which is a linear function of time.

**Referee.** The Authors must justify this choice. Please do not reply that "the model was taken from this or that paper": it is not a scientific reason, for a model must be validated on the available data. Also, the results of suitable Goodness-of-Fit statistical tests must be shown.

**Response:** $\theta_C^t$ is set as follows in the revised manuscript:

$$\theta_C^t = \begin{cases} \text{constant} \\ \theta_0 + \theta_1 t \\ \theta_0 + \theta_1 t + \theta_2 t^2 \end{cases}$$

As shown above, both the linear and quadratic models are assumed as the possible function relation between parameter and time, which would consider the linear and nonlinear trend for parameter. We implemented the goodness of fit test for copulas based on Rosenblatt's transformation (Rosenblatt, 1952).

```

Lines 271-283: "Rosenblatt's transformation (RT) of the time-varying marginal distribution $U = F_X(x|\theta_X^t)$ and $V = F_Y(y|\theta_Y^t)$ of bivariate copula can be defined as follows:

$$\begin{cases} RT_1 = u = F_X(x|\theta_X^t) \\ RT_2 = C(v|u,|\theta_C^t) = C[F_Y(y|\theta_Y^t)|F_X(x|\theta_X^t),|\theta_C^t] \end{cases} \quad (5)$$

Where $C(u|v,|\theta_C^t)$ is just the conditional distribution function of v given by u=$F_X(x|\theta_X^t)$.

According to Rosenblatt, the random variable $RT_1$ and $RT_2$ is independent and uniformly in the interval [0,1]. In order to check this assumption, it is convenient to calculate:

$$S_i = [\Phi^{-1}(F_X(x_i|\theta_X^t))]^2 + [\Phi^{-1}(C[F_Y(y_i|\theta_Y^t)|F_X(x_i|\theta_X^t),|\theta_C^t])]^2 \quad i = 1,\dots,n \quad (6)$$

Here, n is just the length of the data; and in this study we let time t be equal to 1,2,…,n. If the random sample $\{S_i\}$ comes from a $\chi_2^2$ distribution, it can accept the NULL hypothesis ($\mathcal{H}_0$: dependence structure between $X$ and $Y$ obey the time-varying copula $C(u,v|\theta_C^t)$). Then the Anderson–Darling goodness-of-fit test based on RT ($AD_{RT}$) should be used for the above assumption." According to the analysis results, the assumed linear and nonlinear trends of copula parameter can pass the GOF tests based on Rosenblatt's transformation.

**Line(s) 143–144.**

**Authors.** It is however possible that the nonstationary behavior may exist in both the marginal and joint distribution function.

**Referee.** Such an issue was already clearly pointed out and discussed in Salvadori et al.

```

(2018), where a similar case study was investigated, and a thorough statistical analysis was carried out. The Authors must mention this fact, and follow the (proper statistical) guidelines outlined in that paper.

**Response:** We have revised this statement. And we mentioned this fact as:

Lines 134-160. "According to Vezzoli et al. (2017) and Salvadori et al. (2018), a comprehensive statistical analysis which can check the presence of trend and change point should be carried out before we incorporate the nonstationarity into the multivariate hazard assessment. Following the study of Salvadori et al. (2018), the non-parametric change-point statistical tests were implemented to check that whether the marginal or joint distributions are sensitive to changes. These tests can be manipulated in the R package npcp (Kojadinovic, 2017)."

**Line(s) 158-Figure 1.**

The flow-chart shown in Figure 1 provides wrong indications (see also later comments). In fact, the Authors confuse GoF tests with selection criteria. The flow-chart must be rewritten.

**Response:** we have revised the flowchart of this study **(Figure 1)** as follows:

```

[Figure]

**Figure 1.** Flowchart of this study

**Line(s) 161–164**.

**Authors.** In this part, the Generalized Extreme Value (GEV) distribution was used to...

(Cheng and AghaKouchak, 2014).

**Referee.** This reference makes little sense: the features of the GEV have already been stated and described since decades in other (seminal) works. Please use proper references.

**Response:** In revised manuscript, five kinds of marginal distribution have been used as candidate distribution. For space limitations, we have deleted the above statement of GEV features.

```

**Line(s) 166–ff.**

**Authors.** The GEV distribution consists of three control parameters...

**Referee.** The GEV distribution is well known to hydrologists, there is no need to tell again a story that everybody knows.

**Response:** we deleted this statement in revised manuscript.

**Line(s) 176–179.**

**Authors.** In this study, two kinds of nonstationary GEV models (GEVns-1 and GEVns-2) are developed with the shape parameter being constant. It should be emphasized that modelling the time variance in shape parameter needs long-term observations, which are often not available in practice (Cheng et al., 2014).

**Referee.** I recently rejected a paper very similar to the present one, where the GEV shape parameter was kept constant. The shape parameter is the most important one, for it rules the generation of extremes. The assumption adopted is definitely questionable: what (extreme) climate change could you really hope to model with a constant shape parameter? Practically, you are trying to model climate changes where the statistics of the extremes do not change with time: it makes little sense. In addition (see also later comments), some estimates of the GEV shape parameter are positive and other negative

```

(Table 3). This entails that, in some cases, the corresponding GEV law is upper-bounded, i.e. unable to model an extreme behavior: this is a well known feature of the GEV. I agree that the GEV is the right distribution to be used in your analysis (Block Maxima), but the question is: how can you claim that the phenomenon you are modeling is an extreme one when upper-bounded GEV's are involved? The statistical results seem to tell another story...

**Response:** Thanks a lot for this suggestion. We have consider the trend in shape parameter. For all the three parameter (location, scale, shape), three kinds of forms are considered. Here we take the location parameter as an example:

$$\mu_t = \begin{cases} \text{constant} \\ \mu_0 + \mu_1 t \\ \mu_0 + \mu_1 t + \mu_2 t^2 \end{cases} \tag{2}$$

As stated by reviewer, the GEV can be defined as follows:

$$G(x) = \exp\left[-\left\{1 + \kappa\left(\frac{x-\mu}{\sigma}\right)\right\}_+^{-\frac{1}{\kappa}}\right] \tag{3}$$

where $Z_+ = \max\{z, 0\}$, $\sigma > 0$ and $\mu$ & $\kappa \in (-\infty, \infty)$. When $\kappa < 0$, it recommends the upper-bounded GEV distribution. We have checked the shape parameter of the best fitted GEV models is always positive (station 2,3,6 in **Table 4(a)-(b)**).

```

**S1(a).** Patterns of the time-varying models for 3-parameter marginal distribution (GEV in this study)

| Model | $\mu$ | $\sigma$ | $\kappa$ |
|---|---|---|---|
| GEV0 | constant | constant | constant |
| GEV1 | $\mu = \mu_0 + \mu_1 t$ | constant | constant |
| GEV2 | $\mu = \mu_0 + \mu_1 t + \mu_2 t^2$ | constant | constant |
| GEV3 | constant | $ln\sigma = \sigma_0 + \sigma_1 t$ | constant |
| GEV4 | constant | $ln\sigma = \sigma_0 + \sigma_1 t + \sigma_2 t^2$ | constant |
| GEV5 | constant | constant | $\kappa = \kappa_0 + \kappa_1 t$ |
| GEV6 | constant | constant | $\kappa = \kappa_0 + \kappa_1 t + \kappa_2 t^2$ |
| GEV7 | $\mu = \mu_0 + \mu_1 t$ | $ln\sigma = \sigma_0 + \sigma_1 t$ | constant |
| GEV8 | $\mu = \mu_0 + \mu_1 t$ | constant | $\kappa = \kappa_0 + \kappa_1 t$ |
| GEV9 | constant | $ln\sigma = \sigma_0 + \sigma_1 t$ | $\kappa = \kappa_0 + \kappa_1 t$ |
| GEV10 | $\mu = \mu_0 + \mu_1 t + \mu_2 t^2$ | $ln\sigma = \sigma_0 + \sigma_1 t + \sigma_2 t^2$ | constant |
| GEV11 | constant | $ln\sigma = \sigma_0 + \sigma_1 t + \sigma_2 t^2$ | $\kappa = \kappa_0 + \kappa_1 t + \kappa_2 t^2$ |
| GEV12 | $\mu = \mu_0 + \mu_1 t + \mu_2 t^2$ | constant | $\kappa = \kappa_0 + \kappa_1 t + \kappa_2 t^2$ |
| GEV13 | $\mu = \mu_0 + \mu_1 t$ | $ln\sigma = \sigma_0 + \sigma_1 t + \sigma_2 t^2$ | constant |
| GEV14 | constant | $ln\sigma = \sigma_0 + \sigma_1 t$ | $\kappa = \kappa_0 + \kappa_1 t + \kappa_2 t^2$ |
| GEV15 | $\mu = \mu_0 + \mu_1 t$ | constant | $\kappa = \kappa_0 + \kappa_1 t + \kappa_2 t^2$ |
| GEV16 | $\mu = \mu_0 + \mu_1 t + \mu_2 t^2$ | $ln\sigma = \sigma_0 + \sigma_1 t$ | constant |
| GEV17 | constant | $ln\sigma = \sigma_0 + \sigma_1 t + \sigma_2 t^2$ | $\kappa = \kappa_0 + \kappa_1 t$ |
| GEV18 | $\mu = \mu_0 + \mu_1 t + \mu_2 t^2$ | constant | $\kappa = \kappa_0 + \kappa_1 t$ |
| GEV19 | $\mu = \mu_0 + \mu_1 t + \mu_2 t^2$ | $ln\sigma = \sigma_0 + \sigma_1 t$ | $\kappa = \kappa_0 + \kappa_1 t + \kappa_2 t^2$ |
| GEV20 | $\mu = \mu_0 + \mu_1 t + \mu_2 t^2$ | $ln\sigma = \sigma_0 + \sigma_1 t + \sigma_2 t^2$ | $\kappa = \kappa_0 + \kappa_1 t$ |
| GEV21 | $\mu = \mu_0 + \mu_1 t$ | $ln\sigma = \sigma_0 + \sigma_1 t + \sigma_2 t^2$ | $\kappa = \kappa_0 + \kappa_1 t + \kappa_2 t^2$ |
| GEV22 | $\mu = \mu_0 + \mu_1 t$ | $ln\sigma = \sigma_0 + \sigma_1 t$ | $\kappa = \kappa_0 + \kappa_1 t + \kappa_2 t^2$ |
| GEV23 | $\mu = \mu_0 + \mu_1 t + \mu_2 t^2$ | $ln\sigma = \sigma_0 + \sigma_1 t$ | $\kappa = \kappa_0 + \kappa_1 t$ |
| GEV24 | $\mu = \mu_0 + \mu_1 t$ | $ln\sigma = \sigma_0 + \sigma_1 t + \sigma_2 t^2$ | $\kappa = \kappa_0 + \kappa_1 t$ |
| GEV25 | $\mu = \mu_0 + \mu_1 t$ | $ln\sigma = \sigma_0 + \sigma_1 t$ | $\kappa = \kappa_0 + \kappa_1 t$ |
| GEV26 | $\mu = \mu_0 + \mu_1 t + \mu_2 t^2$ | $ln\sigma = \sigma_0 + \sigma_1 t + \sigma_2 t^2$ | $\kappa = \kappa_0 + \kappa_1 t + \kappa_2 t^2$ |

```

**S1(b).** Patterns of the time-varying models for 3-parameter marginal distribution (PIII in this study)

| Model | $\mu$ | $\sigma$ | $\kappa$ |
|---|---|---|---|
| PIII0 | constant | constant | constant |
| PIII1 | $\mu = M * \sin(\mu_0 + \mu_1 t)$ | constant | constant |
| PIII2 | $\mu = M * \sin(\mu_0 + \mu_1 t + \mu_2 t^2)$ | constant | constant |
| PIII3 | constant | $\sigma = \sigma_0 + \sigma_1 t$ | constant |
| PIII4 | constant | $\sigma = \sigma_0 + \sigma_1 t + \sigma_2 t^2$ | constant |
| PIII5 | constant | constant | $ln\kappa = \kappa_0 + \kappa_1 t$ |
| PIII6 | constant | constant | $ln\kappa = \kappa_0 + \kappa_1 t + \kappa_2 t^2$ |
| PIII7 | $\mu = M * \sin(\mu_0 + \mu_1 t)$ | $\sigma = \sigma_0 + \sigma_1 t$ | constant |
| PIII8 | $\mu = M * \sin(\mu_0 + \mu_1 t)$ | constant | $ln\kappa = \kappa_0 + \kappa_1 t$ |
| PIII9 | constant | $\sigma = \sigma_0 + \sigma_1 t$ | $ln\kappa = \kappa_0 + \kappa_1 t$ |
| PIII10 | $\mu = M * \sin(\mu_0 + \mu_1 t + \mu_2 t^2)$ | $\sigma = \sigma_0 + \sigma_1 t + \sigma_2 t^2$ | constant |
| PIII11 | constant | $\sigma = \sigma_0 + \sigma_1 t + \sigma_2 t^2$ | $ln\kappa = \kappa_0 + \kappa_1 t + \kappa_2 t^2$ |
| PIII12 | $\mu = M * sin(\mu_0 + \mu_1 t + \mu_2 t^2)$ | constant | $ln\kappa = \kappa_0 + \kappa_1 t + \kappa_2 t^2$ |
| PIII13 | $\mu = M * \sin(\mu_0 + \mu_1 t)$ | $\sigma = \sigma_0 + \sigma_1 t + \sigma_2 t^2$ | constant |
| PIII14 | constant | $\sigma = \sigma_0 + \sigma_1 t$ | $ln\kappa = \kappa_0 + \kappa_1 t + \kappa_2 t^2$ |
| PIII15 | $\mu = M * \sin(\mu_0 + \mu_1 t)$ | constant | $ln\kappa = \kappa_0 + \kappa_1 t + \kappa_2 t^2$ |
| PIII16 | $\mu = M * \sin(\mu_0 + \mu_1 t + \mu_2 t^2)$ | $\sigma = \sigma_0 + \sigma_1 t$ | constant |
| PIII17 | constant | $\sigma = \sigma_0 + \sigma_1 t + \sigma_2 t^2$ | $ln\kappa = \kappa_0 + \kappa_1 t$ |
| PIII18 | $\mu = M * sin(\mu_0 + \mu_1 t + \mu_2 t^2)$ | constant | $ln\kappa = \kappa_0 + \kappa_1 t$ |
| PIII19 | $\mu = M * sin(\mu_0 + \mu_1 t + \mu_2 t^2)$ | $\sigma = \sigma_0 + \sigma_1 t$ | $ln\kappa = \kappa_0 + \kappa_1 t + \kappa_2 t^2$ |
| PIII20 | $\mu = M * sin(\mu_0 + \mu_1 t + \mu_2 t^2)$ | $\sigma = \sigma_0 + \sigma_1 t + \sigma_2 t^2$ | $ln\kappa = \kappa_0 + \kappa_1 t$ |
| PIII21 | $\mu = M * sin(\mu_0 + \mu_1 t)$ | $\sigma = \sigma_0 + \sigma_1 t + \sigma_2 t^2$ | $ln\kappa = \kappa_0 + \kappa_1 t + \kappa_2 t^2$ |
| PIII22 | $\mu = M * sin(\mu_0 + \mu_1 t)$ | $\sigma = \sigma_0 + \sigma_1 t$ | $ln\kappa = \kappa_0 + \kappa_1 t + \kappa_2 t^2$ |
| PIII23 | $\mu = M * \sin(\mu_0 + \mu_1 t + \mu_2 t^2)$ | $\sigma = \sigma_0 + \sigma_1 t$ | $ln\kappa = \kappa_0 + \kappa_1 t$ |
| PIII24 | $\mu = M * \sin(\mu_0 + \mu_1 t)$ | $\sigma = \sigma_0 + \sigma_1 t + \sigma_2 t^2$ | $ln\kappa = \kappa_0 + \kappa_1 t$ |
| PIII25 | $\mu = M * \sin(\mu_0 + \mu_1 t)$ | $\sigma = \sigma_0 + \sigma_1 t$ | $ln\kappa = \kappa_0 + \kappa_1 t$ |
| PIII26 | $\mu = M * \sin(\mu_0 + \mu_1 t + \mu_2 t^2)$ | $\sigma = \sigma_0 + \sigma_1 t + \sigma_2 t^2$ | $ln\kappa = \kappa_0 + \kappa_1 t + \kappa_2 t^2$ |

Note: M represents the minimum value of the observed time series.

```
```

**S1(c).** Patterns of the time-varying models for 2-parameter marginal distribution containing scale and shape parameter (Weibull, Gamma in this study).

| Model | $\sigma$ | $\kappa$ |
|---|---|---|
| WE0/GA0 | constant | constant |
| WE1/GA1 | $ln\sigma = \sigma_0 + \sigma_1 t$ | constant |
| WE2/GA2 | $ln\sigma = \sigma_0 + \sigma_1 t + \sigma_2 t^2$ | constant |
| WE3/GA3 | constant | $ln\kappa = \kappa_0 + \kappa_1 t$ |
| WE4/GA4 | constant | $ln\kappa = \kappa_0 + \kappa_1 t + \kappa_2 t^2$ |
| WE5/GA5 | $ln\sigma = \sigma_0 + \sigma_1 t$ | $ln\kappa = \kappa_0 + \kappa_1 t$ |
| WE6/GA6 | $ln\sigma = \sigma_0 + \sigma_1 t$ | $ln\kappa = \kappa_0 + \kappa_1 t + \kappa_2 t^2$ |
| WE7/GA7 | $ln\sigma = \sigma_0 + \sigma_1 t + \sigma_2 t^2$ | $ln\kappa = \kappa_0 + \kappa_1 t$ |
| WE8/GA8 | $ln\sigma = \sigma_0 + \sigma_1 t + \sigma_2 t^2$ | $ln\kappa = \kappa_0 + \kappa_1 t + \kappa_2 t^2$ |

**S1(d).** Patterns of the time-varying models for 2-parameter marginal distribution containing location and scale parameter (Lognormal function in this study).

| Model | $\mu$ | $\sigma$ |
|---|---|---|
| LOGN0 | constant | constant |
| LOGN1 | $\mu = \mu_0 + \mu_1 t$ | constant |
| LOGN2 | $\mu = \mu_0 + \mu_1 t + \mu_2 t^2$ | constant |
| LOGN3 | constant | $ln\sigma = \sigma_0 + \sigma_1 t$ |
| LOGN4 | constant | $ln\sigma = \sigma_0 + \sigma_1 t + \sigma_2 t^2$ |
| LOGN5 | $\mu = \mu_0 + \mu_1 t$ | $ln\sigma = \sigma_0 + \sigma_1 t$ |
| LOGN6 | $\mu = \mu_0 + \mu_1 t$ | $ln\sigma = \sigma_0 + \sigma_1 t + \sigma_2 t^2$ |
| LOGN7 | $\mu = \mu_0 + \mu_1 t + \mu_2 t^2$ | $ln\sigma = \sigma_0 + \sigma_1 t$ |
| LOGN8 | $\mu = \mu_0 + \mu_1 t + \mu_2 t^2$ | $n\sigma = \sigma_0 + \sigma_1 t + \sigma_2 t^2$ |

```
```

**Line(s) 184–186, Eq.s (3)–(4).**

You must justify the assumptions/relations implicit in these equations. Why should the position and scale parameters change according to Eq.s(3)–(4)? Did you carry out any valuable/reliable fit? What are the p-Values? And, again, why should the shape parameter be constant instead? Incidentally, these are the same equations used in the paper I recently rejected.

**Response:** We have taken this advice by considering trend in shape parameters as shown in the former response. Also we have taken the nonstationary K-S tests on the assumed linear and nonlinear trend (**Table 4(a)-(b)**).

**Line(s) 191–194.**

**Authors.** Simultaneously, the Deviance Information Criterion (DIC) and Bayes factors (BF) for different stationary and nonstationary models were calculated to select the best fitted marginal model. The minimum DIC value yielded the best performance, while BF smaller than 1 indicated the best fitting.

**Referee.** This is a typical fatal error of practitioners. These are only selection criteria, not Goodness-of-Fit tests. You must first use (non-stationary) GoF tests to check whether a model is admissible! Otherwise, without first checking the models via suitable GoF tests, you may end up choosing non-admissible ones. This work has no statistical bases.

**Response:** We have taken this advice by investigating the univariate and bivariate GOF tests as presented in (lines 225-233 and lines 267- 283).

```

**Line(s) 191–194.**

**Authors.** In multivariate hydrological frequency analysis, two kinds of copulas, named elliptical and Archimedean copulas are widely used in hydrological applications.

**Referee.** So what? The fact that these copulas were used in other works is not, and cannot be, a scientific justification. This is the usual approach of practitioners that use the copulas provided by Matlab. Given my experience, I do not really think that Nature (especially considering the generation of Extremes) gets stick to just these dependence structures—see also later comments. And, worst of all, you did not even check these copula models via suitable multivariate GoF tests (which are available in Literature, and some certified software is even for free—see below): this work has no statistical bases.

**Response:** In the revised manuscript, we firstly enrich the kinds of candidate copulas: five kinds of 1-parameter copula (Joe, Frank, Gumbel, Clayton and Gaussian) and five kinds of 2-parameter copulas (Clayton-Gumbel (BB1), Student t, Joe-Gumbel (BB6), Joe-Clayton (BB7) and Joe-Frank (BB8)). In addition, each copula can be rotated at 90, 180 (Survival Copula), 270 degrees. For this study, the rotated copula at 90 and 270 degrees are not considered because of the Kendall's tau values corresponding to each dependence structure of *Ps* and *Im* for each stations are positive (**Table 5(a)**). Considering the trend forms for parameter, almost 100 kinds of copula models are considered. In particular, the BB1, BB6, BB7 and BB8 Copula parameter are set in different parameter intervals of the corresponding copula because of numerically instabilities for large parameters in the R package CDVine (Brechmann and Schepsmeier, 2013). Take the BB1 copula as an example, the parameter interval for two parameters are $(0, \infty)$ and $[1, \infty)$ as usual while in the

```

CDVine package the parameter interval would be (0,7) and [1,7]. As a result, it is necessary to add

constraint functions $(3.5 + 3.5\sin(\theta_0 + \theta_1 t)$ for $\theta_C^t$ and $4 + 3\sin(\beta_0 + \beta_1 t)$ for $\beta_C^t)$.

**S2(a)** Patterns of the time-varying models for 1-parameter bivariate copula model considering different parameter ranges for different copulas

| Model | $\theta$ |
|---|---|
| GAU0 | constant |
| GAU1 | $\theta = \sin(\theta_0 + \theta_1 t)$ |
| GAU2 | $\theta = \sin(\theta_0 + \theta_1 t + \theta_2 t^2)$ |
| GU0/J0/SGU0/SJ0 | constant |
| GU1/J1/SGU1/SJ1 | $\theta = 1 + \exp(\theta_0 + \theta_1 t)$ |
| GU2/J2/SGU2/SJ2 | $\theta = 1 + \exp(\theta_0 + \theta_1 t + \theta_2 t^2)$ |
| CL0/SCL0 | constant |
| CL1/SCL1 | $\theta = \exp(\theta_0 + \theta_1 t)$ |
| CL2/SCL2 | $\theta = \exp(\theta_0 + \theta_1 t + \theta_2 t^2)$ |
| FR0 | constant |
| FR1 | $\theta = \theta_0 + \theta_1 t$ |
| FR2 | $\theta = \theta_0 + \theta_1 t + \theta_2 t^2$ |

GAU: Gaussian copula; GU: Gumbel copula; J: Joe copula; SGU: survival Gumbel copula; SJ: survival Joe copula; CL: Clayton copula; SCL: survival Clayton copula; FR: Frank copula;

**S2(b)** Patterns of the time-varying models for 2-parameter student t copula

| Model | $\theta$ | $\beta$ |
|---|---|---|
| ST0 | constant | constant |
| ST1 | $\theta = \sin(\theta_0 + \theta_1 t)$ | constant |
| ST2 | $\theta = \sin(\theta_0 + \theta_1 t + \theta_2 t^2)$ | constant |
| ST3 | constant | $\beta = 2 + \exp(\beta_0 + \beta_1 t)$ |
| ST4 | constant | $\beta = 2 + \exp(\beta_0 + \beta_1 t + \beta_2 t^2)$ |
| ST5 | $\theta = \sin(\theta_0 + \theta_1 t)$ | $\beta = 2 + \exp(\beta_0 + \beta_1 t)$ |
| ST6 | $\theta = \sin(\theta_0 + \theta_1 t)$ | $\beta = 2 + \exp(\beta_0 + \beta_1 t + \beta_2 t^2)$ |
| ST7 | $\theta = \sin(\theta_0 + \theta_1 t + \theta_2 t^2)$ | $\beta = 2 + \exp(\beta_0 + \beta_1 t)$ |
| ST8 | $\theta = \sin(\theta_0 + \theta_1 t + \theta_2 t^2)$ | $\beta = 2 + \exp(\beta_0 + \beta_1 t + \beta_2 t^2)$ |

CG: Clayton-Gumbel copula; SCG: survival Clayton-Gumbel copula;

```
```

**S2(c)** Patterns of the time-varying models for 2-parameter (survival) Clayton-Gumbel copula (BB1 copula)

| Model | $\theta$ | $\beta$ |
|---|---|---|
| CG0/SCGO | constant | constant |
| CG1/SCG1 | $\theta = 3.5 + 3.5\sin(\theta_0 + \theta_1 t)$ | constant |
| CG2/SCG2 | $\theta = 3.5 + 3.5\sin(\theta_0 + \theta_1 t + \theta_2 t^2)$ | constant |
| CG3/SCG3 | constant | $\beta = 4 + 3sin(\beta_0 + \beta_1 t)$ |
| CG4/SCG4 | constant | $\beta = 4 + 3sin(\beta_0 + \beta_1 t + \beta_2 t^2)$ |
| CG5/SCG5 | $\theta = 3.5 + 3.5\sin(\theta_0 + \theta_1 t)$ | $\beta = 4 + 3sin(\beta_0 + \beta_1 t)$ |
| CG6/SCG6 | $\theta = 3.5 + 3.5\sin(\theta_0 + \theta_1 t)$ | $\beta = 4 + 3sin(\beta_0 + \beta_1 t + \beta_2 t^2)$ |
| CG7/SCG7 | $\theta = 3.5 + 3.5\sin(\theta_0 + \theta_1 t + \theta_2 t^2)$ | $\beta = 4 + 3sin(\beta_0 + \beta_1 t)$ |
| CG8/SCG8 | $\theta = 3.5 + 3.5\sin(\theta_0 + \theta_1 t + \theta_2 t^2)$ | $\beta = 4 + 3sin(\beta_0 + \beta_1 t + \beta_2 t^2)$ |

CG: Clayton-Gumbel copula; SCG: survival Clayton-Gumbel copula;

**S3(d).** Patterns of the time-varying models for 2-parameter (survival) Joe-Gumbel copula (BB6 copula)

| Model | $\theta$ | $\beta$ |
|---|---|---|
| JG0/SJGO | constant | constant |
| JG1/SJG1 | $\theta = 3.5 + 2.5sin(\theta_0 + \theta_1 t)$ | constant |
| JG2/SJG2 | $\theta = 3.5 + 2.5sin(\theta_0 + \theta_1 t + \theta_2 t^2)$ | constant |
| JG3/SJG3 | constant | $\beta = 4.5 + 3.5 * sin(\beta_0 + \beta_1 t)$ |
| JG4/SJG4 | constant | $\beta = 4.5 + 3.5 * sin(\beta_0 + \beta_1 t + \beta_2 t^2)$ |
| JG5/SJG5 | $\theta = 3.5 + 2.5sin(\theta_0 + \theta_1 t)$ | $\beta = 4.5 + 3.5 * sin(\beta_0 + \beta_1 t)$ |
| JG6/SJG6 | $\theta = 3.5 + 2.5sin(\theta_0 + \theta_1 t)$ | $\beta = 4.5 + 3.5 * sin(\beta_0 + \beta_1 t + \beta_2 t^2)$ |
| JG7/SJG7 | $\theta = 3.5 + 2.5sin(\theta_0 + \theta_1 t + \theta_2 t^2)$ | $\beta = 4.5 + 3.5 * sin(\beta_0 + \beta_1 t)$ |
| JG8/SJG8 | $\theta = 3.5 + 2.5sin(\theta_0 + \theta_1 t + \theta_2 t^2)$ | $\beta = 4.5 + 3.5 * sin(\beta_0 + \beta_1 t + \beta_2 t^2)$ |

JG: Joe-Gumbel copula; SJG: survival Joe-Gumbel copula;

**S3(e).** Patterns of the time-varying models for 2-parameter (survival) Joe-Clayton copula (BB7copula)

| Model | $\theta$ | $\beta$ |
|---|---|---|
| JC0/SJC0 | constant | constant |
| JC1/SJC1 | $\theta = 3.5 + 2.5sin(\theta_0 + \theta_1 t)$ | constant |
| JC2/SJC2 | $\theta = 3.5 + 2.5sin(\theta_0 + \theta_1 t + \theta_2 t^2)$ | constant |
| JC3/SJC3 | constant | $\beta = 37.5 + 37.5sin(\beta_0 + \beta_1 t)$ |
| JC4/SJC4 | constant | $\beta = 37.5 + 37.5sin(\beta_0 + \beta_1 t + \beta_2 t^2)$ |
| JC5/SJC5 | $\theta = 3.5 + 2.5sin(\theta_0 + \theta_1 t)$ | $\beta = 37.5 + 37.5sin(\beta_0 + \beta_1 t)$ |
| JC6/SJC6 | $\theta = 3.5 + 2.5sin(\theta_0 + \theta_1 t)$ | $\beta = 37.5 + 37.5sin(\beta_0 + \beta_1 t + \beta_2 t^2)$ |
| JC7/SJC7 | $\theta = 3.5 + 2.5sin(\theta_0 + \theta_1 t + \theta_2 t^2)$ | $\beta = 37.5 + 37.5sin(\beta_0 + \beta_1 t)$ |
| JC8/SJC8 | $\theta = 3.5 + 2.5sin(\theta_0 + \theta_1 t + \theta_2 t^2)$ | $\beta = 37.5 + 37.5sin(\beta_0 + \beta_1 t + \beta_2 t^2)$ |

JC: Joe-Clayton copula; SJC: survival Joe-Clayton copula;

```

**S3(f).** Patterns of the time-varying models for 2-parameter (survival) Joe-Frank copula (BB8 copula)

| Model | $\theta$ | $\beta$ |
|---|---|---|
| JF0/SJF0 | constant | constant |
| JF1/SJF1 | $\theta = 4.5 + 3.5sin(\theta_0 + \theta_1 t)$ | constant |
| JF2/SJF2 | $\theta = 4.5 + 3.5sin(\theta_0 + \theta_1 t + \theta_2 t^2)$ | constant |
| JF3/SJF3 | constant | $\beta = \dfrac{1 + 1e - 4}{2} + \dfrac{1 - 1e - 4}{2}\sin(\beta_0 + \beta_1 t)$ |
| JF4/SJF4 | constant | $\beta = \dfrac{1 + 1e - 4}{2} + \dfrac{1 - 1e - 4}{2}sin(\beta_0 + \beta_1 t + \beta_2 t^2)$ |
| JF5/SJF5 | $\theta = 4.5 + 3.5sin(\theta_0 + \theta_1 t)$ | $\beta = \dfrac{1 + 1e - 4}{2} + \dfrac{1 - 1e - 4}{2}\sin(\beta_0 + \beta_1 t)$ |
| JF6/SJF6 | $\theta = 4.5 + 3.5sin(\theta_0 + \theta_1 t)$ | $\beta = \dfrac{1 + 1e - 4}{2} + \dfrac{1 - 1e - 4}{2}sin(\beta_0 + \beta_1 t + \beta_2 t^2)$ |
| JF7/SJF7 | $\theta = 4.5 + 3.5sin(\theta_0 + \theta_1 t + \theta_2 t^2)$ | $\beta = \dfrac{1 + 1e - 4}{2} + \dfrac{1 - 1e - 4}{2}\sin(\beta_0 + \beta_1 t))$ |
| JF8/SJF8 | $\theta = 4.5 + 3.5sin(\theta_0 + \theta_1 t + \theta_2 t^2)$ | $\beta = \dfrac{1 + 1e - 4}{2} + \dfrac{1 - 1e - 4}{2}sin(\beta_0 + \beta_1 t + \beta_2 t^2)$ |

JF: Joe-Frank copula; SJF: survival Joe-Frank copula;

**Line(s) 203–205.**

**Authors.** The Gaussian copula was not used in this study because of its deficiency in describing dependencies of extremes (Renard and Lang, 2007).

**Referee.** The Authors are clearly considering the concept of Tail Dependence. Well, also the Frank family has no tail dependence, while the Clayton family only has lower tail dependence (possibly, of no interest here), the Gumbel family only has upper tail dependence, and the Student family has both lower and upper tail dependence (but they must be equal, and, most of all, they both must exist at the same time!). There are more suitable families of copulas for modeling extremes: again, the ones used by the Authors are simply those provided by Matlab, as (unfortunately, too) many practitioners do, preventing a reliable/valuable investigation and modeling of the phenomenon of

```

interest.

**Response:** As shown in the former response, we have enriched the kinds of candidate copulas.

**Line(s) 208, Eq. (5).**

Again, as above, you must justify the assumptions/relations shown in this equation. Why should the copula parameter change according to Eq. (5)? Did you carry out any investigation? What are the p-Values?

**Response:** We implemented the goodness of fit test for copulas based on Rosenblatt's transformation (Rosenblatt, 1952).

**Line(s) 212–214.**

**Authors.** The Corrected Akaike Information Criterion (AICc; Hurvich and Tsai, 1989) was employed to make a goodness-of-fit...

**Referee.** NO.This a typical fatal error of practitioners. The AIC (corrected or not) is only a selection criterion, not a GoF procedure. You must first show that a copula is statistically admissible, e.g. via suitable Monte Carlo Cramer-von Mises or Kolmogorov-Smirnov tests, as in the R package "copula". Then, and only then, you may compare (only) the admissible copulas (if any) and select the "best" one according to some suitable criterion (e.g., the AICc, the BIC, the NLL, etc...).

**Response:** We implemented the goodness of fit test for copulas based on Rosenblatt's transformation which can also be used to implement GOF tests for nonstationary copula.

```

**Line(s) 214–216.**

**Authors**. Obviously, the presence of nonstationarity in the copula parameter was determined by comparison of the AICc value.

**Referee.** This sentence is obscure. Are you saying that, since the non-stationary model performs better, then the phenomenon is non-stationary? If so, this makes no statistical and philosophical sense. It looks like you are using your models to "decide" how the real world should work: this is contrary to every scientific principle. This work is also bugged from an epistemological perspective.

**Response:** We make the LR tests which is a statistical test to check whether the trend in the parameter. If value of the LR tests is smaller than 5%, it recommends the trend existed in the parameter at 5% significance level. And then AICc criterion is used as model selection criterion.

**Line(s) 217–ff., Sec. 2.3.**

**Authors.** "2.3. Joint return period and risk analysis based on KEN's and AND's methods"

**Referee.** Multivariate failure probabilities have been well mathematically formalized in Salvadori et al. (2016), by originally defining and exploiting suitable Hazard Scenarios and copulas' relations. The Authors must take this work into serious account, and mention it.

**Response:** we have made multivariate hazard assessment based on the Average Annual Reliability (AAR) in the revised manuscript. And we also added the reference Salvadori

```

et al. (2016) when we wanted to use the joint exceedance probability under AND scenario.

**Line(s) 241, Eq. (7)**.

See the more general approach and discussion in (Salvadori et al., 2016, Eq.s (33)-(35)).

**Response:** we have made multivariate hazard assessment based on the Average Annual Reliability (AAR) in the revised manuscript. So this formula has been deleted.

**Line(s) 262, Eq. (11).**

Why in Eq. (11) the parameters of the marginals $F_X$ ,$F_Y$, used as arguments in the copula C, do not vary with time?

**Response:** it is an error of formula definition. We have taken this suggestion**.**

**Line(s) 268–269.**

**Authors.** The most likely event at the $T_0$ -year level can be calculated as (Graler et al., 2013)...

**Referee.** NO. The Most Likely technique was first introduced in Salvadori et al. (2011): always give credits to whom deserve credits. In addition, it is not the only possible one, as shown in the same paper (viz., the Component-wise Excess method). Moreover, further approaches are outlined in Corbella and Stretch (2012) and Salvadori et al. (2014). Why was the Most Likely approach chosen in this work?

**Response:** We have added the suggested reference to the manuscript. In order to

```

simplify the process of generating the extreme rainfall quantiles at each ARR level, the most-likely technique was implemented by choosing a certain quantile pair which has the largest joint probability than other combinations at the same level (Salvadori et al., 2011). And we proposed the algorithm to capture the numeric solution for the bivariate quantiles (shown in 332-349).

**Line(s) 289, Eq. (17).**

In Eq. (17), why is the modulus used? Obviously $\Delta R$ will always be positive. And even in this latter case, there is no quantification of any "scale" on which $\Delta R$ should be evaluated (when is it large? when is it small?). Such a number tells nothing to me.

**Response:** we deleted the formula because of adopting the ARR-based quantile estimation from hydrologic design.

**Line(s) 330–331.**

**Authors.** As shown in Figure 3, concurrences of univariate and bivariate trends, the nonstationarities in rainfall extremes can be detected at several stations...

**Referee.** This is simply because you use a 10% critical α-level, entailing a large probability of rejecting the Null Hypothesis of non-stationarity. For instance, at a standard 5% level, no one of the Univariate and Multivariate MK tests would fail, only two (at most three) out of 12 of the Univariate Pettitt tests would fail, and only one out of 6 of the Multivariate Pettitt tests would fail. In turn, the conclusions of the Authors are definitely questionable: in my opinion, in general, there is no clear statistical

```

evidence of non-stationarity (not to say if the standard 1% level were used, for in this case stationarity would be fully supported). Apparently, the Authors manipulate statistics according to their convenience, in order to show what they want to show.

**Response**: We considered more extreme series from more stations in the study area. Because of the insignificant trend denoted by reviewer #1 in the original manuscript, we changed the original 95-th percentile threshold for *Ps* to 0.90. After above modification of extreme values, the significant trend and change point at 5% significance level could be detected in 3 stations by nonparametric tests. And there is no station which can exhibit concurrences of univariate and bivariate trends, the nonstationarities according to the results in revised manuscript. The significance level of 5% is just the minimum standard. And it is not right and statistical objective to manipulate statistics according to our convenience, in order to show what we want to show. We have deeply realized the seriousness of the problem and corrected it in revised manuscript.

**Line(s) 353–354.**

**Authors.** The location parameter ($\mu$) and scale parameter ($\sigma$) are regarded as time variant, while the shape parameter $\kappa$ is time invariant...

**Referee.** As above, it is a dream to try and model time-variation of extremes using a constant shape parameter: it is the only one that matters in these kind of analyses. In addition, why should the other parameters vary according to Eq.s (3)–(5)? Simply because the same relations were used in other papers (again, without justification)? This

```
```

paper has no scientific objective grounds.

**Response**: as shown above, we have incorporate the potential trends into shape parameter. And K-S based GOF tests were conducted to verify its rationality.

**Line(s) 356–358.**

**Authors.** Despite the exception of Im for station 4, the shape parameter κ for most fitted models was in the interval of [-0.3,0.3]...

**Referee.** Tables 3 provide little statistical information, for no suitable confidence intervals are shown: this may have considerable consequences regarding the conclusions drawn by the Authors in later sections. In fact, they did not carry out any Monte Carlo analysis, and hence their results do not take into account the estimates' uncertainties (as if the Authors were stating the absolute Truth). To be clear, no confidence bands are plotted in later figures. This is not a scientific way of proceeding: the Authors must provide plots such as the ones shown in Salvadori et al. (2018), which may give an idea of the uncertainties at play (which may be huge, especially when a GEV is used, and may completely change the interpretation of the results, as I suspect). In addition, as above, some of the fitted values of the shape parameter would imply that the corresponding GEV is Upper Bounded, entailing that the corresponding variable cannot be an Extreme one. Furthermore, the fact that the range of the shape parameter is "in accordance with previous studies" is not significant and relevant at all (also given the fact that the range is quite large).

**Response**: The parametric bootstrap (Efron and Tibshirani, 1993) method was

```

implemented for parameter estimation and quantile estimation based on ARR level. It can provide a confidence bands (90% in this study) of parameter or the quantile estimation.

**Line(s) 359–360.**

**Authors.** The best fitted model was selected by performing the minimum DIC criterion combined with the Bayes factor (BF) test.

**Referee.** Again, you did not show that it is an admissible one! This work has no statistical bases. Line(s) 380–382.

**Authors.** Table 4(a)-(b) illustrates the results of best fitted copula, based on the minimum AICc and maximum logllikelihood value (LL).

**Referee.** Again, AIC and LL are not GoF criteria: the chosen models can be non-admissible! This work has no statistical bases.

**Response:** These two comments are replied by taking K-S based GOF tests for marginal distribution and A-D with Rosenblatt's transformation for copula models.

**Line(s) 433–435.**

**Authors.** Although the copula model for station 5 was stationary, it was regarded as a nonstationary model because of the marginal nonstationary GEVns-2 model for Ps or Im, which existed at other stations.

**Referee.** This makes no sense. The Authors do not understand the basic fact that the

```
```

dependence structure is independent of the marginals (as stated by Sklar's representation Theorem): even if the marginals are non-stationary, the copula may be stationary. The introduction of non-stationary copulas is arbitrary, without any justification: you cannot manipulate the results in this way!

**Response:** we have deleted this nonsense statement as suggestion.

**Line(s) 440–ff.**

**Authors.** Figure 5 shows isolines of Kendall return period and AND-based return period...

**Referee.** Given the uncertainties mentioned above (not considered by the Authors), I strongly suspect that the interpretation of the results shown in Figure 5 could be quite different if suitable confidence bands were plotted. This work lacks of elementary statistical bases.

**Response:** The parametric bootstrap (Efron and Tibshirani, 1993) method was implemented for parameter estimation and quantile estimation based on ARR level. It can provide a confidence bands (90% in this study) of parameter or the quantile estimation.

**Line(s)** 537–ff., Sec. 4.6.

In the light of the objections given above, the "Further discussion" section (4.6) makes no sense.

```

**Response:** we have modified the further discussion after we revised the manuscript as

reviewer 2 suggested.

```